# Tweedie Moment Projected Diffusions for Inverse Problems

**Benjamin Boys**                                                               *bb515@cam.ac.uk*
*Department of Engineering, University of Cambridge, Cambridge, United Kingdom*

**Jakiw Pidstrigach**                                          *jakiw.pidstrigach@stats.ox.ac.uk*
*Department of Statistics, University of Oxford, United Kingdom*

**Mark Girolami**                                                               *mag92@cam.ac.uk*
*Department of Engineering, University of Cambridge, Cambridge, United Kingdom*
*The Alan Turing Institute, London, United Kingdom*

**Sebastian Reich**                                                    *sereich@uni-potsdam.de*
*Universitat Potsdam, Berlin, Germany*

**Alan Mosca**                                                                    *alan@nplan.io*
*nPlan*

**O. Deniz Akyildiz**                                           *deniz.akyildiz@imperial.ac.uk*
*Imperial College London, London, United Kingdom*

**Reviewed on OpenReview:** *https://openreview.net/forum?id=4unJiOqrTE&noteId=WhEZC9S1r4*

## Abstract

Diffusion generative models unlock new possibilities for inverse problems as they allow for the incorporation of strong empirical priors in scientific inference. Recently, diffusion models are repurposed for solving inverse problems using Gaussian approximations to conditional densities of the reverse process via Tweedie's formula to parameterise the mean, complemented with various heuristics. To address various challenges arising from these approximations, we leverage higher order information using Tweedie's formula and obtain a statistically principled approximation. We further provide a theoretical guarantee specifically for posterior sampling which can lead to a better theoretical understanding of diffusion-based conditional sampling. Finally, we illustrate the empirical effectiveness of our approach for general linear inverse problems on toy synthetic examples as well as image restoration. We show that our method (i) removes any time-dependent step-size hyperparameters required by earlier methods, (ii) brings stability and better sample quality across multiple noise levels, (iii) is the only method that works in a stable way with variance exploding (VE) forward processes as opposed to earlier works.

## 1 Introduction

Due to the ease of scalability, diffusion models (Song et al., 2020; Ho et al., 2020) have received increased attention, producing a variety of improvements in the conditional generation setting such as adding classifier guidance (Dhariwal & Nichol, 2021), classifier free guidance (Ho & Salimans, 2022), conditional diffusion models (Batzolis et al., 2021; Karras et al., 2022; 2024) and DEFT (Doob's h-transform Efficient FineTuning) (Denker et al., 2024). These methods are useful for problems where paired $(\mathbf{x}_0, \mathbf{y})$ data are available, see, e.g., Saharia et al. (2022); Abramson et al. (2024). However, paired data is not always available, and there are cases where we would like to study alternatives to classical Bayesian approaches to solve inverse problems (Tarantola, 2005; Stuart, 2010) to build samplers for complicated conditional distributions with a known observation operator mapping $\mathbf{x}_0$ to the observed data $\mathbf{y}$ that is used to express a known likelihood $p(\mathbf{y}|\mathbf{x}_0)$. With their

flexibility, diffusion models replace handcrafted priors on the latent signal with pretrained and strong empirical priors. For example, given a latent signal $\mathbf{x}_0$ (say a face image), we can *train* a diffusion model to sample from the prior $p(\mathbf{x}_0)$. The idea for solving inverse problems is to leverage the extraordinary modelling power of diffusion models to learn samplers for priors and couple this with a given likelihood $p(\mathbf{y}|\mathbf{x}_0)$ for a given data $\mathbf{y}$ to sample from the posterior $p(\mathbf{x}_0|\mathbf{y})$ for inverse problems. However, designing a diffusion model for the posterior comes with challenges due to intractability. Despite its challenges, this approach has been recently taking off with lots of activity in the field, e.g. for compressed sensing (Bora et al., 2017; Kadkhodaie & Simoncelli, 2021), projecting score-based stochastic differential equations (SDEs) (Song et al., 2021b), gradient-based approaches (Daras et al., 2022; Chung et al., 2022b), magnetic resonance imaging (MRI) by approximating annealed Langevin dynamics with approximate scores (Jalal et al., 2021), image restoration (Kawar et al., 2022), score-based models as priors but with a normalizing flow approach (Feng et al., 2023), variational approaches (Mardani et al., 2023; Feng & Bouman, 2023). Most relevant ideas to us, which we will review in detail in Section 5, use Tweedie's formula (Efron, 2011; Laumont et al., 2022) to approximate the smoothed likelihood, e.g. Diffusion posterior sampling (DPS) (Chung et al., 2022a; 2023) and pseudo-guided diffusion models (ΠGDM) (Song et al., 2023). Similar approaches are also exploited using singular-value decomposition (SVD) based approaches (Kawar et al., 2021). In this work, we develop an approach which builds a tighter approximation to optimal formulae for approximating the scores of the posterior diffusion model.

This paper is devoted to developing novel methods to solving inverse problems, given a latent (target) signal $\mathbf{x}_0 \in \mathbb{R}^{d_x}$, noisy observed data $\mathbf{y} \in \mathbb{R}^{d_y}$, a known linear observation map $\mathbf{H}$, and a *pretrained* diffusion prior. The main tool we use is Tweedie's formula to obtain both the mean and the covariance for approximating diffused likelihoods, to be used for building the final posterior score approximation. This is as opposed to previous works which only utilised first moment approximations using Tweedie's formula (Chung et al., 2022a; Song et al., 2023). We show that utilising covariance approximation, with moment projections, provides a principled scheme with improved performance, which we term *Tweedie Moment Projected Diffusions* (TMPD).

To demonstrate our method briefly, Figure 1 demonstrates a sampling scenario of a *Gaussian random field* (GRF) whose mean and variance entries are plotted under "Analytic" column[1]. We demonstrate the approximations under this setting provided by our method (TMPD) and its diagonal (cheaper) version (DTMPD), compared with ΠGDM (Song et al., 2023), and DPS (Chung et al., 2022a). The figure demonstrates the optimality of our method: Our first and second moment approximations become exact in this case. This results in a drastic performance improvement stemming from the statistical optimality of our method for near-Gaussian settings and also unlocks a possible line for theoretical research for understanding similar diffusion models for inverse problems.

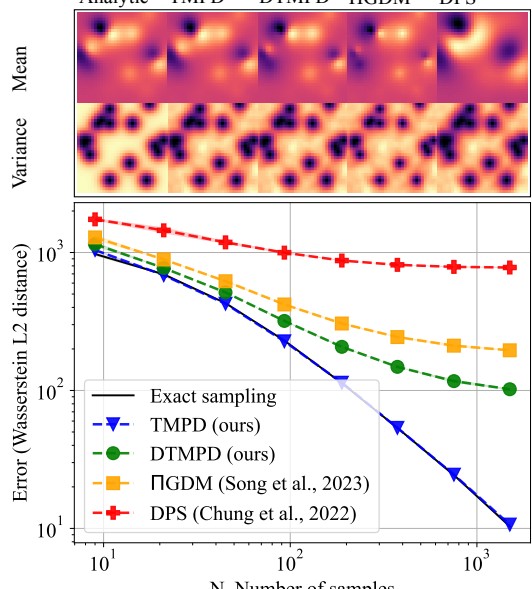

Figure 1: Error to target posterior for a *Gaussian random field*. (Top row) visualisation of the empirical mean and variance of the 1500 samples that were used to compute this error against the analytical moments. (Bottom) Wasserstein distances of different methods w.r.t. sample size. For details, see Appendix E.2.

In what follows, we will first introduce the technical background in Section 2 and then describe TMPD in detail in Section 3. We will then provide some theoretical results about our method in Section 4 and provide a discussion to closely related work in literature in Section 5. Finally, Section 6 will present experiments on Gaussian

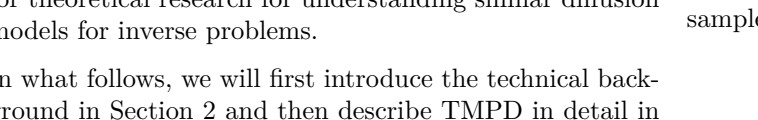

---

[1]We only plot the variances for visualisation while the GRF has a full covariance.

mixtures, image inpainting and super-resolution, demonstrating quantitative and qualitative improvements provided by TMPD.

## 2 Technical Background

In score based generative models (SGMs) (Song et al., 2020) and denoising diffusion probabilistic models (DDPM) (Ho et al., 2020), the goal is to sample from a target distribution $p_0 := p_{\text{data}}$. To that end, an artificial path $p_t$ is introduced, with the property that $p_t$ will approach $\mathcal{N}(0, I)$ for large $t$, i.e., $p_t \to \mathcal{N}(0, I_d)$ as $t \to \infty$. Then, one learns to reverse this process, in order to transform samples from a standard normal distribution into samples from $p_{\text{data}}$. More recent advances in generative modelling include methods with artificial paths based on stochastic interpolants or continuous normalizing flows, see, e.g., Albergo & Vanden-Eijnden (2022); Lipman et al. (2022). Conditional flow matching draws inspiration from the denoising score matching approach, but generalizes to matching vector fields directly. We have focused our approach to denoising-diffusion models in the SGM paradigm because of readily available pretrained diffusion models.

In the SGM paradigm, a stochastic differential equation (SDE) is used to noise the data, and the interpolation parameter $t$ will take continuous values in $t \in [0, T]$. In the DDPM setting, $t$ is discrete. However, the DDPM Markov chain can be seen as a discretization of the SDE (Song et al., 2020).

Recent developments have improved the noising schedule of score-based diffusion models for image data that parameterise the transition kernels of the forward process in terms of the signal-to-noise ratio $\sigma_t^2$, $p_t(\mathbf{x}_t|\mathbf{x}_0) = \mathcal{N}(\mathbf{x}_t; s_t\mathbf{x}_0, s_t^2\sigma_t^2\mathbf{I}_{d_x})$ (Karras et al., 2022; 2024). In this paper, we focus derivations on the Variance Preserving (VP) SDE formulation, although our approach can be generalised and derived for other SDEs, such as the Variance Exploding (VE) SDE (see Appendix C). The VP transition kernel that we choose to focus our derivation is given by setting

$$p_t(\mathbf{x}_t|\mathbf{x}_0) = \mathcal{N}(\mathbf{x}_t; \sqrt{\alpha_t}\mathbf{x}_0, v_t\mathbf{I}_{d_x}) \quad \text{where} \quad \alpha_t := \exp\left(-\int_0^t \beta(s)\mathrm{d}s\right) \quad \text{and} \quad v_t := 1 - \alpha_t,$$

which are the transition kernels for the time-rescaled Ornstein-Uhlenbeck process:

$$\mathrm{d}\mathbf{x}_t = -\frac{1}{2}\beta(t)\mathbf{x}_t\mathrm{d}t + \sqrt{\beta(t)}\mathrm{d}\mathbf{w}_t, \quad \mathbf{x}_0 \sim p_0 = p_{\text{data}}. \tag{1}$$

The corresponding reverse SDE is then given by

$$\mathrm{d}\mathbf{z}_t = \frac{1}{2}\beta(T-t)\mathbf{z}_t\mathrm{d}t + \beta(T-t)\nabla_{\mathbf{z}_t}\log p_{T-t}(\mathbf{z}_t)\mathrm{d}t + \sqrt{\beta(T-t)}\mathrm{d}\bar{\mathbf{w}}_t, \quad \mathbf{z}_0 \sim p_T.$$

A parameterisation that performs well in practice is $\beta(t) = \beta_{\min} + t(\beta_{\max} - \beta_{\min})$. In the diffusion modelling literature, the time-rescaled OU process is also sometimes called a Variance Preserving SDE. This is not the only SDE that is suitable for the forward process. See Appendix C for details on a time-rescaled Brownian motion (Variance Exploding SDE (Song et al., 2020)).

There are two usual approximations to solve the SDE in Equation 2. First, we do not know $p_T$, since it is a noised version of the distribution $p_{\text{data}}$. However for $T$ large enough, we can approximate $p_T \approx p_{\text{ref}} = \mathcal{N}(0, I_{d_x})$. We also do not have $\nabla\log p_{T-t}$ which we need for the drift in Equation 2. This can be circumvented by approximating the drift using score-matching techniques (Hyvärinen, 2005; Ho et al., 2020). These methods construct an estimate of the score function by solving the score matching problem in the form of $\mathbf{s}_\theta(\mathbf{x}_t, t) \approx \nabla_{\mathbf{x}_t}\log p_t(\mathbf{x}_t)$. This score can also be used in the setting of DDPM (Ho et al., 2020).

### 2.1 Conditional sampling for the linear inverse problem

In the preceding section we introduced diffusion models as a method to sample from a target distribution $p_{\text{data}}$. We now suppose that we have access to measurements, or observations $\mathbf{y} \in \mathbb{R}^{d_y}$ of $\mathbf{x}_0 \in \mathbb{R}^{d_x}$:

$$\mathbf{y} = \mathbf{H}\mathbf{x}_0 + \mathbf{u}, \quad \mathbf{u} \sim \mathcal{N}(0, \sigma_y^2\mathbf{I}_{d_y}). \tag{2}$$

We would then be interested in sampling from the conditional distribution of $\mathbf{x}_0$ given $\mathbf{y}$, i.e., $p_{\text{data}}(\,\cdot\,|\mathbf{y})$. To that end, we have to modify the reverse SDE. We would like to sample from the reverse SDE targeting $p_{\text{data}}(\,\cdot\,|\mathbf{y})$, instead of the one targeting $p_{\text{data}}$.

Optimally, we would want to replace the score $\nabla_{\mathbf{z}_t} \log p_{T-t}(\mathbf{z}_t)$ in Equation 2 with the posterior score $\nabla_{\mathbf{z}_t} \log p_{T-t|\mathbf{y}}(\mathbf{z}_t|\mathbf{y})$. Written in terms of the forward process, this coincides with

$$\nabla_{\mathbf{x}_t} \log p_{t|\mathbf{y}}(\mathbf{x}_t|\mathbf{y}) = \nabla_{\mathbf{x}_t} \log p_t(\mathbf{x}_t) + \nabla_{\mathbf{x}_t} \log p_{\mathbf{y}|t}(\mathbf{y}|\mathbf{x}_t). \tag{3}$$

The term $p_{\mathbf{y}|t}(\mathbf{y}|\mathbf{x}_t)$ is given by the integral

$$p_{\mathbf{y}|t}(\mathbf{y}|\mathbf{x}_t) = \int p_{\mathbf{y}|0}(\mathbf{y}|\mathbf{x}_0)p_{0|t}(\mathbf{x}_0|\mathbf{x}_t)\mathrm{d}\mathbf{x}_0, \tag{4}$$

which involves a marginalization over $\mathbf{x}_0$. The above integral is difficult to evaluate since the term $p_{0|t}(\mathbf{x}_0|\mathbf{x}_t)$ is only defined implicitly through running the diffusion model. One way around this is to train a neural network to directly approximate $\nabla \log p_{t|\mathbf{y}}(\mathbf{x}_t|\mathbf{y})$ (Batzolis et al., 2021; Karras et al., 2022). Alternatively, if one already has access to an approximation of $\nabla_{\mathbf{x}_t} \log p_t(\mathbf{x}_t)$, one can train an auxiliary network to approximate the term $\nabla \log p_{\mathbf{y}|t}(\mathbf{y}|\mathbf{x}_t)$ in Equation 3, (Dhariwal & Nichol, 2021; Denker et al., 2024). However, these methods can be time and training-data intensive, as it is necessary to retrain networks for each conditional task as well have access to paired training data from the joint distribution of $(\mathbf{x}_0, \mathbf{y})$. Alternatively, one could try to do a Monte-Carlo approximation of the score corresponding to Equation 4. But this needs evaluating the probability flow ODE together with its derivative (Song et al., 2020, Section D.2) for each sample, which is prohibitive, also suffers from high variance (Mohamed et al., 2020, Section 3).

## 3 Tweedie Moment Projected Diffusions

In this section, we first introduce *Tweedie moment projections* in Section 3.1 below. Our method relies on the approximation $p_{0|t}(\mathbf{x}_0|\mathbf{x}_t) \approx \mathcal{N}\left(\mathbf{x}_0; \mathbf{m}_{0|t}(\mathbf{x}_t), \mathbf{C}_{0|t}(\mathbf{x}_t)\right)$ to make the sampling process tractable. In that case, since the conditional distribution of $\mathbf{y}$ given $\mathbf{x}_0$ is also Gaussian, we can *compute the integral in Equation 4 analytically* — $p_{\mathbf{y}|t}(\mathbf{y}|\mathbf{x}_t)$ will just be another Gaussian in that case, its mean and covariance being determined through $\mathbf{m}_{0|t}$, $\mathbf{C}_{0|t}$, $\mathbf{H}$ and $\sigma_y$. In particular, we can then use this Gaussian to approximate $\nabla \log p_{\mathbf{y}|t}(\mathbf{y}|\mathbf{x}_t)$, since the score of a Gaussian is available in closed form.

### 3.1 Tweedie moment projections

Instead of just approximating the variance of $p_{0|t}(\mathbf{x}_0|\mathbf{x}_t)$ heuristically, we approximate it by projecting onto the closest Gaussian distribution using Tweedie's formula for the second moment. Our approximation at this stage consists of two main steps: (i) Find the mean and covariance of $p_{0|t}(\mathbf{x}_0|\mathbf{x}_t)$ using Tweedie's formula, and (ii) approximate this density with a Gaussian using the mean and covariance of $p_{0|t}(\mathbf{x}_0|\mathbf{x}_t)$ (moment projection). Due to this approximation, we will refer to the resulting methods as *Tweedie Moment Projected Diffusions* (TMPD). We will first introduce Tweedie's formula for the mean and covariance and then describe the moment projection.

**Proposition 1** (Tweedie's formula). *Let $\mathbf{m}_{0|t}$ and $\mathbf{C}_{0|t}$ be the mean and the covariance of $p_{0|t}(\mathbf{x}_0|\mathbf{x}_t)$, respectively. Then given the marginal density $p_t(\mathbf{x}_t)$, the mean is given as*

$$\mathbf{m}_{0|t} = \mathbb{E}[\mathbf{x}_0|\mathbf{x}_t] = \frac{1}{\sqrt{\alpha_t}}(\mathbf{x}_t + v_t \nabla_{\mathbf{x}_t} \log p_t(\mathbf{x}_t)), \tag{5}$$

*and the covariance $\mathbf{C}_{0|t}$ is given by*

$$\begin{aligned}
\mathbf{C}_{0|t} &= \mathbb{E}\left[(\mathbf{x}_0 - \mathbf{m}_{0|t})(\mathbf{x}_0 - \mathbf{m}_{0|t})^\top \mid \mathbf{x}_t\right] \\
&= \frac{v_t}{\alpha_t}(\mathbf{I}_{d_x} + v_t \nabla^2 \log p_t(\mathbf{x}_t)) = \frac{v_t}{\sqrt{\alpha_t}} \nabla_{\mathbf{x}_t} \mathbf{m}_{0|t}.
\end{aligned} \tag{6}$$

The proof is an adaptation of Meng et al. (2021, Theorem 1), see Appendix A.1. While $\mathbf{m}_{0|t}$ and $\mathbf{C}_{0|t}$ give us the moments of the density $p_{0|t}(\mathbf{x}_0|\mathbf{x}_t)$, we do not have the exact form of this density. At this stage, we employ *moment projection*, i.e., we choose the closest Gaussian in Kullback-Leibler (KL) divergence which is a distribution with the same first and second moments, as formalised next.

**Proposition 2** (Moment projection). *Let $p_{0|t}(\mathbf{x}_0|\mathbf{x}_t)$ be a distribution with mean $\mathbf{m}_{0|t}$ and covariance $\mathbf{C}_{0|t}$. Let $\hat{p}_{0|t}(\mathbf{x}_0|\mathbf{x}_t)$ be the the closest Gaussian in KL divergence to $p_{0|t}(\mathbf{x}_0|\mathbf{x}_t)$, i.e.,*

$$\hat{p}_{0|t}(\mathbf{x}_0|\mathbf{x}_t) = \arg\min_{q \in \mathcal{Q}} D_{\mathrm{KL}}(p_{0|t}(\mathbf{x}_0|\mathbf{x}_t)||q), \tag{7}$$

*where $\mathcal{Q}$ is the family of multivariate Gaussian distributions. Then*

$$\hat{p}_{0|t}(\mathbf{x}_0|\mathbf{x}_t) = \mathcal{N}(\mathbf{x}_0; \mathbf{m}_{0|t}, \mathbf{C}_{0|t}). \tag{8}$$

This is a well-known moment matching result, see, e.g., Bishop (2006, Section 10.7). Merging Propositions 1 and 2 leads to the following *Tweedie moment projection*:

$$p_{0|t}(\mathbf{x}_0|\mathbf{x}_t) \approx \mathcal{N}\left(\mathbf{x}_0; \mathbf{m}_{0|t}, \frac{v_t}{\sqrt{\alpha_t}}\nabla_{\mathbf{x}_t}\mathbf{m}_{0|t}\right), \tag{9}$$

where $\mathbf{m}_{0|t}$ is given in Proposition 1. In the next section, we demonstrate how to use this approximation to obtain approximate likelihoods.

### 3.2 Tweedie Moment Projected Likelihood Approximation

We next use the approximation in Equation 9 to compute the following integral *analytically*

$$\begin{aligned} p_{\mathbf{y}|t}(\mathbf{y}|\mathbf{x}_t) &\approx \int p_{\mathbf{y}|0}(\mathbf{y}|\mathbf{x}_0)\hat{p}_{0|t}(\mathbf{x}_0|\mathbf{x}_t)\mathrm{d}\mathbf{x}_t \\ &= \mathcal{N}\left(\mathbf{y}; \mathbf{H}\mathbf{m}_{0|t}, \mathbf{H}\mathbf{C}_{0|t}\mathbf{H}^\top + \sigma_{\mathbf{y}}^2\mathbf{I}_{d_{\mathbf{y}}}\right). \end{aligned} \tag{10}$$

Let us recall that mean and covariance terms are a function of $\mathbf{x}_t$ by making them explicit in the notation, i.e., $\mathbf{m}_{0|t}(\mathbf{x}_t)$ and $\mathbf{C}_{0|t}(\mathbf{x}_t)$ for the mean and covariance respectively. To compute $\nabla_{\mathbf{x}_t}\log p_{\mathbf{y}|t}(\mathbf{y}|\mathbf{x}_t)$, we require further approximations since we have $\mathbf{x}_t$ dependence in both the mean and the covariance of Equation 10 which is computationally infeasible to differentiate through. For this reason, we treat the matrix $\mathbf{C}_{0|t}$ like a *constant* w.r.t. $\mathbf{x}_t$ when computing the gradient (which is the case if $p_{\mathrm{data}}$ is Gaussian). For non-Gaussian $p_{\mathrm{data}}$, this results in a computationally efficient sampler using $\mathbf{C}_{0|t}$ as a preconditioner for the step size, as otherwise, the resulting terms can be expensive to compute. This leads to an approximation of the gradient

$$\begin{aligned} f^{\mathbf{y}}(\mathbf{x}_t) &:= \nabla_{\mathbf{x}_t}\mathbf{m}_{0|t}(\mathbf{x}_t)\mathbf{H}^\top(\mathbf{H}\mathbf{C}_{0|t}(\mathbf{x}_t)\mathbf{H}^\top + \sigma_{\mathbf{y}}^2\mathbf{I}_{d_{\mathbf{y}}})^{-1}(\mathbf{y} - \mathbf{H}\mathbf{m}_{0|t}(\mathbf{x}_t)) \\ &\approx \nabla_{\mathbf{x}_t}\log p_{\mathbf{y}|t}(\mathbf{y}|\mathbf{x}_t), \end{aligned} \tag{11}$$

where $\nabla_{\mathbf{x}_t}$ only operates on $\mathbf{m}_{0|t}$ in Equation 11. Another way to write this approximation is to use Equation 6, which leads to

$$f^{\mathbf{y}}(\mathbf{x}_t) := \frac{\sqrt{\alpha_t}}{v_t}\mathbf{C}_{0|t}(\mathbf{x}_t)\mathbf{H}^\top(\mathbf{H}\mathbf{C}_{0|t}(\mathbf{x}_t)\mathbf{H}^\top + \sigma_{\mathbf{y}}^2\mathbf{I}_{d_{\mathbf{y}}})^{-1}(\mathbf{y} - \mathbf{H}\mathbf{m}_{0|t}(\mathbf{x}_t))$$

### 3.3 Algorithms

Plugging the approximation in Equation 11 into the reverse SDE in Equation 2 together with the prior score as described in Section 2.1 results in a TMPD for conditional sampling to solve inverse problems. The SDE we will approximate numerically to sample from the conditional distribution is given by

$$\mathrm{d}\mathbf{z}_t = \frac{1}{2}\beta(T-t)\mathbf{z}_t\mathrm{d}t + \beta(T-t)(\nabla_{\mathbf{z}_t}\log p_{T-t}(\mathbf{z}_t) + f^{\mathbf{y}}_{T-t}(\mathbf{z}_t))\mathrm{d}t + \sqrt{\beta(T-t)}\mathrm{d}\bar{\mathbf{w}}_t \tag{12}$$

where $\mathbf{z}_0 \sim p_T$ and $f_t^{\mathbf{y}}(\mathbf{z}_t) \approx \nabla \log p_{\mathbf{y}|T-t}(\mathbf{y}|\mathbf{z}_t)$ is our approximation to the data likelihood, given by Equation 11. We call this SDE the *TMPD SDE*.

We have two options to convert TMPD SDE into implementable methods: (1) score-based samplers (Song & Ermon, 2020), which we abbreviate as TMPD since they are Euler-Maruyama discretizations of the TMPD SDE; and (2) denoising diffusion models (TMPD-D). The denoising diffusion approach is derived from approximate reverse Markov chains and is the approach of DDPM and DPS methods (Ho et al., 2020; Chung et al., 2022a). We note that the Gaussian projection can be used in this discrete setting, assuming that the conditional density is available analytically as in Ho et al. (2020), and can be written as $p_{n|0}(\mathbf{x}_n|\mathbf{x}_0) = \mathcal{N}(\mathbf{x}_n; \sqrt{\alpha_n}\mathbf{x}_0, v_n\mathbf{I}_{d_x})$. The idea is to update the unconditional mean $\mathbf{m}_{0|n}(\mathbf{x}_n)$ of the density $p_{0|n}(\mathbf{x}_0|\mathbf{x}_n)$ with a Bayesian update: $p(\mathbf{x}_0|\mathbf{x}_n, \mathbf{y}) \propto p(\mathbf{y}|\mathbf{x}_n)p_{0|n}(\mathbf{x}_0|\mathbf{x}_n)$. Given a similar formulation as above, assuming we have a readily available approximation $p_{0|n}(\mathbf{x}_0|\mathbf{x}_n) \approx \mathcal{N}(\mathbf{x}_0; \mathbf{m}_{0|n}, \mathbf{C}_{0|n})$ and a likelihood similar to Equation 10 where $t$ can be replaced by $n$, we can compute the moments of $p(\mathbf{x}_0|\mathbf{x}_n, \mathbf{y})$ analytically, which we denote $\mathbf{m}_{0|n}^{\mathbf{y}}$ and $\mathbf{C}_{0|n}^{\mathbf{y}}$. The Bayes update for Gaussians gives (Bishop, 2006)

$$\mathbf{m}_{0|n}^{\mathbf{y}} = \mathbf{m}_{0|n} + \mathbf{C}_{0|n}\mathbf{H}^\top(\mathbf{H}\mathbf{C}_{0|n}\mathbf{H}^\top + \sigma_y^2\mathbf{I}_{d_x})^{-1}(\mathbf{y} - \mathbf{H}\mathbf{m}_{0|n}). \tag{13}$$

Incorporating Equation 13 for $n = N - 1, \ldots, 0$ into the usual Ancestral sampling (Ho et al., 2020) steps leads to Algorithm 1, termed TMPD-Denoising (TPMD-D). The update in Equation 13 can be used in any discrete sampler such as denoising diffusion implicit models (DDIM) (Song et al., 2021a).

### 3.4 Computationally cheaper approximation of Moment Projection

We show in our experiments promising results for TMPD motivating the exploration of less computationally expensive approximations to the full Jacobian. In particular, we empirically study a computationally inexpensive method that applies to inpainting and super-resolution, below.

To make the computational cost of TMPD smaller, we can make an approximation of the Gaussian Projection that requires fewer vector-Jacobian products and does not require linear solves. One approximation that we found useful for sampling from high dimensional diffusion models, e.g., high resolution images, is denoted here as diagonal Tweedie Moment Projection (DTMPD). Instead of the full second moment, DTMPD uses the diagonal of the second moment $\nabla_{\mathbf{x}_t}\mathbf{m}_{0|t} \approx \mathrm{diag}(\nabla_{\mathbf{x}_t}\mathbf{m}_{0|t})$. Intuitively, this approximation will perform well empirically since it is a similar approximation to ΠGDM that assumes dimensional independence of the distribution $p(\mathbf{x}_0|\mathbf{x}_t)$, but unlike ΠGDM, this diagonal approximation is the same as using the closest dimensionally independent Gaussian in KL divergence to $p_{0|t}(\mathbf{x}_0|\mathbf{x}_t)$.

The biggest drawback of our method is that without further approximation, it doesn't scale up to the high dimensions of image data. This is because even calculating the diagonal of a Jacobian requires computing $d_x$ vector-Jacobian products since in general every element of the Jacobian at a location $\mathbf{x}_t$, $\nabla_{\mathbf{x}_t}\mathbf{m}_{0|t}$ depends on every element of $\mathbf{x}_t$. Therefore we must resort to a further approximation that exploits knowledge of the observation operator $\mathbf{H}$.

For the cases of super-resolution and inpainting, a further approximation that allows scaling up to the dimensions of image data is approximating the diagonal of the Jacobian by the row sum of the Jacobian which only requires a single vector-Jacobian product and brings the memory and time complexity of DTMPD down to that of ΠGDM. We exploit the sparsity of $\mathbf{H}$ to make the rowsum approximation of the diagonal more accurate by masking out (zeroing) the values in the vector-Jacobian product that that will not contribute to the diagonal of $\mathbf{H}\mathbf{C}_{0|n}\mathbf{H}^\top$. We discuss a justification of this approximation in Ap. E.1. We use this approximation in the image experiments and find that in practice it is only $(1.5 \pm 0.1)\times$ slower than ΠGDM and DPS across all of our experiments (Sec. 6), with competitive sample quality for noisy inverse problems and without the need for expensive hyperparameter tuning. Finally, we note a very recent work (Rozet et al., 2024) that circumvents our heuristic by applying the conjugate gradient (CG) method, unlocking using the approximation Eq. 11 to be used in practice for non-sparse $\mathbf{H}$ (see Ap. E.1 for more details).

# 4 Theoretical Guarantees

Because of the approximations, our method, as well as ΠGDM Song et al. (2023) and DPS (Chung et al., 2022a) do not sample the exact posterior for general prior distributions. Therefore, one cannot hope for these methods to sample the true posterior and a priori it is not even clear how the sampled distribution relates to the true posterior. Without further justification, such methods should only be interpreted as *guidance methods*, where paths are guided to regions where a given observation **y** is more likely, not as posterior sampling methods.

We justify our approximation by showing that the TMPD-SDE in Equation 12 is able to sample the exact posterior in the Gaussian case. One can see that this contrasts with ΠGDM and DPS in our numerical experiments or by explicitly evaluating their approximations on simple one-dimensional examples.

**Proposition 3** (Gaussian data distribution). *Assume that $p_{\mathrm{data}}$ is Gaussian. Then, the posterior score expression using Equation 11 is exact, i.e., if there are no errors in the initial condition and drift approximation $\mathsf{s}_\theta(\mathbf{x}_t, t) = \nabla_{\mathbf{x}_t} \log p_t(\mathbf{x}_t)$, the TMPD will sample $p_{\mathrm{data}}(\cdot|\mathbf{y})$ at its final time.*

The proof is given in Appendix B.1. However, most distributions will not be Gaussian. The following theorem generalizes the above proposition to *non-Gaussian distributions*, as long as they have a density with respect to a Gaussian. We study how close our sampled measure will be to the true posterior distribution and give explicit bounds on the total variation distance in terms of the regularity properties of the density:

**Theorem 1** (General data distribution). *Assume that the data distribution $p_{\mathrm{data}}$ can be written as*

$$p_{\mathrm{data}}(\mathbf{x}_0) = \exp(\Phi(\mathbf{x}_0))\mathcal{N}(\mathbf{x}_0; \boldsymbol{\mu}_0, \boldsymbol{\Sigma}_0), \tag{14}$$

*for some $\boldsymbol{\mu}_0$ and $\boldsymbol{\Sigma}_0$. We furthermore assume that for some $M \geq 1$, it holds that $1/M \leq \exp(\Phi(\mathbf{x})) \leq M$ and $\|\nabla_{\mathbf{x}} \Phi(\mathbf{x})\| \leq L$ for all $\mathbf{x} \in \mathbb{R}^{d_x}$. Then*

$$\|p_{\mathrm{data}}(\cdot|\mathbf{y}) - q_T(\cdot)\|_{\mathrm{TV}} \leq C(1 + T^{1/2})\left((M^{5/2} - 1)(L^{1/2} + 1) + L^{1/2}\right), \tag{15}$$

*where $q_t$ denotes the law of the corresponding reverse-time process for Equation 12 at time t and the constant C that only depends on $\mathbf{y}, \mathbf{H}, \sigma_y, \boldsymbol{\mu}_0$ and $\boldsymbol{\Sigma}_0$.*

See Appendix B.2 for a proof.

In the limit, when $p_{\mathrm{data}}$ becomes more similar to a Gaussian, the $\Phi$ in Equation 14 converges to zero, and therefore $M \to 1$ and $L \to 0$. In particular, the right hand side in Equation 15 converges to 0 and we recover the result of Proposition 3. When $p_{\mathrm{data}}$ is not Gaussian, the right hand side of Equation 15 gives us an upper bound of our sample distribution to the true posterior.

# 5 Related Work

In this section, we review two closely related methods, that we use as benchmarks, and summarise the relationship between them.

The first work by Chung et al. (2022a), abbreviated DPS-D here[2] introduced the use of Tweedie's formula to approximate $p_t(\mathbf{x}_0|\mathbf{x}_t)$ with a Dirac delta (point) distribution centred at $\mathbf{m}_{0|t}$. In our framework this corresponds to choosing a zero covariance, i.e.,

$$\mathbf{m}_{0|t}^{\mathrm{DPS-D}} = \mathbf{m}_{0|t} \qquad \text{and} \qquad \mathbf{C}_{0|t}^{\mathrm{DPS-D}} = 0. \tag{16}$$

In the work Song et al. (2023), abbreviated ΠGDM-D here[3] the same estimator for the mean is chosen. However, the variance is set to a multiple of the identity, corresponding to choices

$$\mathbf{m}_{0|t}^{\mathrm{\Pi G}} = \mathbf{m}_{0|t} \qquad \text{and} \qquad \mathbf{C}_{0|t}^{\mathrm{\Pi G}} = r_t^2 \mathbf{I}_{d_x} \tag{17}$$

---

[2]Since the authors run DDPM-type sampler, we re-abbreviate DPS as DPS-D in this work (as DPS approximation itself can also be run with Euler-Maruyama schemes).

[3]Since the authors run DDIM-type sampler, we re-abbreviate ΠGDM as ΠGDM-D in this work (as ΠGDM approximation itself can also be run with Euler-Maruyama schemes).

The choice of $r_t$ is such that it matches the variance of the reverse SDE if $p_{\text{data}}$ would be a standard normal distribution. Since they employ a different forward SDE to ours (variance exploding SDE), $r_t$ is set, when the variance of the data is 1, to be equal to $v_t/(v_t + 1)$. In our case, with the OU/variance preserving SDE as a forward process, $r_t$ would be equal to $v_t/(v_t + \alpha_t)$, when the variance of the data is 1.

Another relevant very recent work by Finzi et al. (2023) arrives at the approximation matching ours in the context of modelling physical constraints. However, we focus on general linear inverse problems outside the physical domain in this paper together with a novel theoretical result. Also, we note the work of Stevens et al. (2023) who consider the maximum-a-posteriori (MAP) approach to find the moments of $p_{0|t}$.

Finally, sequential Monte Carlo (SMC) methods are particle-based posterior sampling schemes that are asymptotically exact in the limit of infinitely many particles. Cardoso et al. (2023) target the posterior for linear inverse problems, and our first experiment makes a comparison to Cardoso et al. (2023) for a Gaussian Mixture model. However, for high dimensional data (such as images), this method can exhibit the well studied SMC problem of weight degeneracy, where particle weights collapse. We note a recent state of the art work (Wu et al., 2024) that tackles weight degeneracy by introducing the Twisted Diffusion Sampler (TDS), with heuristic twisting functions approximating $p_{\mathbf{y}|t}(\mathbf{y}|\mathbf{x}_t)$ that increase computational efficiency of SMC. The method is still asymptotically exact as long as the approximation of $p_{\mathbf{y}|t}(\mathbf{y}|\mathbf{x}_t)$ converges to $p(\mathbf{y}|\mathbf{x}_0)$ as $t$ approaches 0. The authors note that the efficiency of TDS (Wu et al., 2024) depends on how closely the twisting function approximates the exact likelihood. TMPD (Equation 10) could be a good twisting candidate.

## 6 Experiments

In this section, we demonstrate our results as well as the peformance of other approximations to the likelihood provided in Chung et al. (2022a); Song et al. (2023). In particular, we perform comparisons for two of our methods TMPD (an SGM using our approximation) and TMPD-D (a DDPM sampler using Equation 13). We compare these to DPS (an SGM sampler using the posterior approximation in Equation 16), DPS-D (Chung et al., 2022a) (a DDPM-type sampler using Equation 16), ΠGDM (Song et al., 2023) (an SGM sampler using the posterior approximation in Equation 17), and finally ΠGDM-D (a DDIM-type sampler using Equation 17, but in our experiments a DDPM-type sampler since we set the DDIM hyperparameter $\eta = 1.0$ which is defined in Algorithm 1 by Song et al. (2021a) who show that this is equivalent to a DDPM-type sampler).

The code for all of the experiments and instructions to run them are available at github.com/bb515/tmpdjax and github.com/bb515/tmpdtorch.

### 6.1 Gaussian Mixture Model

We now demonstrate a nonlinear SDE example and follow the Gaussian mixture model example of Cardoso et al. (2023) where the data distribution $p_0(\mathbf{x}_0)$ is a mixture of 25 Gaussian distributions. The means and variances of the components of the mixture are given in Appendix E.3. In this case, for each choice of observation $\mathbf{y}$, observation map $\mathbf{H}$ and measurement noise standard deviation $\sigma_{\mathbf{y}}$, the target posterior can be computed explicitly (see Appendix E.3).

To investigate the performance of posterior sampling methods, for each pair of dimensions and observation noise $(d_{\mathbf{x}}, d_{\mathbf{y}}, \sigma_{\mathbf{y}}) \in \{8, 80, 800\} \times \{1, 2, 4\} \times \{10^{-2}, 10^{-1}, 10^0\}$ we randomly generate multiple measurement models $(\mathbf{y}, \mathbf{H}) \in \mathbb{R}^{d_{\mathbf{y}}} \times \mathbb{R}^{d_{\mathbf{y}} \times d_{\mathbf{x}}}$, and equally weight each component of the Gaussian mixture. Further details are given in Appendix E.3. We chose to control the dimension to gain insight into the performance of posterior sampling methods under varying dimensions. We also chose to control the noise level since the different posterior sampling methods have accuracy that depends on the signal-to-noise ratio. Through randomly varying the observation model, we gain an insight into the performance of the posterior sampling methods with different levels of posterior multimodality. This example is interesting because it allows us to study the behaviour of our methods on non-Gaussian problems in high dimensions whilst having access to the target posterior with which to compare (usually, obtaining a 'ground-truth' posterior is not feasible for non-Gaussian problems).

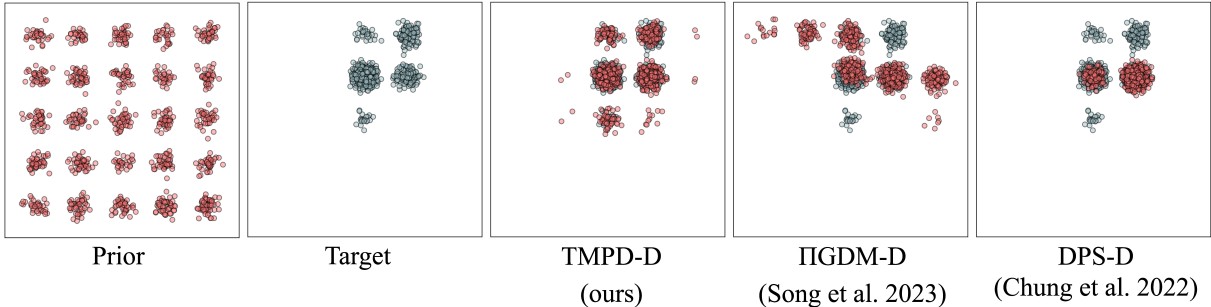

Figure 2: We display the first two dimensions of the GMM inverse problem for one of the measurement models tested $(\mathbf{H}, \sigma_y = 0.1, (d_x, d_y) = (80, 1))$. The blue dots represent samples from the target posterior, while the red dots correspond to samples generated by each of the algorithms used (the names of the algorithms are given at the bottom of each column).

Table 1: Sliced Wasserstein for the GMM case. The full table is in Ap. E.3.

| $d_x$ | 8 | 8 | 8 | 80 | 80 | 80 | 800 | 800 | 800 |
|---|---|---|---|---|---|---|---|---|---|
| $d_y$ | 1 | 2 | 4 | 1 | 2 | 4 | 1 | 2 | 4 |
| TMPD-D | 1.6 | **0.7** | **0.3** | 2.7 | **1.0** | **0.3** | 3.1 | **1.4** | **0.4** |
| DTMPD-D | 1.8 | 3.3 | **0.4** | 2.8 | 3.2 | 0.7 | 3.7 | 3.5 | 0.7 |
| DPS-D | 4.7 | 1.8 | 0.7 | 5.6 | 3.2 | 1.2 | 5.8 | 3.5 | 1.4 |
| ΠGDM-D | 2.6 | 2.1 | 3.8 | 3.2 | 2.8 | 0.6 | 3.5 | 3.1 | **0.4** |
| TMPD-D | **1.4** | **0.9** | **0.3** | **2.3** | **1.2** | **0.4** | **2.9** | **1.3** | **0.4** |
| DTMPD-D | 1.8 | 2.7 | 0.5 | 2.6 | 3.2 | 0.8 | 3.4 | 3.4 | 0.8 |
| DPS-D | 4.7 | 1.5 | 0.8 | 5.1 | 3.1 | 1.0 | 5.7 | 3.1 | 1.3 |
| ΠGDM-D | 2.2 | 1.6 | 3.8 | 2.9 | 2.7 | 0.6 | 3.3 | 2.7 | 0.4 |
| TMPD-D | **0.9** | **0.9** | 0.6 | 1.5 | 1.1 | 0.9 | 1.5 | **1.2** | 0.9 |
| DTMPD-D | **0.9** | 1.7 | 0.9 | 1.4 | 2.1 | 0.9 | 1.4 | 2.0 | 1.1 |
| DPS-D | 5.2 | 3.5 | 2.5 | 6.9 | 3.9 | 1.7 | 6.8 | 4.7 | 0.9 |
| ΠGDM-D | 1.5 | 2.3 | 1.8 | **1.6** | 1.4 | **0.9** | 2.0 | 2.0 | **0.6** |

We use the sliced Wasserstein (SW) distance defined in Appendix E.3 to compare the posterior distribution estimated by each algorithm with the target posterior distribution. We use $10^4$ slices for the SW distance and compare 1000 samples of TMPD-D, ΠGDM-D and DPS-D in Tables 1 obtained using 1000 denoising steps and 1000 samples of the true posterior distribution.

Table 1 indicates the Central Limit Theorem (CLT) 95% confidence intervals obtained by considering 20 randomly selected measurement models ($\mathbf{H}$) for each setting $(d_x, d_x, \sigma_y)$. Figure 2 shows the first two dimensions of the estimated posterior distributions corresponding to the configurations $(80, 1)$ from Table 1 for one of the randomly generated measurement model ($\mathbf{H}, \sigma_y = 0.1$). These illustrations give us insight into the behaviour of the algorithms and their ability to accurately estimate the posterior distribution. We observe that TMPD-D estimates the target posterior well compared to ΠGDM-D and DPS-D. TMPD-D covers all of the modes, whereas ΠGDM-D and DPS-D do not.

We perform the same experiment using 1000 samples of TMPD, DTMPD, DPS and ΠGDM, obtained using 1000 Euler-Maruyama time-steps, and results are shown in Appendix E.3.

A direct comparison to Cardoso et al. (2023) using their original experimental setup is shown in Table 5 in Appendix E.3, which shows competitive performance for posterior sampling compared to Sequential Monte-Carlo, an exact sampling method.

## 6.2 Noisy observation inpainting and super-resolution

We consider inpainting and super-resolution problems on the FFHQ $256 \times 256$ (Karras et al., 2019) and CIFAR-10 $32 \times 32$ (Krizhevsky et al., 2009) datasets. We compare TMPD to ΠDGM and DPS. We also compare score-based diffusion models with their denoising-diffusion counterparts (denoted with suffix, -D).

Firstly, we follow the benchmark used by Chung et al. (2022a) and use a Variance Preserving (VP) SDE, using a DDPM sampler, on FFHQ $256 \times 256$ using 1k validation images. The pre-trained diffusion model for FFHQ was taken from Chung et al. (2022a) and was used directly without any finetuning. We follow Chung et al. (2022a) and use various forward operators. For super-resolution, we use a downsampling ratio of 4 ($256 \times 256 \to 64 \times 64$) and bicubic interpolation; for 'box' mask inpainting we mask out $128 \times 128$ region and for 'random' mask inpainting we choose a random mask for each image masking between 30% and 70% of the pixels. Images are normalized to the range $[0, 1]$ and it is on this scale that we add Gaussian measurement noise with standard deviation $\sigma_y \in \{0.01, 0.05, 0.1, 0.2\}$. For quantitative comparison, we focus on two widely used perception distances, Fréchet Inception Distance (FID) and Learned Perceptual Image Patch Similarity (LPIPS) distance. FID evaluates consistency with the whole dataset using summary statistics from the FFHQ-50k dataset. We also evaluate observation data similarity using various distances between a sampled image and ground truth image: LPIPS, mean-squared-error (MSE), peak signal-to-noise-ratio (PSNR) and structural similarity index measure (SSIM). For ΠGDM-D we use the algorithm and default hyperparameters as described in Song et al. (2023). For DPS-D we use the algorithm in the codebase provided by the authors Chung et al. (2022a) and we use their default hyperparameters such as their suggested step-size hyperparameter for this task, and static-thresholding (clipping the denoised image at each step to a range $[-1, 1]$) whereas TMPD-D does not require hyperparameter tuning or static-thresholding. The results for FFHQ sampled using VP DDPM are shown in the appendix Table 7. We observe that DTMPD-D is competitive with DPD-D over a range of noise levels, however, ΠGDM-D is not able to produce high quality reconstructions for larger noise levels.

We note that the heuristics used in the DPS and ΠGDM implementations have been designed to work with the VP-SDE (DDPM sampler), and therefore the performance may not be robust to the choice of SDE. Fig 14 illustrates this when each method is applied to the VE-SDE on a sample from the FFHQ validation dataset. We next compare performance to TMPD across VP and VE-SDE samplers and a range of noise levels on CIFAR-10 $64 \times 64$ using 1k validation images.

We use pretrained denoising networks for CIFAR-10 that are available here. For inpainting, we use 'box' and 'half' mask. For 'half'-type inpainting, we mask out a $16 \times 16$ right half region of the image; for box-type inpainting, we mask out an $8 \times 8$ box region following Cardoso et al. (2023). For super-resolution, we use a downsampling ratio of 2 on each axis ($32 \times 32 \to 16 \times 16$) with a nearest-neighbour downsampling method; and a downsampling ratio of 4 ($32 \times 32 \to 8 \times 8$) with bicubic downsampling. Images are normalized to the range $[0, 1]$ and it is on this scale that we add Gaussian measurement noise with standard deviation $\sigma_y \in \{0.01, 0.05, 0.1\}$. Whereas no hyperparameters are required for our method, we chose the DPS scale hyperparameter by optimising LPIPS, MSE, PSNR and SSIM on a validation set of 128 images (see Fig. 9 for an example). We found that static thresholding (clipping the denoised image estimate to a range $[0, 1]$ at each sampling step) is critical for the stability and performance of both DPS-D and ΠGDM-D. Stability was noted as a limitation in Chung et al. (2022a), and they suggest that devising methods to stabilize the samplers would be a promising direction of research. We find that our TMPD-D method and the diagonal approximation DTMPD-D is stable across SDE, noise-level and observation maps, without the need for static thresholding. Whilst for VE ΠGDM-D we found the original algorithm in Song et al. (2023) to be stable, for VP ΠGDM-D, the original algorithm, whilst stable for FFHQ, was not stable, even with static-thresholding, for CIFAR-10. We chose to bring ΠGDM-D a step closer to our algorithm by substituting their likelihood score into an Ancestral sampling algorithm, instead of a DDIM algorithm as suggested in Song et al. (2023), which produced stable samples.

The methods TMPD, ΠGDM and DPS all have the same numerical solver of their respective reverse-SDE, and DTMPD-D, ΠGDM-D and DPS-D all use DDPM since DDIM and DDPM are equivalent algorithms with our chosen DDIM hyperparameter $\eta = 1.0$. Therefore, the sampling methods being compared only differ in the $\nabla_{\mathbf{x}_t} \log p_{\mathbf{y}|t}(\mathbf{y}|\mathbf{x}_t)$ term in their reverse-SDE, and so this experiment allows us to study the

Table 2: Summary of results using the VE and VP-SDE for increasingly noisy $\sigma_y \in \{0.01, 0.05, 0.1\}$ observation inpainting and super-resolution problems on FFHQ 1k validation set. The full results are in Ap. E.4.

| SDE | | VE-SDE | | VP-SDE | |
|---|---|---|---|---|---|
| PROBLEM | METHOD | FID ↓ | LPIPS ↓ | FID ↓ | LPIPS ↓ |
| $\sigma_y = 0.01$ | DTMPD-D | **32.3** | **0.203** $_{\pm 0.039}$ | **29.6** | 0.230 $_{\pm 0.034}$ |
| 4× 'BICUBIC' | DPS-D | 47.0 | 0.273 $_{\pm 0.031}$ | 31.4 | 0.234 $_{\pm 0.048}$ |
| SUPER-RESOLUTION | ΠGDM-D | 37.4 | 0.244 $_{\pm 0.030}$ | 29.7 | **0.198** $_{\pm 0.037}$ |
| $\sigma_y = 0.05$ | DTMPD-D | **32.1** | **0.268** $_{\pm 0.048}$ | 32.7 | 0.304 $_{\pm 0.043}$ |
| 4× 'BICUBIC' | DPS-D | 105.9 | 0.590 $_{\pm 0.036}$ | **29.3** | 0.280 $_{\pm 0.051}$ |
| SUPER-RESOLUTION | ΠGDM-D | 106.8 | 0.592 $_{\pm 0.041}$ | 45.1 | 0.311 $_{\pm 0.047}$ |
| $\sigma_y = 0.1$ | DTMPD-D | **32.7** | **0.310** $_{\pm 0.053}$ | 38.0 | 0.348 $_{\pm 0.048}$ |
| 4× 'BICUBIC' | DPS-D | 114.0 | 0.569 $_{\pm 0.044}$ | **30.9** | 0.318 $_{\pm 0.051}$ |
| SUPER-RESOLUTION | ΠGDM-D | 206.0 | 0.724 $_{\pm 0.034}$ | 119.6 | 0.589 $_{\pm 0.047}$ |
| $\sigma_y = 0.01$ | DTMPD-D | 30.2 | 0.114 $_{\pm 0.029}$ | **25.7** | 0.153 $_{\pm 0.033}$ |
| 'BOX' MASK | DPS-D | **23.9** | **0.093** $_{\pm 0.019}$ | 31.5 | 0.175 $_{\pm 0.038}$ |
| INPAINTING | ΠGDM-D | 27.1 | 0.108 $_{\pm 0.025}$ | 143.8 | 0.247 $_{\pm 0.024}$ |
| $\sigma_y = 0.05$ | DTMPD-D | **33.6** | **0.186** $_{\pm 0.036}$ | **27.0** | 0.240 $_{\pm 0.038}$ |
| 'BOX' MASK | DPS-D | 39.7 | 0.318 $_{\pm 0.044}$ | 30.7 | 0.228 $_{\pm 0.046}$ |
| INPAINTING | ΠGDM-D | 49.5 | 0.354 $_{\pm 0.044}$ | 159.3 | 0.448 $_{\pm 0.046}$ |
| $\sigma_y = 0.1$ | DTMPD-D | **34.0** | **0.223** $_{\pm 0.041}$ | 29.6 | 0.292 $_{\pm 0.049}$ |
| 'BOX' MASK | DPS-D | 59.1 | 0.467 $_{\pm 0.053}$ | **29.3** | 0.259 $_{\pm 0.049}$ |
| INPAINTING | ΠGDM-D | 72.6 | 0.529 $_{\pm 0.047}$ | 165.7 | 0.539 $_{\pm 0.083}$ |

behaviour of our method on inpainting and super-resolution compared to the different approximations of the smoothed likelihood. A summary of the results for CIFAR-10 sampled using VE DDPM are shown in Table 2. The complete results are in Tables 8 and 12 for VP and VE DDPM respectively, and Tables 9 and 13 for score-based VP and VE-SDE, respectively. For more experimental details including illustration of samples used to generate the tables can be found in Appendix E.4. Our method is the only method able to provide high-quality reconstructions independently of the SDE, time discretization or noise level used. On the other hand, we see that DPS-D and ΠGDM-D are not able to provide high-quality reconstructions for the VE-SDE. For the continuous time methods, ΠGDM and DPS are outperformed by TMPD for both VE and VP-SDEs in the majority of tasks.

# 7 Discussions, limitations and future work

In this paper, we introduced TMPD, a diffusion modelling approach to solve inverse problems and sample from conditional distributions using unconditional diffusion models. On various tasks on the VP-SDE, TMPD achieves competitive quality with other methods that aim to solve the noisy, linear inverse problem while avoiding the expensive, problem-specific training of conditional models. Our method is more versatile since it can also be used for the VE-SDE, and for large noise and different time discretizations.

TMPD is slower, as each iteration costs more memory and compute due to the Jacobian over the score model. Even with a diagonal and row-sum approximation to the Jacobian, the method is around $1.5\times$ slower than DPS and ΠGDM. The row-sum approximation is not suitable for inverse problems with more complicated, non-diagonal and nonlinear observation operators, therefore, it would be helpful to explore methods to circumvent heuristics. For example, heuristics can be circumvented by noting in the definition of Equation 11 does not require the inverse but rather solving a system of linear equations with right hand side $\mathbf{y} - \mathbf{Hm}_{0|t}(\mathbf{x}_t)$. A very recent work (Rozet et al., 2024), uses the natural choice of the conjugate gradient (CG) method to solve this linear system with success on non-diagonal and nonlinear observation operators, even for a small number of iterations of the CG method.

Using flow based methods (Albergo & Vanden-Eijnden, 2022; Lipman et al., 2022) as unconditional priors and building conditional sampling methods that leverage flows as pretrained models (Ben-Hamu et al., 2024; Pandey et al., 2024) is related to the approach used in this paper. For example, it is possible to directly substitute in the TMPD approximation to the PiGDM method applied to flows (Pokle et al., 2023), which would be a fruitful direction for future work.

Many state of the art diffusion models operate in a latent space which would make any observation operator nonlinear, and thus linear observations maps do not suffice in the latent diffusion setting. Furthermore, for inverse problems in general it may be excessive to train a diffusion model for the full data distribution, and it could be difficult to outperform a diffusion model trained for a specific inverse problem. For example, using diffusion models trained specifically for the super-resolution problem in a cascade together with noise conditioning augmentation (Ho et al., 2022) has been extremely effective in progressively generating high-fidelity images, see e.g., Saharia et al. (2022).

A limitation of our method is that without approximations, the full method may be costly to implement in high dimensions of image data. On the positive side, our approach does not require any hyperparameter tuning, does not require static-thresholding of the denoised image, and is more principled when compared to existing approaches, as shown in Section 4. We show that our method, unlike ΠGDM, does not fail for the cases where the additive Gaussian noise is significant. We provided a way to analyse similar methods and our moment approximations can be analysed more rigorously to provide deeper theoretical results for these kinds of methods. Our future work plans include expanding the analysis we provided in this work.

## Acknowledgements

This work has been supported by The Alan Turing Institute through the Theory and Methods Challenge Fortnights event *Accelerating generative models and nonconvex optimisation*, which took place on 6-10 June 2022 and 5-9 Sep 2022 at The Alan Turing Institute headquarters. JP and SR acknowledge funding by Deutsche Forschungsgemeinschaft (DFG) – Project-ID 318763901 - SFB1294. M. Girolami was supported by a Royal Academy of Engineering Research Chair grant RCSRF1718/6/34, and EPSRC grants EP/W005816/1, EP/V056441/1, EP/V056522/1, EP/R018413/2, EP/R034710/1, EP/Y028805/1, and EP/R004889/1. The authors would also like to thank the Isaac Newton Institute for Mathematical Sciences, Cambridge, for support and hospitality during the programme *The Mathematical and Statistical Foundation of Future Data-Driven Engineering* where work on this paper was undertaken. This work was supported by EPSRC grant no EP/R014604/1. BB gratefully acknowledges the EPSRC for funding this research through the EPSRC Centre for Doctoral Training in Future Infrastructure and Built Environment: Resilience in a Changing World (EPSRC grant reference number EP/S02302X/1); and the support of nPlan, and in particular Damian Borowiec and Peter A. Zachares, for the invaluable facilitation of work that was completed whilst on internship with nPlan and access to A100 GPUs.

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

# A Proofs for Section 3

## A.1 Proof of Proposition 1

In order to prove this result, we adapt Theorem 1 of Meng et al. (2021). We write the proof for generic exponential family which can be adapted to our case easily. Let us consider

$$p_{t|0}(\mathbf{x}_t|\mathbf{x}_0) = \mathcal{N}(\mathbf{x}_t; \sqrt{\alpha_t}\mathbf{x}_0, v_t\mathbf{I}_{d_x}).$$

and write $p_t(\mathbf{x}_t) = \int p_{t|0}(\mathbf{x}_t|\mathbf{x}_0)p_0(\mathbf{x}_0)\mathrm{d}\mathbf{x}_0$. We are interested in finding the moments of the posterior

$$p_{0|t}(\mathbf{x}_0|\mathbf{x}_t) = \frac{p_{t|0}(\mathbf{x}_t|\mathbf{x}_0)p_0(\mathbf{x}_0)}{p_t(\mathbf{x}_t)}.$$

Let us redefine the right handside in this Bayes' rule using the exponential family parameterisation of the Gaussian $p_{t|0}(\mathbf{x}_t|\mathbf{x}_0)$

$$p(\boldsymbol{\eta}_0|\mathbf{x}_t) = \frac{p_{t|0}(\mathbf{x}_t|\boldsymbol{\eta}_0)q_0(\boldsymbol{\eta}_0)}{p_t(\mathbf{x}_t)},$$

where $\boldsymbol{\eta}_0 = \mathbf{x}_0\sqrt{\alpha_t}/v_t$ and

$$p_{t|0}(\mathbf{x}_t|\boldsymbol{\eta}_0) = e^{\boldsymbol{\eta}_0^\top \mathbf{x}_t - \psi(\boldsymbol{\eta}_0)}q(\mathbf{x}_t),$$

where

$$q(\mathbf{x}_t) = ((2\pi)^d v^{2d})^{-1/2} e^{-\mathbf{x}_t^\top \mathbf{x}_t / 2v_t}$$

Now let

$$\lambda(\mathbf{x}_t) = \log \frac{p_t(\mathbf{x}_t)}{q(\mathbf{x}_t)},$$

we can rewrite

$$p(\boldsymbol{\eta}_0 | \mathbf{x}_t) = e^{\boldsymbol{\eta}_0^\top \mathbf{x}_t - \psi(\mathbf{x}_t) - \lambda(\mathbf{x}_t)} q_0(\boldsymbol{\eta}_0).$$

It is then easy to show that the moments of $\boldsymbol{\eta}_0$ are given by (Meng et al., 2021)

$$\mathbb{E}[\boldsymbol{\eta}_0 | \mathbf{x}_t] = \mathbf{J}_\lambda(\mathbf{x}_t),$$
$$\mathbb{E}[\boldsymbol{\eta}_0^\top | \mathbf{x}_t] = \mathbf{J}_\lambda(\mathbf{x}_t)^\top,$$
$$\mathbb{E}[\boldsymbol{\eta}_0 \boldsymbol{\eta}_0^\top | \mathbf{x}_t] = \mathbf{S}_\lambda(\mathbf{x}_t) + \mathbf{J}_\lambda(\mathbf{x}_t) \mathbf{J}_\lambda(\mathbf{x}_t)^\top,$$

where $\mathbf{J}_\lambda$ is the Jacobian of $\lambda$ w.r.t. $\mathbf{x}_t$ and $\mathbf{S}_\lambda$ is the Hessian of $\lambda$ w.r.t. $\mathbf{x}_t$. Recall now that $\boldsymbol{\eta}_0 = \mathbf{x}_0 \sqrt{\alpha_t}/v_t$ and

$$\lambda(\mathbf{x}_t) = \log p_t(\mathbf{x}_t) + \frac{\mathbf{x}_t^\top \mathbf{x}_t}{2v_t} + C$$
$$\mathbf{J}_\lambda(\mathbf{x}_t) = \nabla_{\mathbf{x}_t} \log p_t(\mathbf{x}_t) + \frac{\mathbf{x}_t}{v_t}$$
$$\mathbf{S}_\lambda(\mathbf{x}_t) = \nabla_{\mathbf{x}_t}^2 \log p_t(\mathbf{x}_t) + \frac{1}{v_t} \mathbf{I}_{d_x}.$$

This implies that

$$\mathbb{E}[\mathbf{x}_0 | \mathbf{x}_t] = \frac{1}{\sqrt{\alpha_t}} (\mathbf{x}_t + v_t \nabla_{\mathbf{x}_t} \log p_t(\mathbf{x}_t)),$$

which proves the Tweedie's formula for the mean. For the covariance, note that

$$\mathrm{Cov}(\boldsymbol{\eta_0} | \mathbf{x}_t) = \mathbb{E}[\boldsymbol{\eta}_0 \boldsymbol{\eta}_0^\top | \mathbf{x}_t] - \mathbb{E}[\boldsymbol{\eta}_0 | \mathbf{x}_t] \mathbb{E}[\boldsymbol{\eta}_0 | \mathbf{x}_t]^\top$$
$$= \nabla^2 \log p_t(\mathbf{x}_t) + \frac{1}{v_t} \mathbf{I}_{d_x}.$$

Since

$$\mathrm{Cov}(\boldsymbol{\eta_0} | \mathbf{x}_t) = \frac{\alpha_t}{v_t^2} \mathrm{Cov}(\mathbf{x}_0 | \mathbf{x}_t),$$

we conclude

$$\mathrm{Cov}(\mathbf{x}_0 | \mathbf{x}_t) = \frac{v_t}{\alpha_t} (\mathbf{I}_{d_x} + v_t \nabla^2 \log p_t(\mathbf{x}_t)),$$

which concludes the proof. $\square$

## B Proofs for Section 4

### B.1 Proof of Proposition 3

If $p_{\mathrm{data}}$ is Gaussian, then the full process $\mathbf{x}_t$ is a Gaussian process. In particular, $(\mathbf{x}_0, \mathbf{x}_t)$ are jointly Gaussian:

$$\mathbf{x}_0 | \mathbf{x}_t \propto \mathcal{N}(\mathbb{E}[\mathbf{x}_0 | \mathbf{x}_t], \mathbb{V}[\mathbf{x}_0 | \mathbf{x}_t]),$$

where $\mathbb{V}$ denotes the covariance. However, the right-hand side is precisely the approximation we make, due to Proposition 1. Therefore, the approximation we make in Equation 11 is correct. Adding this to the learned drift $\mathbf{s}_\theta(\mathbf{x}_t, t) = \nabla \log p_t(\mathbf{x}_t)$, we get an expression for $\nabla \log p_t(\mathbf{x}_t | \mathbf{y})$ by Equation 3.

## B.2 Proof of Theorem 1

We first introduce three SDEs.

**Conditioned SDEs** : The first pair of SDEs is given by an OU process, but started in the conditional distribution:

$$d\mathbf{x}_t^c = -\frac{1}{2}\mathbf{x}_t^c dt + d\mathbf{w}_t, \quad \mathbf{x}_0^c \sim p_0^c := p_{\text{data}}(\mathbf{x}_0|\mathbf{y}). \tag{18}$$

We denote the marginals of $\mathbf{x}_t^c$ by $p_t^c$ to differentiate them from the marginals $p_t$ of the OU process started in the correct distribution, defined in Equation 2. Note that $p_t^c = p_t(\mathbf{x}_t|\mathbf{y})$. The time reversal $\mathbf{z}_t = \mathbf{x}_{T-t}$ then satisfies the SDE

$$\begin{aligned}
d\mathbf{z}_t =& \frac{1}{2}\mathbf{z}_t dt + \nabla_{\mathbf{z}_t} \log p_{T-t}^c(\mathbf{z}_t) dt + d\mathbf{w}_t \\
=& \frac{1}{2}\mathbf{z}_t dt + \nabla_{\mathbf{z}_t} \log p_{T-t}(\mathbf{z}_t) dt + \nabla_{\mathbf{z}_t} \log p_{\mathbf{y}|T-t}(\mathbf{y}|\mathbf{z}_t) dt + d\mathbf{w}_t \\
\mathbf{z}_0 \sim& p_T^c,
\end{aligned} \tag{19}$$

where we used Equation 3.

Solutions to the above reverse SDE will sample our target measure $p_0^c = p_{\text{data}}(\mathbf{x}_0|\mathbf{y})$ at final time. Therefore, we want to study how our algorithm approximates solutions of $\mathbf{x}^c$.

**Intermediate Gaussian SDE:** Instead of bounding the distance of solutions to the above conditioned reverse SDE to our algorithm, we will instead introduce an intermediate process, which we will later use in a triangle inequality. The process will be a Gaussian process. Therefore, we denote it with a superscript $G$.

The process $\mathbf{x}_t^G$ will be defined analogous to Equation 2, but assuming that it is started in $\mathcal{N}(\mathbf{x}_0; \boldsymbol{\mu}_0, \boldsymbol{\Sigma}_0)$ instead of $p_{\text{data}}$. Since the forward SDE in Equation 2 is linear, all of the marginals of $\mathbf{x}_t^G$, which we denote by $p_t^G$, will also be Gaussian.

Again, we define a conditioned version of $\mathbf{x}$, called $\mathbf{x}^{G,c}$ analogous to Equation 18, i.e. $\mathbf{x}_0^{G,c}$ will be distributed as $\mathbf{x}_0^G$ conditioned on $\mathbf{y} = y$, i.e. $p^G(\mathbf{x}_0|\mathbf{y} = y)$. Since we have a linear observation model, $p^G(\mathbf{x}_0|\mathbf{y} = y)$ is still a Gaussian, and therefore $\mathbf{x}^{G,c}$ will also be a Gaussian process. Its reverse SDE $\mathbf{z}^{G,c}$ is defined through Equation 19, just that every appearance of $p$ will also have a superscript $G$,

$$d\mathbf{z}_t = \frac{1}{2}\mathbf{z}_t dt + \nabla_{\mathbf{z}_t} \log p_{T-t}^G(\mathbf{z}_t) dt + \nabla_{\mathbf{z}_t} \log p_{T-t}^G(\mathbf{y}|\mathbf{z}_t) dt + d\mathbf{w}_t \quad \mathbf{z}_0 \sim p_T^c, \tag{20}$$

**Algorithm SDE:** Finally, we define a third reverse SDE, which is the SDE that we are discretizing when implementing our algorithm. It is given by

$$d\mathbf{z}_t = \frac{1}{2}\mathbf{z}_t dt + \nabla_{\mathbf{z}_t} \log p_{T-t}(\mathbf{z}_t) dt + f_{T-t}(\mathbf{z}_t) dt + d\mathbf{w}_t \quad \mathbf{z}_0 \sim p_T^c, \tag{21}$$

where

$$f_t(\mathbf{z}_t) = \nabla_{\mathbf{z}_t} \mathbb{E}[\mathbf{x}_0|\mathbf{x}_t = z_t] \mathbf{H}^\top (\mathbf{H}\mathbb{V}[\mathbf{x}_0|\mathbf{x}_t = z_t]\mathbf{H}^\top + \sigma_y^2 \mathbf{I})^{-1}(\mathbf{y} - \mathbf{H}\mathbb{E}[\mathbf{x}_0|\mathbf{x}_t = z_t]) \tag{22}$$

and $\mathbb{V}$ denotes the conditional covariance. Except for approximation errors due to time discretization and the initial conditions, this is the SDE we are sampling from in our algorithm. We denote the marginal of $\mathbf{z}_t$ by $q_t$. Now we are ready to write our proof.

*Proof of Theorem 1.* We have that $p_{\text{data}}(\cdot|\mathbf{y}) = p_0^c$

$$\|p_0^c - q_T\|_{\text{TV}} \leq \|p_0^c - p_0^{G,c}\|_{\text{TV}} + \|p_0^{G,c} - q_T\|_{\text{TV}}$$

We will bound the second term using Pinsker's inequality to get a KL term on the path space, as done in Chen et al. (2023). The proof then consists of bounding the first total variation term and the resulting KL term.

**Bounding the first term:** The first term is the total variation distance between $p_0(\mathbf{x}_0|\mathbf{y} = y)$ and $p_0^G(\mathbf{x}_0^G|\mathbf{y} = y)$. Now,

$$
\begin{aligned}
p_0(\mathbf{x}_0 = x_0|\mathbf{y} = y) &= \frac{p_{\mathbf{y}|0}(\mathbf{y} = y|\mathbf{x}_0 = x_0)p_0(\mathbf{x}_0 = x_0)}{p_{\mathbf{y}}(\mathbf{y} = y)} \\
&= \frac{p_{\mathbf{y}|0}^G(\mathbf{y} = y|\mathbf{x}_0 = x_0)p_0^G(\mathbf{x}_0 = x_0)}{p_{\mathbf{y}}^G(\mathbf{y} = y)} \frac{p_0(\mathbf{x}_0 = x_0)}{p_0^G(\mathbf{x}_0^G = x_0^G)} \frac{p_{\mathbf{y}}^G(\mathbf{y} = y)}{p_{\mathbf{y}}(\mathbf{y} = y)} \\
&= p_{\mathbf{y}|0}^G(\mathbf{x}_0 = x_0|\mathbf{y} = y) \exp(\Phi(x_0)) \frac{p_{\mathbf{y}}^G(\mathbf{y} = y)}{p_{\mathbf{y}}(\mathbf{y} = y)},
\end{aligned}
$$

where we used that the conditional distribution of $\mathbf{y}$ given $\mathbf{x}_0$ does not depend on the distribution of $\mathbf{x}_0$. For the last term we get

$$
\begin{aligned}
p_{\mathbf{y}}(\mathbf{y} = y) &= \int p_{\mathbf{y}|0}(\mathbf{y} = y|\mathbf{x}_0 = x_0)p_0(\mathbf{x}_0 = x_0)\mathrm{d}x_0 \\
&= \int p_{\mathbf{y}|0}^G(\mathbf{y} = y|\mathbf{x}_0 = x_0)p_0^G(\mathbf{x}_0 = x_0) \exp(\Phi(x_0))\mathrm{d}x_0 \\
&= N p_{\mathbf{y}}^G(\mathbf{y} = y),
\end{aligned}
$$

with $N \in [1/M, M]$. Therefore,

$$
\|p_0^c - p_0^{G,c}\|_{\mathrm{TV}} \leq \int |\frac{p_0^c}{p_0^{G,c}}(x_0) - 1|p_0^{G,c}(x_0)\mathrm{d}x_0 \leq M^2 - 1,
$$

where we used that $M - 1$ is always greater or equal to $1 - \frac{1}{M}$, since $M \geq 1$.

**Bounding the second term:** For the second term we get that $p_0^{G,c}$ is the final marginal of Equation 20, while $q_T$ is the final marginal of Equation 21. We now use Pinsker's inequality,

$$
\|p_0^{G,c} - q_T\|_{\mathrm{TV}} \leq \sqrt{\mathrm{KL}(p_0^{G,c}\|q_T)}.
$$

We denote the path measures induced by Equation 20 and Equation 21 by $\mathbb{P}^{G,c}$ and $\mathbb{Q}$ respectively. They have final time marginals $p_0^{G,c}$ and $q_T$. Therefore, we can bound the KL-Divergence of the marginals $p^{G,c}$ and $q_T$ by the KL-Divergence on the full path space:

$$
\mathrm{KL}(p_0^{G,c}\|q_T) \leq \mathrm{KL}(\mathbb{P}^{G,c}\|\mathbb{Q}).
$$

We will assume that we can apply Girsanovs theorem and show later that this is justified. Using Girsanovs theorem, we can evaluate the Radon-Nikodym derivative $\frac{\mathrm{d}\mathbb{Q}}{\mathrm{d}\mathbb{P}^{G,c}}$ on the path space and therefore calculate the KL-divergence:

$$
\begin{aligned}
\mathrm{KL}(\mathbb{P}^{G,c}\|\mathbb{Q}) =&\mathbb{E}_{\mathbb{P}^{G,c}}[\log \frac{\mathrm{d}\mathbb{P}^{G,c}}{\mathrm{d}\mathbb{Q}}] \\
=&\mathbb{E}_{\mathbb{P}^{G,c}}\left[-\int_0^t \nabla \log p_t(x_t) + f_{T-t}(x_t) - \nabla \log p_{T-t}^{G,c}(x_t)\mathrm{d}w_t\right] \\
&+\mathbb{E}_{\mathbb{P}^{G,c}}\left[\int_0^t \|\nabla \log p_t(x_t) + f_{T-t}(x_t) - \nabla \log p_{T-t}^{G,c}(x_t)\|^2\mathrm{d}t\right] \\
=&\int_0^t \mathbb{E}_{p_t^{G,c}}[\|\nabla \log p_t(x_t) + f_{T-t}(x_t) - \nabla \log p_{T-t}^{G,c}(x_t)\|^2]\mathrm{d}t,
\end{aligned}
$$

where the term on the second line drops out because stochastic integrals have expectation 0. Now we have the drift of the Gaussian SDE given by

$$
\begin{aligned}
&\nabla \log p_t^{G,c}(x_t) \\
=&\nabla \log p_t^G(x_t) + \nabla \log p^G(\mathbf{y}|\mathbf{x}_t^G = x_t) \\
=&\nabla \log p_t^G(x_t) \\
&+\nabla_{x_t}\mathbb{E}[\mathbf{x}_0^G|\mathbf{x}_t^G = x_t]\mathbf{H}^\top(\mathbf{H}\mathbb{V}[\mathbf{x}_0^G|\mathbf{x}_t^G = x_t]\mathbf{H}^\top + \sigma_y^2\mathbf{I}_{d_y})^{-1}(\mathbf{y} - \mathbf{H}\mathbb{E}[\mathbf{x}_0^G|\mathbf{x}_t^G = x_t]) \\
=&:\nabla \log p_t^G(x_t) + \tilde{f}_t(x_t),
\end{aligned}
\tag{23}
$$

see Equation 48, while the drift of the algorithm SDE is given by

$$
\nabla \log p_t(x_t) + f_t(x_t),
\tag{24}
$$

where $f_t$ is given by Equation 22. We see that the difference in the SDE drifts mainly consists of differences between conditional moments of $p_t$ and $p_t^G$, as well as the derivatives of the conditional expectations. Therefore, the main difficulty of the proof is to bound these.

The density of the conditional distribution of $p_{0|t}$ is given by

$$
\begin{aligned}
p_{0|t}(x_0|x_t) =&\frac{p_{0,t}(x_0, x_t)}{p_t(x_t)} = \frac{p_{0,t}^G(x_0, x_t)\exp(\Phi(x_0))}{p(x_t)} = \frac{p_{0|t}^G(x_0|x_t)p^G(x_t)\exp(\Phi(x_0))}{p(x_t)} \\
=&p_{0|t}^G \exp(\Phi(x_0) - \Phi(x_t)),
\end{aligned}
\tag{25}
$$

where

$$
\exp(\Phi_t(x_t)) = \frac{p_t(x_t)}{p_t^G(x_t)} = \mathbb{E}[\exp(\Phi(\mathbf{x}_0))|\mathbf{x}_t = x_t].
\tag{26}
$$

In the last equality, we used that

$$
\frac{d\mathbb{P}}{d\mathbb{P}^G}(x_{[0,t]}) = \exp(\Phi(x_0)),
$$

where we denoted by $\mathbb{P}$ the path measure induced by Equation 18. Therefore, also their marginals $p_t = \mathbb{P}_t^G$ and $p_t = \mathbb{P}_t$ have relative densities, which are given by integrating out the density to time $t$, as we did in Equation 26.

By assumption $\exp(\Phi)$ is bounded from above and below by $M$ and $1/M$ respectively, and by Equation 26 the same holds for $\exp(\Phi_t)$. Therefore, by Equation 25, $p_{t|0}$ is absolutely continuous with respect to $p_{t|0}^G$ with a density that is bounded from above and below by $M^2$ and $1/M^2$ respectively. We now obtain

$$
\begin{aligned}
&|\mathbb{E}[\mathbf{x}_0|\mathbf{x}_t = x_t] - \mathbb{E}[\mathbf{x}_0^G|\mathbf{x}_t^G = x_t]| \\
=&\left|\int x_0[p(x_0|x_t) - p^G(x_0|x_t)]dx_0\right| = \left|\int x_0 p^G(x_0|x_t)(\exp(\Phi(x_0) - \Phi(x_t)) - 1)dx_0\right| \\
\leq&(M^2 - 1)\mathbb{E}[|\mathbf{x}_0^G||\mathbf{x}_t^G = x_t]
\end{aligned}
$$

The same holds for every entry of the covariance matrix. We denote

$$
\begin{aligned}
V_{ij} :=&\mathbb{V}[\mathbf{x}_0|\mathbf{x}_t]_{ij}, \\
V_{ij}^G :=&\mathbb{V}[\mathbf{x}_0^G|\mathbf{x}_t^G]_{ij} = N_{ij}, \\
|V_{ij} - V_{ij}^G| \leq&(M^2 - 1)N_{ij} \\
N_{ij} :=&\mathbb{E}[|\mathbf{x}_{0,i}^G\mathbf{x}_{0,j}^G| \,|\mathbf{x}_t^G]
\end{aligned}
$$

Now we get that

$$
\|\mathbf{V} - \mathbf{V}^G\|_F = \left(\sum_{ij}(V_{ij} - V_{ij}^G)^2\right)^{1/2} \leq (M^2 - 1)\|\mathbf{V}^G\|_F \max_{ij}(N_{ij}).
$$

We can bound $N_{ij}$ by

$$N_{ij} \leq \mathbb{E}[(\mathbf{x}_{0,i}^G)^2 + (\mathbf{x}_{0,j}^G)^2 | \mathbf{x}_t^G] \leq \text{Tr}(\mathbb{V}[\mathbf{x}_0^G | \mathbf{x}_t^G = x_t]). \tag{27}$$

The latter term does not actually depend on $x_t$, but only on $t$. Using the formulas in Equation 44 we see that it can be bounded independently of $t$ (only depending on $m_0$ and $\Sigma_0$). Therefore, we can put it into a constant. We get that

$$\|\mathbf{V} - \mathbf{V}^G\|_F \lesssim (M^2 - 1)\|\mathbf{V}^G\|_F,$$

where

$$a \lesssim b \quad \Leftrightarrow \quad a \leq Cb,$$

with a constant $C$ only depending on $m_0$, $\Sigma_0$, $H$, $\sigma_y$ and the observation $y$. Now let $v$ be a vector with $\|v\| = 1$. It holds that

$$
\begin{aligned}
v^\top V_{ij} v &= \int (v^\top (x_0 - \mathbb{E}[\mathbf{x}_0 | \mathbf{x}_t = x_t]))^2 p_{0|t}^G(x_0 | x_t) \exp(\Phi(x_0) - \Phi_t(x_t)) \mathrm{d}x_0 \\
&= \frac{1}{M} v^\top V_{ij}^G v + N \int (v^\top (\mathbb{E}[\mathbf{x}_0^G | \mathbf{x}_t^G = x_t] - \mathbb{E}[\mathbf{x}_0 | \mathbf{x}_t = x_t]))^2 p_{0|t}(x_0 | x_t) \mathrm{d}x_0 \\
&\geq v^\top V_{ij}^G v \frac{1}{M}
\end{aligned}
\tag{28}
$$

with $N \geq 1/M$. Since $V$ and $V^G$ are positive semidefinite, symmetric matrices, this implies that all eigenvalues of $V$ are bounded by the lowest eigenvalue of $V^G$ times $1/M$. We define

$$\mathbf{T}_1 = \mathbf{H}\mathbf{V}^G\mathbf{H}^\top + \sigma_y^2 \mathbf{I}_{d_y} \tag{29}$$

$$\mathbf{T}_2 = \mathbf{H}\mathbf{V}\mathbf{H}^\top + \sigma_y^2 \mathbf{I}_{d_y}. \tag{30}$$

We then need to bound

$$\|\mathbf{T}_1^{-1} - \mathbf{T}_2^{-1}\| \leq \|\mathbf{T}_1 - \mathbf{T}_2\|(\|\mathbf{T}_1^{-1}\| + \|\mathbf{T}_2^{-1}\|).$$

We start with

$$\|\mathbf{T}_1 - \mathbf{T}_2\|_{\text{op}} \leq \|\mathbf{H}\|^2 \|\mathbf{V} - \mathbf{V}^G\|_{\text{op}} \lesssim \|\mathbf{V} - \mathbf{V}^G\|_F \lesssim (M^2 - 1)\|\mathbf{V}^G\|_F.$$

We also used the equivalence of the Frobenius and operator norm here. Due to our eigenvalue bound (Equation 28) on $\mathbf{V}$, we also get an analogous bound on $\mathbf{V} + \sigma_y \mathbf{I}_{d_y}$. Therefore,

$$\|\mathbf{T}_1^{-1} - \mathbf{T}_2^{-1}\|_{\text{op}} \lesssim (M^2 - 1)\|\mathbf{V}^G\|_F \|\mathbf{V}^G\|_F^{-1}(1 + M) \leq (M^2 - 1)(M + 1) \leq M^3 - 1$$

where we used that the operator norm of the inverse, is the inverse of the operator norm and the equivalence of the operator norm to the Frobenius norm.

Finally, we need to bound

$$
\begin{aligned}
&\nabla_{x_t} \int x_0 p(x_0 | x_t) \mathrm{d}x_0 \\
&= \int x_0 \nabla_{x_t} p^G(x_0 | x_t) \exp(\Phi(x_0) - \Phi(x_t)) \mathrm{d}x_0 \\
&= \int x_0 \exp(\Phi(x_0) - \Phi(x_t)) \nabla_{x_t} p^G(x_0 | x_t) \mathrm{d}x_0 + \int x_0 \nabla_{x_t} \Phi(x_t) \exp(\Phi(x_0) - \Phi(x_t)) p^G(x_0 | x_t) \mathrm{d}x_0 \\
&= \int x_0 \exp(\Phi(x_0) - \Phi(x_t)) \nabla_{x_t} p^G(x_0 | x_t) \mathrm{d}x_0 + \nabla_{x_t} \Phi(x_t) \mathbb{E}[\mathbf{x}_0 | \mathbf{x}_t = x_t].
\end{aligned}
\tag{31}
$$

Also

$$
\left\| \int x_0 \exp(\Phi(x_0) - \Phi(x_t)) \nabla_{x_t} p^G(x_0|x_t) \mathrm{d}x_0 - \nabla_{x_t} \mathbb{E}[\mathbf{x}_0^G | \mathbf{x}_t^G = x_t] \right\|
$$
$$
= \left\| \int \nabla_{x_t} p^G(x_0|x_t) x_0^\top (\exp(\Phi(x_0) - \Phi(x_t)) - 1) \mathrm{d}x_0 \right\|
$$
$$
\leq (M^2 - 1) \int p^G(x_0|x_t) \|\nabla \log p^G(x_0|x_t) x_0^\top\| \mathrm{d}x_0
$$
$$
\leq (M^2 - 1) (\int p^G(x_0|x_t) \|\nabla \log p^G(x_0|x_t) x_0^\top\| \mathrm{d}x_0)^{1/2} \tag{32}
$$
$$
\leq (M^2 - 1) (\int p^G(x_0|x_t) x_0^\top \Sigma_{0|t}^{-1}(x_0 - m_{0|t}^{x_t}) \mathrm{d}x_0)^{1/2}
$$
$$
\lesssim (M^2 - 1) (\int p^G(x_0|x_t)(x_0 - m_{0|t}^{x_t})^\top \Sigma_{0|t}^{-1}(x_0 - m_{0|t}^{x_t}) \mathrm{d}x_0)^{1/2} \lesssim \|x_t\|(M^2 - 1)\sqrt{d_x}
$$
$$
\lesssim \|x_t\|(M^2 - 1),
$$

here we used that we can upper bound the operator norm of a positive semidefinite matrix by its trace from the third to the fourth line. We denoted by $m_{0|t}^{x_t}$ and $\Sigma_{0|t}$ the mean and covariance of $p^G(\mathbf{x}_0^G | \mathbf{x}_t^G = x_t)$. From the second to last to the last line we used that the $m_{0|t}^{x_t}$ depends on $x_t$ linearly, and the magnitude of the linear dependence can be bounded uniformly in $t$ (see Equation 44). The integral in the last line is the variance of a standard normal random variable, which evaluates to $d_x$. Furthermore,

$$
\nabla \Phi_t(x_t) = \mathbb{E}[\nabla \Phi_0(\mathbf{x}_0) | \mathbf{x}_t = x_t] \leq L,
$$

see for example the Proof of Theorem 1 in Pidstrigach et al. (2023). Therefore,

$$
\|\nabla_{x_t} \mathbb{E}[\mathbf{x}_0^G | \mathbf{x}_t^G = x_t] - \nabla_{x_t} \mathbb{E}[\mathbf{x}_0 | \mathbf{x}_t = x_t]\|
$$
$$
\lesssim \|x_t\|(M^2 - 1) + L\|\mathbb{E}[\mathbf{x}_0 | \mathbf{x}_t = x_t]\|
$$
$$
\leq \|x_t\|(M^2 - 1) + L(\|\mathbb{E}[\mathbf{x}_0 | \mathbf{x}_t = x_t] - \mathbb{E}[\mathbf{x}_0^G | \mathbf{x}_t^G = x_t]\| + \|\mathbb{E}[\mathbf{x}_0^G | \mathbf{x}_t^G = x_t]\|)
$$
$$
\lesssim \|x_t\|(M^2 - 1) + L(M^2 - 1 + \|x_t\|) = \|x_t\|(M^2 - 1 + L) + (M^2 - 1)L =: B_{x_t}
$$

Furthermore, from Equation 31 we see that

$$
\|\nabla_{x_t} \mathbb{E}[\mathbf{x}_0 | \mathbf{x}_t = x_t]\| \leq \|x_t\|(M^2 - 1) + L(M^2 - 1 + \|x_t\|) = B_{x_t}
$$

too. We now subtract the full drifts in Equation 23 and Equation 24 and arrive at

$$
\|\nabla \tilde{f}(x_t) - f_t(x_t)\|
$$
$$
\leq B_{x_t} \times \|\mathbf{T}_1^{-1}(y - H\mathbb{E}[\mathbf{x}_0^G | \mathbf{x}_t^G = x_t])\|
$$
$$
\quad + B_{x_t} \|\mathbf{T}_1^{-1} - \mathbf{T}_2^{-1}\|_{\mathrm{op}} \|y - H\mathbb{E}[\mathbf{x}_0^G | \mathbf{x}_t^G = x_t]\|
$$
$$
\quad + B_{x_t} \|\mathbf{T}_2^{-1}\|_{\mathrm{op}} \|\mathbf{H}\mathbb{E}[\mathbf{x}_0^G | \mathbf{x}_t^G = z_t] - \mathbf{H}\mathbb{E}[\mathbf{x}_0 | \mathbf{x}_t = z_t]\|
$$
$$
\lesssim B_{x_t} + B_{x_t}(M^3 - 1) + B_{x_t} M(M^2 - 1)
$$
$$
\leq B_{x_t}(M^3 - 1) = (M^5 - 1)(\|x_t\| + L) + (M^3 - 1)L\|x_t\|.
$$

Furthermore, we get that

$$
\|\nabla \log p_t^G(x_t) - \nabla \log p_t(x_t)\| \leq \|\nabla \log \frac{p_t^G(x_t)}{p_t(x_t)}\| \leq \|\nabla_{x_t} \Phi_t(x_t)\| \leq L.
$$

Plugging in Equation 23, we get that

$$
\|\nabla \log p_t(x_t) + f_t(x_t) - p_t^{G,c}(x_t)\| \leq (M^5 - 1)(\|x_t\| + L) + (M^3 - 1)L\|x_t\| + L.
$$

Using this expression, we see that the drift change is linear in $\|x_t\|$. We now want to apply an iterated version of Novikov's condition to show that our application of Girsanov's Theorem is justified. To that end,

we follow the argument Karatzas & Shreve (2012, Corollary 3.5.16). We repeat it here for completeness. We see that

$$\mathbb{E}_{\mathbb{P}^{G,c}} \left[ \exp \left( \frac{1}{2} \int_{t_i}^{t_{i+1}} \| \nabla \log p_t(x_t) + f_t(x_t) - \nabla \log p_t^{G,c}(x_t) \|^2 \mathrm{d}t \right) \right]$$

$$\leq \mathbb{E}_{\mathbb{P}^{G,c}} \left[ \exp \left( c \int_{t_i}^{t_{i+1}} 1 + \|x_t\|^2 \mathrm{d}t \right) \right].$$

Since the expectation is regarding a Gaussian random variable $x_t$, we can make it finite as long as we pick $\Delta_i = t_{i+1} - t_i$ small enough. By setting $t_0 = 0$ and $t_1 > 0$, we can show equivalence on $[t_0, t_1]$. We can then iterate this procedure, to get equivalence on $[t_1, t_2]$ and so on. Since the lowest and highest eigenvalues of $\Sigma_t$ are bounded from below and above respectively, and $m_t$ is also bounded, the $\Delta_i$ can be bounded from below. Therefore, we get equivalence on $[0, T]$ this way in at most $\lceil T/\Delta \rceil$ steps.

Furthermore, we see that the likelihood KL-divergence can be bounded by

$$\mathbb{E}_{\mathbb{P}^{G,c}} \left[ \frac{1}{2} \int_0^t \| \nabla \log p_t(x_t) + f_t(x_t) - \nabla \log p_t^{G,c}(x_t) \|^2 \mathrm{d}t \right]$$

$$\lesssim T((M^5 - 1)(L+1) + (M^3 - 1)L + L) \leq T((M^5 - 1)(L+1) + L),$$

where we used that the mean and covariance of the Gaussian process $x_t$ (under $\mathbb{P}^{G,c}$) can be bounded by a constant only depending on $y$, $\Sigma_t$ and $m_t$. Taking the square root proves our theorem. □

## C  Variance Exploding SDE (time-rescaled Brownian motion)

A second example of an SDE used frequently in denoising-diffusion is the Variance Exploding (Song et al., 2020) or time-rescaled Brownian motion, is

$$\mathrm{d}\mathbf{x}_t = \sqrt{\frac{\mathrm{d}v(t)}{\mathrm{d}t}} \mathrm{d}\mathbf{w}_t, \quad \mathbf{x}_0 \sim p_0 = p_{\mathrm{data}} \tag{33}$$

where $(\mathbf{w}_t)_{t \in [0,T]}$ is a Brownian motion and $v : \mathbb{R}_{\geq 0} \to \mathbb{R}_{\geq 0}$ is an increasing function where $v(0) = 0$. The transition kernel of the time-rescaled Brownian motion is

$$p_{t|0}(\mathbf{x}_t | \mathbf{x}_0) = \mathcal{N}(\mathbf{x}_0, v_t \mathbf{I}_{d_\mathbf{x}}), \tag{34}$$

denoting $v(t) = v_t$. The signal $\mathbf{x}_0$ is sampled from a second SDE, the reverse process, given in forward time as

$$\mathrm{d}\mathbf{x}_t = -\nabla_{\mathbf{x}_t} \log p_t(\mathbf{x}_t) \mathrm{d}t - \sqrt{\frac{\mathrm{d}v_t}{\mathrm{d}t}} \mathrm{d}\bar{\mathbf{w}}_t, \quad \mathbf{x}_T \sim p_T. \tag{35}$$

## D  Tweedie's formula

In this section, we give an alternative derivation of the Tweedie's formula for VP-SDEs and VE-SDEs.

Tweedie's formula (Robbins, 1992; Efron, 2011) gives the minimum mean squared error (MMSE) estimator of $\mathbf{x}_0 | \mathbf{x}_t$ when $\mathbf{x}_t | \mathbf{x}_0$ is Gaussian and the score $\nabla_{\mathbf{x}_t} \log p_t(\mathbf{x}_t)$ is available. Tweedie's formula applied to the time-rescaled SDEs is given below.

### D.1  Variance Preserving SDE (time-rescaled Ornstein-Uhlenbeck process)

Using the Variance Preserving SDE or time-rescaled Ornstein-Uhlenbeck process in Equation 2, given the random variable

$$\sqrt{\frac{\alpha_t}{v_t}} \mathbf{x}_0 := \mathbf{v}_0 \sim p_{\mathbf{v}_0},$$

where, we now define $p_{\mathbf{v}_0}(\mathbf{v}_0)$ as the probability density of the random variable $\mathbf{v}_0$ evaluated at the realization $\mathbf{v}_0$, then the random variable

$$\frac{\mathbf{x}_t}{\sqrt{v_t}} := \mathbf{v}_t \sim \mathcal{N}(\mathbf{v}_0, \mathbf{I}_{d_{\mathbf{x}}}),$$

has a marginal probability density evaluated at the realization $\mathbf{v}_t$ is the convolution of the random variable $\mathbf{v}_0$ with the Gaussian kernel

$$p_{\mathbf{v}_t}(\mathbf{v}_t) = \int \phi(\mathbf{v}_t - \mathbf{v}_0)p_{\mathbf{v}_0}(\mathbf{v}_0)\mathrm{d}\mathbf{v}_0.$$

Then, Tweedie's formula (Robbins, 1992; Efron, 2011) gives

$$\mathbb{E}_{\mathbf{v}_0 \sim p_{\mathbf{v}_0|\mathbf{v}_t}}[\mathbf{v}_0] = \mathbf{v}_t + \nabla_{\mathbf{v}_t} \log p_{\mathbf{v}_t}(\mathbf{v}_t)$$

From continuous change of random variables, $p_{\mathbf{v}_t}(\mathbf{v}_t) = p_{\mathbf{x}_t}(\mathbf{v}_t\sqrt{v_t})\left|\frac{\mathrm{d}\mathbf{x}_t}{\mathrm{d}\mathbf{v}_t}\right| = p_{\mathbf{x}_t}(\mathbf{x}_t)\sqrt{v_t}$, giving $\log p_{\mathbf{v}_t}(\mathbf{v}_t) = \log p_{\mathbf{x}_t}(\mathbf{x}_t) + \log \sqrt{v_t}$, which yields

$$\nabla_{\mathbf{v}_t} \log p_{\mathbf{v}_t}(\mathbf{v}_t) = \sqrt{v_t}\nabla_{\mathbf{x}_t} \log p_{\mathbf{x}_t}(\mathbf{x}_t).$$

Now, substituting into Tweedie's formula,

$$\mathbb{E}_{\mathbf{v}_0 \sim p_{\mathbf{v}_0|\mathbf{v}_t}}[\mathbf{v}_0] = \frac{\mathbf{x}_t}{\sqrt{v_t}} + \sqrt{v_t}\nabla_{\mathbf{x}_t} \log p_{\mathbf{x}_t}(\mathbf{x}_t),$$

giving

$$\mathbb{E}_{\mathbf{x}_0 \sim p_{\mathbf{x}_0|\mathbf{x}_t}}[\mathbf{x}_0] = \frac{1}{\sqrt{\alpha_t}}\mathbf{x}_t + \frac{1}{\sqrt{\alpha_t}}v_t\nabla_{\mathbf{x}_t} \log p_{\mathbf{x}_t}(\mathbf{x}_t).$$

The covariance is given as (Robbins, 1992; Efron, 2011), $\mathbb{V}_{\mathbf{v}_0 \sim p_{\mathbf{v}_0|\mathbf{v}_t}}[\mathbf{v}_0|\mathbf{v}_t] = I + \nabla_{\mathbf{v}_t}\nabla_{\mathbf{v}_t} \log p_{\mathbf{v}_t}(\mathbf{v}_t)$, where

$$\nabla_{\mathbf{v}_t}\nabla_{\mathbf{v}_t} \log p_{\mathbf{v}_t}(\mathbf{v}_t) = \nabla_{\mathbf{v}_t}(\sqrt{v_t}\nabla_{\mathbf{x}_t} \log p_{\mathbf{x}_t}(\mathbf{x}_t)) \tag{36}$$

$$= v_t\nabla^2_{\mathbf{x}_t} \log p_{\mathbf{x}_t}(\mathbf{x}_t) \tag{37}$$

$$\tag{38}$$

and so

$$\mathbb{V}_{\mathbf{v}_0 \sim p_{\mathbf{v}_0|\mathbf{v}_t}}[\mathbf{v}_0] = \mathbf{I}_{d_{\mathbf{x}}} + v_t\nabla^2_{\mathbf{x}_t} \log p_{\mathbf{x}_t}(\mathbf{x}_t)$$

giving

$$\mathbb{V}_{\mathbf{x}_0 \sim p_{\mathbf{x}_0|\mathbf{x}_t}}[\mathbf{x}_0] = \frac{v_t}{\alpha_t}(\mathbf{I}_{d_{\mathbf{x}}} + v_t\nabla_{\mathbf{x}_t} \log p_{\mathbf{x}_t}(\mathbf{x}_t)).$$

In practice, we make the approximation $\nabla_{\mathbf{x}_t} \log p_{\mathbf{x}_t}(\mathbf{x}_t) \approx \mathbf{s}_\theta(\mathbf{x}_t, t)$, giving

$$\mathbb{E}_{\mathbf{x}_0 \sim p_{\mathbf{x}_0|\mathbf{x}_t}}[\mathbf{x}_0] \approx \mathbf{m}_{0|t} := \frac{1}{\sqrt{\alpha_t}}\mathbf{x}_t + \frac{1}{\sqrt{\alpha_t}}v_t\mathbf{s}_\theta(\mathbf{x}_t, t)$$

and

$$\mathbb{V}_{\mathbf{x}_0 \sim p_{\mathbf{x}_0|\mathbf{x}_t}}[\mathbf{x}_0] \approx \frac{v_t}{\alpha_t}(\mathbf{I}_{d_{\mathbf{x}}} + v_t\nabla_{\mathbf{x}_t}\mathbf{s}_\theta(\mathbf{x}_t, t)),$$

which yields a Gaussian approximation that matches the first and second moments,

$$p_{\mathbf{x}_0|\mathbf{x}_t}(\mathbf{x}_0|\mathbf{x}_t) \approx \mathcal{N}\left(\mathbf{m}_{0|t}, \frac{v_t}{\alpha_t}(\mathbf{I}_{d_{\mathbf{x}}} + v_t\nabla_{\mathbf{x}_t}\mathbf{s}_\theta(\mathbf{x}_t, t))\right) \tag{39}$$

$$= \mathcal{N}\left(\mathbf{m}_{0|t}, \frac{v_t}{\sqrt{\alpha_t}}\nabla_{\mathbf{x}_t}\mathbf{m}_{0|t}\right). \tag{40}$$

The matrix $\frac{v_t}{\alpha_t}(\mathbf{I}_{d_{\mathbf{x}}} + v_t\nabla_{\mathbf{x}_t}\mathbf{s}_\theta(\mathbf{x}_t, t))$ needs to be inverted to calculate the log likelihood, and therefore must be both symmetric and positive definite for all time, which puts the requirement that $\mathbf{s}_\theta(\mathbf{x}_t, t)$ can be written as the negative gradient of a potential, however it is only approximated as such $\mathbf{s}_\theta(\mathbf{x}_t, t) \approx \nabla \log p_{\mathbf{x}_t}(\mathbf{x}_t)$.

## D.2 Variance Exploding SDE (time-rescaled Brownian motion)

Using the Variance Exploding SDE (Song et al., 2020) or time-rescaled Brownian motion, given the random variable

$$\sqrt{\frac{1}{v_t}}\mathbf{x}_0 := \mathbf{v}_0 \sim p_{\mathbf{z}_0}, \tag{41}$$

a similar calculation as in D.1 gives

$$\mathbb{E}_{\mathbf{x}_0 \sim p_{\mathbf{x}_0|\mathbf{x}_t}}[\mathbf{x}_0] = \mathbf{x}_t + v_t \nabla_{\mathbf{x}_t} \log p_{\mathbf{x}_t}(\mathbf{x}_t),$$

and

$$\mathbb{V}_{\mathbf{x}_0 \sim p_{\mathbf{x}_0|\mathbf{x}_t}}[\mathbf{x}_0] = v_t(\mathbf{I}_{d_\mathbf{x}} + v_t \nabla_{\mathbf{x}_t} \log p_{\mathbf{x}_t}(\mathbf{x}_t)).$$

Again, in practice, we make the approximation $\nabla_{\mathbf{x}_t} \log p_{\mathbf{x}_t}(\mathbf{x}_t) = \mathsf{s}_\theta(\mathbf{x}_t, t)$, giving

$$\mathbb{E}_{\mathbf{x}_0 \sim p_{\mathbf{x}_0|\mathbf{x}_t}}[\mathbf{x}_0] \approx \mathbf{m}_{0|t} := \mathbf{x}_t + v_t \mathsf{s}_\theta(\mathbf{x}_t, t)$$

and

$$\mathbb{V}_{\mathbf{x}_0 \sim p_{\mathbf{x}_0|\mathbf{x}_t}}[\mathbf{x}_0] \approx v_t(\mathbf{I}_{d_\mathbf{x}} + v_t \nabla_{\mathbf{x}_t} \mathsf{s}_\theta(\mathbf{x}_t, t)),$$

which yields a Gaussian approximation that matches the first and second moments,

$$p_{\mathbf{x}_0|\mathbf{x}_t}(\mathbf{x}_0|\mathbf{x}_t) \approx \mathcal{N}\left(\mathbf{m}_{0|t}, v_t(\mathbf{I}_{d_\mathbf{x}} + v_t \nabla_{\mathbf{x}_t} \mathsf{s}_\theta(\mathbf{x}_t, t))\right) \tag{42}$$

$$= \mathcal{N}\left(\mathbf{m}_{0|t}, v_t \nabla_{\mathbf{x}_t} \mathbf{m}_{0|t}\right). \tag{43}$$

# E Algorithmic details and numerics

The code for all of the experiments and instructions to run them are available at github.com/bb515/tmpdjax and github.com/bb515/tmpdtorch.

---

**Algorithm 1** TMPD-D (Ancestral sampling, VP)

---

**input** $\mathbf{y}, \sigma_\mathbf{y}$
  $\mathbf{x}_N \sim \mathcal{N}(0, \mathbf{I}_{d_\mathbf{x}})$
  **for** $n = N - 1, \ldots, 0$ **do**
    $\mathbf{m}_{0|t} \leftarrow \frac{1}{\sqrt{\alpha_n}}(\mathbf{x}_n + v_n \mathsf{s}_\theta(\mathbf{x}_n, t_n))$
    $\mathbf{m}_{0|t}^\mathbf{y} \leftarrow \mathbf{m}_{0|t} + \frac{v_n}{\sqrt{\alpha_n}} \nabla_{\mathbf{x}_n} \mathbf{m}_{0|t} \mathbf{H}^\top (\mathbf{H} \frac{v_n}{\sqrt{\alpha_n}} \nabla_{\mathbf{x}_n} \mathbf{m}_{0|t} \mathbf{H}^\top + \sigma_\mathbf{y}^2 \mathbf{I}_{d_\mathbf{y}})^{-1}(\mathbf{y} - \mathbf{H}\mathbf{m}_{0|t})$
    $\mathbf{z}_n \sim \mathcal{N}(0, \mathbf{I}_{d_\mathbf{x}})$
    $\sigma_n \leftarrow \sqrt{(1 - \alpha_{n-1})\beta_n/(1 - \alpha_n)}$
    $\mathbf{x}_{n-1} \leftarrow \frac{\sqrt{1 - \beta_n}(1 - \alpha_{n-1})}{1 - \alpha_n}\mathbf{x}_n + \frac{\sqrt{\alpha_{n-1}}\beta_n}{1 - \alpha_n}\mathbf{m}_{0|t}^\mathbf{y} + \sigma_n \mathbf{z}_n$
  **end for**
**output** $\mathbf{x}_0$

---

## E.1 Computational Complexity

Let $N$ is the number of noise scales, and let $d_\mathbf{y}$ is the dimensions of the observation, $d_\mathbf{x}$ are the dimension of the image and $T_s$ is the time complexity of evaluating the score network. Computing a vector-Jacobian product has time complexity $T_s$. Then, the time complexity of TMPD (not including fast matrix-vector products) is $\mathcal{O}(N(d_\mathbf{y}^3 + T_s d_\mathbf{y} + T_s + T_s))$. ΠGDM comes at a smaller computational cost due to needing only $\mathcal{O}(1)$ vector-jacobian-products, instead of $\mathcal{O}(d_\mathbf{y})$, resulting in a time complexity of $\mathcal{O}(N(T_s + T_s))$. DPS comes at the same complexity as ΠGDM.

Whereas TMPD requires calculating the Jacobian which has memory complexity of atleast $\mathcal{O}(d_\mathbf{x}d_\mathbf{y})$. This is too large for high resolution image problems where the dimension of the observation is large. In comparison,

the memory complexity of ΠGDM depends on the observation operator, but for the class of problems that are explored in Song et al. (2023) is $\mathcal{O}(d_{\mathrm{x}})$. DPS comes at the same memory complexity as ΠGDM.

As mentioned in the text, we make the following approximation

$$\text{DTMPD} \quad \nabla_{\mathbf{x}_t} \log p_{\mathbf{y}|t}(\mathbf{y}|\mathbf{x}_t) \approx \nabla_{\mathbf{x}_t} \mathbf{m}_{0|t} \mathbf{H}^\top (\mathbf{H} \frac{v_t}{\sqrt{\alpha_t}} \mathrm{diag}(\nabla_{\mathbf{x}_t} \mathbf{m}_{0|t}) \mathbf{H}^\top + \sigma_{\mathrm{y}}^2 \mathbf{I}_{d_{\mathrm{y}}})^{-1} (\mathbf{y} - \mathbf{H}\mathbf{m}_{0|t}).$$

Whilst this approximation does not require a linear solve, taking out the $\mathcal{O}(Nd_{\mathbf{y}}^3)$ time complexity term, we would still like to take out the $\mathcal{O}(NT_s d_{\mathrm{y}})$ time complexity and $\mathcal{O}(d_{\mathrm{x}}d_{\mathrm{y}})$ memory complexity term from calculating and storing the Jacobian, respectively, since this is too large for solving high resolution image applications. A further approximation approximates the diagonal of the Jacobian by the row sum of the Jacobian which only requires one vector-jacobian product and brings the memory and time complexity of DTMPD down to that of ΠGDM and DPS. We use this approximation in the image experiments and find that in practice it is only $(1.5 \pm 0.1) \times$ slower than ΠGDM and DPS across all of our experiments. The row sum will be a good approximation of the diagonal when the Jacobian is approximately diagonal, which happens when there is small linear correlation between observation pixels, which we found to work well for super-resolution and inpainting. In inpainting, we have further improved the accuracy of the rowsum by instead of calculating the vector-Jacobian product evaluated at **H1**, masking out values of the vector in the vector-Jacobian product using the inpainting observation operator since they won't contribute to the diagonal values of the of the variance.

However, heuristics can be circumvented by noting in the definition of Equation 11 does not require the inverse but rather solving a system of linear equations with right hand side $\mathbf{y} - \mathbf{H}\mathbf{m}_{0|t}(\mathbf{x}_t)$. A very recent work (Rozet et al., 2024), uses the natural choice of the conjugate gradient (CG) method to solve this linear system noting that $\mathbf{H}\mathbf{C}_{0|t}\mathbf{H}^\top + \sigma_y^2 \mathbf{I}$ is symmetric positive definite (SPD) and is therefore compatible with the conjugate gradient (CG) method. The CG method is an iterative algorithm to solve linear systems of the form $\mathbf{M}\boldsymbol{v} = \boldsymbol{b}$ where SPD matrix $\mathbf{M}$ and vector $\boldsymbol{b}$ are known. Importantly, the CG method only requires implicit access to $\mathbf{M}$ through an operator that performs the matrix-vector product $\mathbf{M}\boldsymbol{v}$ given a vector $\boldsymbol{v}$. In our case, the linear system to solve is $\mathbf{y} - \mathbf{H}\mathbf{m}_{0|t} = (\mathbf{H}\mathbf{C}_{0|t}\mathbf{H}^\top + \sigma_y^2 \mathbf{I})\boldsymbol{v} = \mathbf{H}\frac{\sqrt{\alpha}}{v_t}\nabla_{\mathbf{x}_t}\mathbf{m}_{0|t}\mathbf{H}^\top \boldsymbol{v} + \sigma_y^2 \mathbf{I}\boldsymbol{v}$. The vector-jacobian product $\boldsymbol{v}^\top \mathbf{H} \nabla_{\mathbf{x}_t}^\top \mathbf{m}_{0|t}$ can be cheaply evaluated. In practice, there is no restriction on the score network to be the gradient of a potential, and the gradient of the score need not be SPD. Due to this and numerical errors, Rozet et al. (2024) observed that CG method becomes unstable after a large number of iterations and fails to reach an exact solution. Fortunately, the authors find that using very few iterations (1 to 3) of the CG method as part of the computation of the posterior score approximation leads to significant improvements over using heuristics for the covariance. Rozet et al. (2024) have successfully applied the CG method to non-sparse $\mathbf{H}$, such as accelerated MRI (where $\mathbf{H}$ is the composition of the Fourier transform and frequency subsampling).

### E.2 Gaussian

When the data distribution $p_0(\mathbf{x}_0)$ is a (multivariate) Gaussian, then the reverse SDE is a linear SDE and we can calculate all of the terms needed to sample from the target posterior using diffusion explicitly. Moreover, we can sample from the target posterior using a direct or implicit method such as Cholesky decomposition. In this simple example, we compare samples from the direct method to various conditional diffusion methods (TMPD, ΠGDM, DPM), by plotting a sample estimate of the $L^2$ Wasserstein distance between the sample and the target Gaussian measures (Givens & Shortt, 1984) by using the analytical mean and covariance of the target distribution and empirical estimate of the mean and covariance of the sample distribution. To generate $p_0(\mathbf{x}_0)$, we use an equally spaced grid of vectors $\mathbf{u}_i \in [-5.0, 5.0]^2$ for $i \in 1, 2, ..., 32^2$, pick a Matern 5/2 kernel for the covariance function $k(\mathbf{u}_i, \mathbf{u}_j) = \left(1 + \sqrt{5}|\mathbf{u}_i - \mathbf{u}_j| + \frac{5}{3}|\mathbf{u}_i - \mathbf{u}_j|^2\right) \exp\left(-\sqrt{3}|\mathbf{u}_i - \mathbf{u}_j|\right)$ which defines the prior $p_0(\mathbf{x}_0) = \mathcal{N}(\mathbf{m}_0, \mathbf{C}_0)$ covariance $\mathbf{C}_0 \in \mathbb{R}^{1024 \times 1024}$ where $\mathbf{C}_{0ij} = k(\mathbf{u}_i, \mathbf{u}_j)$ and we define the mean as a zero vector $\mathbf{m}_0 = \mathbf{0} \in \mathbb{R}^{1024}$. To compute analytically the distribution of $p_{0|\mathbf{y}}(\mathbf{x}_0|\mathbf{y})$, we sample $\mathbf{y} = \mathbf{H}\mathbf{x}_0 + \mathbf{z}, \mathbf{z} \sim \mathcal{N}(\mathbf{0}, \sigma_{\mathrm{y}}^2 \mathbf{I}_{d_{\mathrm{y}}})$ and use the standard Gaussian formula to calculate the mean and covariance of $\mathbf{x}_0|\mathbf{y}$ which in this case are a complete description of $p_{0|\mathbf{y}}(\mathbf{x}_0|\mathbf{y})$. The $L^2$ Wasserstein estimate is plotted over an increasing sample size $N \in [9, 1500]$ and for $\sigma_{\mathrm{y}} = 0.1$ in Figure 1.

We provide an illustration of the mean and uncertainty captured by the diffusion samples using 1500 samples of each diffusion model to produce a Monte-Carlo estimate of the mean and diagonal variance vector, and compare these to the exact mean and diagonal variance.

Below, we provide a calculation comparing the exact diffusion posterior for the linear diffusion posterior sde to the approximations used in ΠGDM (Song et al., 2023) and TMPD. Let us assume that the target distribution be known $p_0(\mathbf{x}_0) = \mathcal{N}(\mathbf{m}_0, \mathbf{C}_0)$. Then, via Bayes' rule,

$$
\begin{aligned}
\log p_{0|t}(\mathbf{x}_0|\mathbf{x}_t) = {} & \log p_{t|0}(\mathbf{x}_t|\mathbf{x}_0) + \log p_0(\mathbf{x}_0) + \text{constant} \\
= {} & -\frac{1}{2}(\mathbf{x}_t - \sqrt{\alpha_t}\mathbf{x}_0)^\top (v_t\mathbf{I}_{d_\mathrm{x}})^{-1}(\mathbf{x}_t - \sqrt{\alpha_t}\mathbf{x}_0) \\
& -\frac{1}{2}(\mathbf{x}_0 - \mathbf{m}_0)^\top \mathbf{C}_0^{-1}(\mathbf{x}_0 - \mathbf{m}_0) + \text{constant} \\
= {} & -\frac{1}{2}(\sqrt{\alpha_t}\mathbf{x}_0)^\top (v_t\mathbf{I}_{d_\mathrm{x}})^{-1}(\sqrt{\alpha_t}\mathbf{x}_0) + \mathbf{x}_t^\top (v_t\mathbf{I}_{d_\mathrm{x}})^{-1}(\sqrt{\alpha_t}\mathbf{x}_0) \\
& -\frac{1}{2}\mathbf{x}_0^\top \mathbf{C}_0^{-1}\mathbf{x}_0 + \mathbf{m}_0^\top \mathbf{C}_0^{-1}\mathbf{x}_0 + \text{constant} \\
= {} & -\frac{1}{2}(\mathbf{x}_0 - \mathbf{m}_t)^\top \mathbf{\Sigma}_t^{-1}(\mathbf{x}_0 - \mathbf{m}_t),
\end{aligned}
$$

so

$$
p_{0|t}(\mathbf{x}_0|\mathbf{x}_t) = \mathcal{N}(\mathbf{m}_t, \mathbf{\Sigma}_t) \tag{44}
$$

where

$$
\mathbf{\Sigma}_t = \left( (\frac{v_t}{\alpha_t}\mathbf{I}_{d_\mathrm{x}})^{-1} + \mathbf{C}_0^{-1} \right)^{-1} \tag{45}
$$

and

$$
\mathbf{m}_t = \mathbf{\Sigma}_t \left( \frac{\sqrt{\alpha_t}}{v_t}\mathbf{x}_t + \mathbf{C}_0^{-1}\mathbf{m}_0 \right).
$$

We also have that,

$$
p_t(\mathbf{x}_t) = \mathcal{N}(\sqrt{\alpha_t}\mathbf{m}_0, \mathbf{C}_t)
$$

where $\mathbf{C}_t = \alpha_t\mathbf{C}_0 + v_t\mathbf{I}_{d_\mathrm{x}}$, and thus

$$
\log p_t(\mathbf{x}_t) = -\frac{1}{2}(\mathbf{x}_t - \mathbf{m}_0\sqrt{\alpha_t})^\top \mathbf{C}_t^{-1}(\mathbf{x}_t - \mathbf{m}_0\sqrt{\alpha_t}).
$$

From

$$
p_{\mathbf{y}|t}(\mathbf{y}|\mathbf{x}_t) = \int p_{\mathbf{y}|0}(\mathbf{y}|\mathbf{x}_0)p_{0|t}(\mathbf{x}_0|\mathbf{x}_t)d\mathbf{x}_0 = \mathcal{N}(\mathbf{H}\mathbf{m}_t, \mathbf{H}\mathbf{\Sigma}_t\mathbf{H}^\top + \sigma_\mathrm{y}^2\mathbf{I}_{d_\mathrm{y}}),
$$

we have that

$$
\log p_t(\mathbf{y}|\mathbf{x}_t) = -\frac{1}{2}(\mathbf{y} - \mathbf{H}\mathbf{m}_t)^\top (\mathbf{H}\mathbf{\Sigma}_t\mathbf{H}^\top + \sigma_\mathrm{y}^2\mathbf{I}_{d_\mathrm{y}})^{-1}(\mathbf{y} - \mathbf{H}\mathbf{m}_t).
$$

The posterior score can be calculated (see Cardoso et al. (2023)) from applying Bayes' theorem,

$$
\log p_{t|\mathbf{y}}(\mathbf{x}_t|\mathbf{y}) = \log p_t(\mathbf{x}_t) + \log p_{\mathbf{y}|t}(\mathbf{y}|\mathbf{x}_t) + \text{Constant},
$$

and taking gradients with respect to the state $\mathbf{x}_t$ gives

$$
\nabla_{\mathbf{x}_t} \log p_{t|y}(\mathbf{x}_t|\mathbf{y}) = \nabla_{\mathbf{x}_t} \log p_t(\mathbf{x}_t) + \nabla_{\mathbf{x}_t} \log p_{\mathbf{y}|t}(\mathbf{y}|\mathbf{x}_t) \tag{46}
$$

$$
= -(\alpha_t\mathbf{C}_0 + v_t\mathbf{I}_{d_\mathrm{x}})^{-1}(\mathbf{x}_t - \mathbf{m}_0\sqrt{\alpha_t}) \tag{47}
$$

$$
+ \frac{\sqrt{\alpha_t}}{v_t}\mathbf{\Sigma}_t^\top \mathbf{H}^\top (\mathbf{H}\mathbf{\Sigma}_t\mathbf{H}^\top + \sigma_\mathrm{y}^2\mathbf{I}_{d_\mathrm{y}})^{-1}(\mathbf{y} - \mathbf{H}\mathbf{\Sigma}_t(\frac{\sqrt{\alpha_t}}{v_t}\mathbf{x}_t + \mathbf{C}_0^{-1}\mathbf{m}_0)). \tag{48}
$$

Setting $\mathbf{m}_0 = \mathbf{0}$ in (48) gives,

$$
\nabla_{\mathbf{x}_t} \log p_{t|\mathbf{y}}(\mathbf{x}_t|\mathbf{y}) = -(\alpha_t\mathbf{C}_0 + v_t\mathbf{I}_{d_\mathrm{x}})^{-1}\mathbf{x}_t \tag{49}
$$

$$
+ \frac{\sqrt{\alpha_t}}{v_t}\mathbf{\Sigma}_t^\top \mathbf{H}^\top (\mathbf{H}\mathbf{\Sigma}_t\mathbf{H}^\top + \sigma_\mathrm{y}^2\mathbf{I}_{d_\mathrm{y}})^{-1}(\mathbf{y} - \frac{\sqrt{\alpha_t}}{v_t}\mathbf{H}\mathbf{\Sigma}_t\mathbf{x}_t) \tag{50}
$$

### E.2.1 Comparison of the exact posterior to the approximation made in Pseudo-Inverse-Guidance

We aim to compare (50), to the approximation used in ΠGDM (Song et al., 2023);

$$p_{0|t}(\mathbf{x}_0|\mathbf{x}_t) \approx \mathcal{N}(\mathbf{m}_{0|t}, r_t^2\mathbf{I}_{d_\mathrm{x}}),$$

where $\mathbf{m}_{0|t} = \mathbf{x}_t + v_t\nabla_{\mathbf{x}_t}\log p_t(\mathbf{x}_t)$, which is the minimum mean squared error (MMSE) estimate of $\mathbf{x}_0|\mathbf{x}_t$ and $r_t$ is chosen empirically. This gives $p_{\mathbf{y}|t}(\mathbf{y}|\mathbf{x}_t) \approx \mathcal{N}(\mathbf{H}\mathbf{m}_{0|t}, r_t^2\mathbf{H}\mathbf{H}^\top + \sigma_\mathrm{y}^2\mathbf{I}_{d_\mathrm{y}})$, which in turn gives

$$\nabla_{\mathbf{x}_t}\log p_{t|y}(\mathbf{x}_t|\mathbf{y}) \approx \nabla_{\mathbf{x}_t}\log p_t(\mathbf{x}_t) \tag{51}$$
$$+ \nabla_{\mathbf{x}_t}\mathbf{m}_{0|t}\mathbf{H}^\top(r_t^2\mathbf{H}\mathbf{H}^\top + \sigma_\mathrm{y}^2\mathbf{I}_{d_\mathrm{y}})^{-1}(\mathbf{y} - \mathbf{H}\mathbf{m}_{0|t}) \tag{52}$$

which is computationally tractable in the standard diffusion model setting where the score is nonlinear and approximated as $\nabla_{\mathbf{x}_t}\log p_t(\mathbf{x}_t) \approx \mathbf{s}_\theta(\mathbf{x}_t, t)$. Substituting in the known score and again setting $\mathbf{m}_0 = \mathbf{0}$ score to compare to the linear case (50) gives,

$$\nabla_{\mathbf{x}_t}\log p_{t|\mathbf{y}}(\mathbf{x}_t|\mathbf{y}) \approx \nabla_{\mathbf{x}_t}\log p_t(\mathbf{x}_t) \tag{53}$$
$$+ \nabla_{\mathbf{x}_t}\mathbf{m}_{0|t}\mathbf{H}^\top(r_t^2\mathbf{H}\mathbf{H}^\top + \sigma_\mathrm{y}^2 I_{d_\mathrm{y}})^{-1}(y - \mathbf{H}\mathbf{m}_{0|t}) \tag{54}$$
$$= - (\alpha_t\mathbf{C}_0 + v_t\mathbf{I}_{d_\mathrm{x}})^{-1}\mathbf{x}_t \tag{55}$$
$$+ \frac{1}{\sqrt{\alpha_t}}(\mathbf{I}_{d_\mathrm{x}} - v_t(\alpha_t\mathbf{C}_0 + v_t\mathbf{I}_{d_\mathrm{x}})^{-1})\mathbf{H}^\top(r_t^2\mathbf{H}\mathbf{H}^\top + \sigma_\mathrm{y}^2 I_{d_\mathrm{y}})^{-1} \tag{56}$$
$$(\mathbf{y} - \frac{1}{\sqrt{\alpha_t}}\mathbf{H}(\mathbf{I}_{d_\mathrm{x}} - v_t(\alpha_t\mathbf{C}_0 + v_t\mathbf{I}_{d_\mathrm{x}})^{-1})\mathbf{x}_t) \tag{57}$$

comparing terms, ΠGDM is making the approximation

$$\nabla_{\mathbf{x}_t}\mathbf{m}_{0|t} = \frac{1}{\sqrt{\alpha_t}}\mathbf{I}_{d_\mathrm{x}} - \frac{v_t}{\sqrt{\alpha_t}}(\alpha_t\mathbf{C}_0 + v_t\mathbf{I}_{d_\mathrm{x}})^{-1} \approx \frac{\sqrt{\alpha_t}}{v_t}\mathbf{\Sigma}_t.$$

Note that since,

$$\frac{\sqrt{\alpha_t}}{v_t}\mathbf{\Sigma}_t = \frac{\sqrt{\alpha_t}}{v_t}(\mathbf{C}_0^{-1} + (\frac{v_t}{\alpha_t}I_{d_\mathrm{x}})^{-1})^{-1} \tag{58}$$
$$= \frac{1}{v_t\sqrt{\alpha_t}}((\alpha_t\mathbf{C}_0)^{-1} + (v_t\mathbf{I}_{d_\mathrm{x}})^{-1})^{-1} \tag{59}$$
$$= \frac{1}{v_t\sqrt{\alpha_t}}(v_t\mathbf{I} - v_t^2(\alpha_t\mathbf{C}_0 + v_t\mathbf{I}_{d_\mathrm{x}})^{-1}) \quad \text{Woodbury identity} \tag{60}$$
$$= \frac{1}{\sqrt{\alpha_t}}(\mathbf{I} - v_t(\alpha_t\mathbf{C}_0 + v_t\mathbf{I}_{d_\mathrm{x}})^{-1}) \tag{61}$$
$$= \nabla_{\mathbf{x}_t}\mathbf{m}_{0|t}, \tag{62}$$

this is exact. The other approximation ΠGDM is making is $r_t^2\mathbf{I} \approx \mathbf{\Sigma}_t$, and note that this approximation is accurate with $r_t^2 = v_t$ as $t \to 0$ but inaccurate as $t \to 1$. But the exact linear SDE can be recovered by instead using TMPD,

$$\nabla_{\mathbf{x}_t}\log p_{t|y}(\mathbf{x}_t|\mathbf{y}) = \nabla_{\mathbf{x}_t}\log p_t(\mathbf{x}_t) \tag{63}$$
$$+ \nabla_{\mathbf{x}_t}\mathbf{m}_{0|t}\mathbf{H}^\top(\mathbf{H}\frac{v_t}{\sqrt{\alpha_t}}\nabla_{\mathbf{x}_t}\mathbf{m}_{0|t}\mathbf{H}^\top + \sigma_\mathrm{y}^2 I_{d_\mathrm{y}})^{-1}(\mathbf{y} - \mathbf{H}\mathbf{m}_{0|t}) \tag{64}$$
$$= \nabla_{\mathbf{x}_t}\log p_t(\mathbf{x}_t) \tag{65}$$
$$+ \frac{\sqrt{\alpha_t}}{v_t}\mathbf{\Sigma}_t\mathbf{H}^\top(\mathbf{H}\mathbf{\Sigma}_t\mathbf{H}^\top + \sigma_\mathrm{y}^2 I_{d_\mathrm{y}})^{-1}(\mathbf{y} - \frac{\sqrt{\alpha_t}}{v_t}\mathbf{H}\mathbf{\Sigma}_t\mathbf{x}_t) \tag{66}$$

Table 3: Sliced Wasserstein for the GMM example using the reverse VP-SDEs discretized with Euler-Maruyama.

| | | $\sigma_\mathbf{y} = 0.01$ | | | | $\sigma_\mathbf{y} = 0.1$ | | | | $\sigma_\mathbf{y} = 1.0$ | | | |
|---|---|---|---|---|---|---|---|---|---|---|---|---|---|
| $d_x$ | $d_y$ | TMPD | DTMPD | ΠGDM | DPS | TMPD | DTMPD | ΠGDM | DPS | TMPD | DTMPD | ΠGDM | DPS |
| 8 | 1 | 1.5 ± 0.5 | 1.5 ± 0.5 | 1.5 ± 0.4 | 5.7 ± 2.2 | 1.4 ± 0.5 | 1.4 ± 0.5 | 1.2 ± 0.4 | 5.6 ± 2.1 | 0.9 ± 0.3 | 0.9 ± 0.3 | 0.9 ± 0.3 | 0.9 ± 0.3 |
| 8 | 2 | 0.7 ± 0.3 | 3.2 ± 1.4 | 0.4 ± 0.3 | 6.2 ± 0.8 | 0.9 ± 0.3 | 2.7 ± 1.1 | 0.5 ± 0.3 | 6.2 ± 2.4 | 0.9 ± 0.2 | 1.8 ± 0.8 | 1.0 ± 0.3 | 1.2 ± 0.4 |
| 8 | 4 | 0.3 ± 0.3 | 0.6 ± 0.4 | 0.1 ± 0.1 | - | 0.3 ± 0.2 | 0.7 ± 0.4 | 0.1 ± 0.0 | 8.4 ± 3.1 | 0.6 ± 0.2 | 0.9 ± 0.5 | 0.2 ± 0.1 | 0.3 ± 0.2 |
| 80 | 1 | 2.7 ± 0.7 | 2.7 ± 0.7 | 2.9 ± 1.4 | 9.1 ± 1.3 | 2.3 ± 0.7 | 2.3 ± 0.7 | 2.1 ± 1.1 | 4.7 ± 1.8 | 1.5 ± 0.7 | 1.5 ± 0.7 | 1.8 ± 0.8 | 1.9 ± 0.9 |
| 80 | 2 | 1.0 ± 0.5 | 3.3 ± 1.0 | 0.8 ± 0.7 | 2.2 ± 0.9 | 1.2 ± 0.5 | 3.3 ± 1.0 | 0.8 ± 0.7 | 6.0 ± 2.1 | 1.1 ± 0.2 | 2.2 ± 1.0 | 1.3 ± 0.5 | 1.5 ± 0.5 |
| 80 | 4 | 0.3 ± 0.1 | 0.9 ± 0.5 | 0.1 ± 0.0 | - | 0.4 ± 0.2 | 1.0 ± 0.5 | 0.1 ± 0.1 | 4.4 ± 1.6 | 0.9 ± 0.2 | 1.0 ± 0.4 | 0.4 ± 0.2 | 0.5 ± 0.3 |
| 800 | 1 | 3.1 ± 0.7 | 3.1 ± 0.7 | 3.2 ± 1.0 | 6.8 ± 1.2 | 2.9 ± 0.6 | 2.9 ± 0.6 | 2.8 ± 0.7 | 6.4 ± 1.5 | 1.5 ± 0.4 | 1.5 ± 0.4 | 1.3 ± 0.3 | 1.3 ± 0.3 |
| 800 | 2 | 1.3 ± 0.4 | 3.6 ± 1.2 | 0.8 ± 0.5 | 7.4 ± 0.9 | 1.3 ± 0.3 | 3.2 ± 1.1 | 0.8 ± 0.4 | 6.4 ± 1.9 | 1.2 ± 0.3 | 1.9 ± 0.5 | 1.1 ± 0.3 | 1.1 ± 0.3 |
| 800 | 4 | 0.3 ± 0.2 | 0.9 ± 0.6 | 0.6 ± 0.5 | - | 0.4 ± 0.2 | 0.9 ± 0.6 | 0.1 ± 0.0 | 5.8 ± 1.4 | 0.9 ± 0.2 | 1.1 ± 0.5 | 0.4 ± 0.2 | 0.4 ± 0.2 |

## E.3 GMM

For a given dimension $d_\mathrm{x}$, we consider $p_0$ a mixture of 25 Gaussian random variables. The components have mean $\mu_{i,j} := (8i, 8j, ..., 8i, 8j) \in \mathbb{R}^{d_\mathrm{x}}$ for $(i,j) \in -2, -1, 0, 1, 2^2$ and unit variance. We have set the associated unnormalized weights $\omega_{i,j} = 1.0$. We have set $\sigma_\delta^2 = 10^{-4}$.

Note that $p_t(\mathbf{x}_t) = \int p_{t|0}(\mathbf{x}_t|\mathbf{x}_0)p_0(\mathbf{x}_0)d\mathbf{x}_0$. As $p_0(\mathbf{x}_0)$ is a mixture of Gaussians, $p_t(\mathbf{x}_t)$ is also a mixture of Gaussians with means $\sqrt{\alpha_t}\mu_{i,j}$ and unitary variances. Therefore, using automatic differentiation libraries, we can calculate $\nabla_{\mathbf{x}_t} \log p_t(\mathbf{x}_t)$. We chose $\beta_{\max} = 500.0$ and $\beta_{\min} = 0.1$. We use 1000 timesteps for the time-discretization. For the pair of dimensions and chosen observation noise standard deviation $(d_\mathrm{x}, d_\mathrm{y}, \sigma_\mathrm{y})$ the measurement model $(y, \mathbf{H})$ is drawn as follows:

- **H**: We first draw $\tilde{\mathbf{H}} \sim \mathcal{N}(\mathbf{0}_{d_\mathrm{y} \times d_\mathrm{x}}, \mathbf{I}_{d_\mathrm{y} \times d_\mathrm{x}})$ and compute the SVD decomposition of $\tilde{\mathbf{H}} = \mathbf{USV}^\top$. Then, we sample for $(i,j) \in -2, -1, 0, 1, 2^2$, $s_{i,j}$ according to a uniform in $[0,1]$. Finally, we set $\mathbf{H} = \mathbf{U}\mathrm{diag}(s_{i,j(i,j)\in-2,-1,0,1,2^2})\mathbf{V}^\top$.

- **y**: We then draw $\mathbf{x}_* \sim p_0$ and set $\mathbf{y} := \mathbf{H}x_* + \mathbf{z}$ where $\mathbf{z} \sim \mathcal{N}(\mathbf{0}, \sigma_\mathrm{y}^2\mathbf{I}_{d_\mathrm{y}})$.

Once we have drawn both $\mathbf{x}_* \sim p_0$ and $(\mathbf{y}, \mathbf{H}, \sigma_\mathrm{y})$, the posterior can be exactly calculated using Bayes formula and gives a mixture of Gaussians with mixture components $c_{i,j}$ and associated weights $\tilde{\omega}_{i,j}$,

$$c_{i,j} := \mathcal{N}(\mathbf{\Sigma}(\mathbf{H}^\top\mathbf{y}/\sigma_\mathrm{y}^2 + \mu_{i,j}), \mathbf{\Sigma}), \tag{67}$$

$$\tilde{\omega}_i := \omega_i\mathcal{N}(\mathbf{y}; \mathbf{H}\mu_{i,j}, \sigma_\delta^2\mathbf{I}_{d_\mathrm{x}} + \mathbf{HH}^\top), \tag{68}$$

where $\mathbf{\Sigma} = (\mathbf{I}_{d_\mathrm{x}} + \frac{1}{\sigma_\delta^2}\mathbf{H}^\top\mathbf{H})^{-1}$.

**Euler-Maruyama solver** To compare the posterior distribution estimated by each algorithm with the target posterior distribution, we use $10^4$ slices for the SW distance and compare 1000 samples of the continuous SDEs defined by the TMPD, DTMPD, Song et al. (2023) and Chung et al. (2022a) approximations obtained using 1000 Euler-Maruyama time-steps with 1000 samples of the true posterior distribution. Table 3 indicates the Central Limit Theorem (CLT) 95% confidence intervals obtained by considering 20 randomly selected measurement models (**H**) for each setting $(d_\mathrm{x}, d_\mathrm{x}, \sigma_\mathrm{y})$.

**DDPM** Table 4 compares 1000 samples of TMPD-D, ΠGDM-D and DPS-D which are obtained using 1000 denoising steps and is the extended version of Table 1. We follow Cardoso et al. (2023) and compute the sliced Wasserstein distance using Wasserstein-1 distance. Figure 2 shows the first two dimensions of the estimated posterior distributions corresponding to the configurations $(80, 1)$ and $(800, 1)$ from Table 4 for one of the randomly generated measurement model (**H**). These illustrations give us insight into the behaviour of the algorithms and their accuracy in estimating the posterior distribution. We observe that TMPD-D (and the Euler-Maruyama method TMPD) is the only method that covers the modes of the posterior distribution. Finally, a direct comparison to Cardoso et al. (2023) using their original experimental setup is shown in Table 5, which shows competitive performance for posterior sampling compared to Sequential Monte-Carlo, an exact sampling method.

Table 4: Sliced Wasserstein for the GMM example using VP DDPM.

| | | $\sigma_{\mathbf{y}} = 0.01$ | | | | $\sigma_{\mathbf{y}} = 0.1$ | | | | $\sigma_{\mathbf{y}} = 1.0$ | | | |
|---|---|---|---|---|---|---|---|---|---|---|---|---|---|
| $d_x$ | $d_y$ | TMPD-D | DTMPD-D | ΠGDM-D | DPS-D | TMPD-D | DTMPD-D | ΠGDM-D | DPS-D | TMPD-D | DTMPD-D | ΠGDM-D | DPS-D |
| 8 | 1 | 1.6 ± 0.5 | 1.8 ± 0.6 | 2.6 ± 0.9 | 4.7 ± 1.5 | 1.4 ± 0.5 | 1.8 ± 0.7 | 2.2 ± 0.9 | 4.7 ± 1.6 | 0.9 ± 0.3 | 0.9 ± 0.2 | 1.5 ± 0.4 | 5.2 ± 1.3 |
| 8 | 2 | 0.7 ± 0.3 | 3.3 ± 1.5 | 2.1 ± 1.0 | 1.8 ± 1.5 | 0.9 ± 0.3 | 2.7 ± 1.1 | 1.6 ± 0.6 | 1.5 ± 0.9 | 0.9 ± 0.2 | 1.7 ± 0.8 | 2.3 ± 0.4 | 3.5 ± 1.2 |
| 8 | 4 | 0.3 ± 0.3 | 0.4 ± 0.2 | 3.8 ± 2.3 | 0.7 ± 0.6 | 0.3 ± 0.2 | 0.5 ± 0.2 | 3.8 ± 2.2 | 0.8 ± 0.6 | 0.6 ± 0.2 | 0.9 ± 0.5 | 1.8 ± 0.3 | 2.5 ± 0.9 |
| 80 | 1 | 2.7 ± 0.7 | 2.8 ± 0.9 | 3.2 ± 1.0 | 5.6 ± 1.8 | 2.3 ± 0.7 | 2.6 ± 0.9 | 2.9 ± 0.8 | 5.1 ± 1.8 | 1.5 ± 0.7 | 1.4 ± 0.6 | 1.6 ± 0.5 | 6.9 ± 1.8 |
| 80 | 2 | 1.0 ± 0.5 | 3.2 ± 1.1 | 2.8 ± 1.3 | 3.2 ± 1.9 | 1.2 ± 0.5 | 3.2 ± 1.1 | 2.7 ± 1.2 | 3.1 ± 1.9 | 1.1 ± 0.2 | 2.1 ± 1.0 | 1.4 ± 0.2 | 3.9 ± 1.2 |
| 80 | 4 | 0.3 ± 0.1 | 0.7 ± 0.4 | 0.6 ± 0.4 | 1.2 ± 1.1 | 0.4 ± 0.2 | 0.8 ± 0.4 | 0.6 ± 0.4 | 1.0 ± 1.1 | 0.9 ± 0.3 | 0.9 ± 0.4 | 0.9 ± 0.2 | 1.7 ± 0.6 |
| 800 | 1 | 3.1 ± 0.7 | 3.7 ± 0.7 | 3.5 ± 1.1 | 5.8 ± 1.6 | 2.9 ± 0.6 | 3.4 ± 0.7 | 3.3 ± 0.9 | 5.7 ± 1.6 | 1.5 ± 0.4 | 1.4 ± 0.4 | 2.0 ± 0.4 | 6.8 ± 1.0 |
| 800 | 2 | 1.4 ± 0.4 | 3.5 ± 0.7 | 3.1 ± 1.1 | 3.5 ± 1.7 | 1.3 ± 0.3 | 3.4 ± 0.7 | 2.7 ± 0.9 | 3.1 ± 1.4 | 1.2 ± 0.3 | 2.0 ± 0.4 | 2.0 ± 0.5 | 4.7 ± 1.3 |
| 800 | 4 | 0.4 ± 0.2 | 0.7 ± 0.5 | 0.4 ± 0.2 | 1.4 ± 1.0 | 0.4 ± 0.2 | 0.8 ± 0.5 | 0.4 ± 0.2 | 1.3 ± 0.9 | 0.9 ± 0.2 | 1.1 ± 0.5 | 0.6 ± 0.2 | 0.9 ± 0.4 |

Table 5: Comparison to Cardoso et al. (2023) using their original experimental setup. TMPD-D and DTMPD-D use 1000 steps of DDPM.

| $d_x$ | $d_y$ | MCG$_{\text{DIFF}}$ | TMPD-D | DTMPD-D |
|---|---|---|---|---|
| 8 | 1 | 1.43 ± 0.55 | 1.83 ± 0.5 | 1.82 ± 0.5 |
| 8 | 2 | 0.49 ± 0.24 | 0.95 ± 0.3 | 2.27 ± 0.9 |
| 8 | 4 | 0.38 ± 0.25 | 0.61 ± 0.3 | 0.72 ± 0.4 |
| 80 | 1 | 1.39 ± 0.45 | 2.81 ± 0.8 | 2.81 ± 0.8 |
| 80 | 2 | 0.67 ± 0.24 | 1.14 ± 0.4 | 2.82 ± 0.9 |
| 80 | 4 | 0.28 ± 0.14 | 0.95 ± 0.5 | 0.95 ± 0.5 |
| 800 | 1 | 2.40 ± 1.00 | 2.96 ± 0.6 | 2.96 ± 0.6 |
| 800 | 2 | 1.31 ± 0.60 | 1.60 ± 0.5 | 3.07 ± 1.1 |
| 800 | 4 | 0.47 ± 0.19 | 0.60 ± 0.2 | 0.84 ± 0.5 |

### E.4 Inpainting and super-resolution

Since the DPS-D method was derived and tuned specifically for VP-SDE, we look at the VP-SDE experiments in Section E.4.1 separately from the VE-SDE experiments in Section E.4.2. In each comparison, we use the same score network for each method and the same sampling or discretization numerical method. All methods are discretized using 1000 denoising steps. For the Markov chain methods we use DDPM and for the SDE methods we use an Euler-Maruyama discretization. In contrast to DPS(-D) ΠGDM(-D), we observe the robustness of our method across both SDEs and inpainting and super-resolution observation maps.

### E.4.1 VP-SDE

**Imagenet 256×256** For VP-SDE, our Imagenet 256×256 experiment compares SNIPS (Kawar et al., 2021) to diffusion methods (DDPM) DTMPD-D, DPS-D and ΠGDM-D, and results are shown in Table 6.

**FFHQ 256×256** For VP-SDE, our FFHQ 256×256 experiment compares diffusion methods (DDPM) DTMPD-D to DPS-D and ΠGDM-D, and results are shown in Table 7. Fig 3, Fig 4 and Fig 5 are a visual summary of Table 7, plotting the LPIPS, SSIM and FID metrics against increasing noise for different observation maps. Some uncurated samples that were used to generate Table 7 are shown in Fig 6, 7 and 8. We observe that all methods can successfully produce high quality reconstructions in the low noise regime

Table 6: Comparison to SNIPS. 4× noiseless and noisy (images have an additive noise of $\sigma_y = 0.05$) super-resolution results on ImageNet 1K (256 × 256).

| Method | 4 × noiseless super-resolution | | | 4 × noisy super-resolution | | |
|---|---|---|---|---|---|---|
| | PSNR↑ | SSIM↑ | KID↓ | PSNR↑ | SSIM↑ | KID↓ |
| SNIPS | 17.6 | 0.22 | 35.2 | 16.3 | 0.14 | 67.8 |
| ΠGDM-D | 26.0 | 0.75 | 1.30 | 20.7 | 0.43 | 17.8 |
| DPS-D | 25.1 | 0.69 | 4.08 | 23.4 | 0.63 | 3.09 |
| DTMPD-D | 26.0 | 0.75 | 1.10 | 23.1 | 0.62 | 8.85 |

Table 7: Quantitative evaluation of solving linear inverse problems for VP DDPM with increasing noise on FFHQ 256×256-1k validation dataset.

| PROBLEM | METHOD | FID ↓ | LPIPS ↓ | MSE ↓ | PSNR ↑ | SSIM ↑ |
|---|---|---|---|---|---|---|
| $\sigma_y = 0.01$ | DTMPD-D | **29.6** | 0.230 $_{\pm\,0.034}$ | 1.60E-03 $_{\pm\,7.74\text{E-}04}$ | 28.4 $_{\pm\,1.9}$ | 0.784 $_{\pm\,0.046}$ |
| 4× 'BICUBIC' | DPS-D | 31.4 | 0.234 $_{\pm\,0.048}$ | 1.90E-03 $_{\pm\,1.07\text{E-}03}$ | 27.8 $_{\pm\,2.2}$ | 0.776 $_{\pm\,0.062}$ |
| SUPER-RESOLUTION | ΠGDM-D | 29.7 | **0.198** $_{\pm\,0.037}$ | **1.56e-03** $_{\pm\,8.72\text{E-}04}$ | 28.6 $_{\pm\,2.1}$ | 0.809 $_{\pm\,0.051}$ |
| $\sigma_y = 0.05$ | DTMPD-D | 32.7 | 0.304 $_{\pm\,0.043}$ | 2.90E-03 $_{\pm\,5.64\text{E-}03}$ | 26.0 $_{\pm\,1.7}$ | 0.699 $_{\pm\,0.060}$ |
| 4× 'BICUBIC' | DPS-D | **29.3** | 0.280 $_{\pm\,0.051}$ | 2.90E-03 $_{\pm\,5.73\text{E-}03}$ | 26.0 $_{\pm\,1.8}$ | 0.719 $_{\pm\,0.066}$ |
| SUPER-RESOLUTION | ΠGDM-D | 45.1 | 0.311 $_{\pm\,0.047}$ | 3.08E-03 $_{\pm\,5.79\text{E-}03}$ | 25.7 $_{\pm\,1.7}$ | 0.682 $_{\pm\,0.062}$ |
| $\sigma_y = 0.1$ | DTMPD-D | 38.0 | 0.348 $_{\pm\,0.048}$ | 4.33E-03 $_{\pm\,4.72\text{E-}03}$ | 24.0 $_{\pm\,1.6}$ | 0.635 $_{\pm\,0.066}$ |
| 4× 'BICUBIC' | DPS-D | **30.9** | 0.318 $_{\pm\,0.051}$ | 4.06E-03 $_{\pm\,5.38\text{E-}03}$ | 24.4 $_{\pm\,1.6}$ | 0.664 $_{\pm\,0.069}$ |
| SUPER-RESOLUTION | ΠGDM-D | 119.6 | 0.589 $_{\pm\,0.047}$ | 1.10E-02 $_{\pm\,5.56\text{E-}03}$ | 19.7 $_{\pm\,1.0}$ | 0.376 $_{\pm\,0.055}$ |
| $\sigma_y = 0.2$ | DTMPD-D | 45.6 | 0.401 $_{\pm\,0.049}$ | 7.03E-03 $_{\pm\,2.56\text{E-}03}$ | 21.8 $_{\pm\,1.5}$ | 0.559 $_{\pm\,0.071}$ |
| 4× 'BICUBIC' | DPS-D | **38.1** | 0.385 $_{\pm\,0.061}$ | 7.59E-03 $_{\pm\,3.50\text{E-}03}$ | 21.6 $_{\pm\,1.8}$ | 0.570 $_{\pm\,0.081}$ |
| SUPER-RESOLUTION | ΠGDM-D | 295.7 | 0.780 $_{\pm\,0.033}$ | 5.65E-02 $_{\pm\,5.20\text{E-}03}$ | 12.5 $_{\pm\,0.4}$ | 0.117 $_{\pm\,0.035}$ |
| $\sigma_y = 0.01$ | DTMPD-D | **25.7** | 0.153 $_{\pm\,0.033}$ | 9.04E-03 $_{\pm\,7.25\text{E-}03}$ | 21.4 $_{\pm\,2.9}$ | 0.829 $_{\pm\,0.031}$ |
| 'BOX' MASK | DPS-D | 31.5 | 0.175 $_{\pm\,0.038}$ | 7.79E-03 $_{\pm\,6.89\text{E-}03}$ | 22.4 $_{\pm\,3.3}$ | 0.833 $_{\pm\,0.035}$ |
| INPAINTING | ΠGDM-D | 143.8 | 0.247 $_{\pm\,0.024}$ | 2.58E-02 $_{\pm\,7.11\text{E-}03}$ | 16.1 $_{\pm\,1.3}$ | 0.759 $_{\pm\,0.017}$ |
| $\sigma_y = 0.05$ | DTMPD-D | **27.0** | 0.240 $_{\pm\,0.038}$ | 9.68E-03 $_{\pm\,6.62\text{E-}03}$ | 20.9 $_{\pm\,2.6}$ | 0.760 $_{\pm\,0.036}$ |
| 'BOX' MASK | DPS-D | 30.7 | 0.228 $_{\pm\,0.046}$ | 8.28E-03 $_{\pm\,7.93\text{E-}03}$ | 22.0 $_{\pm\,3.1}$ | 0.782 $_{\pm\,0.047}$ |
| INPAINTING | ΠGDM-D | 159.3 | 0.448 $_{\pm\,0.046}$ | 2.71E-02 $_{\pm\,7.91\text{E-}03}$ | 15.8 $_{\pm\,1.2}$ | 0.504 $_{\pm\,0.080}$ |
| $\sigma_y = 0.1$ | DTMPD-D | 29.6 | 0.292 $_{\pm\,0.049}$ | 1.06E-02 $_{\pm\,1.15\text{E-}02}$ | 20.6 $_{\pm\,2.5}$ | 0.709 $_{\pm\,0.053}$ |
| 'BOX' MASK | DPS-D | **29.3** | 0.259 $_{\pm\,0.049}$ | 8.03E-03 $_{\pm\,6.97\text{E-}03}$ | 22.0 $_{\pm\,2.9}$ | 0.746 $_{\pm\,0.051}$ |
| INPAINTING | ΠGDM-D | 165.7 | 0.539 $_{\pm\,0.083}$ | 2.84E-02 $_{\pm\,6.57\text{E-}03}$ | 15.6 $_{\pm\,1.0}$ | 0.418 $_{\pm\,0.185}$ |
| $\sigma_y = 0.2$ | DTMPD-D | **33.8** | 0.346 $_{\pm\,0.061}$ | 1.13E-02 $_{\pm\,8.65\text{E-}03}$ | 20.2 $_{\pm\,2.4}$ | 0.649 $_{\pm\,0.070}$ |
| 'BOX' MASK | DPS-D | 34.9 | 0.337 $_{\pm\,0.056}$ | 8.01E-03 $_{\pm\,4.22\text{E-}03}$ | 21.5 $_{\pm\,2.0}$ | 0.662 $_{\pm\,0.067}$ |
| INPAINTING | ΠGDM-D | 199.7 | 0.590 $_{\pm\,0.099}$ | 3.17E-02 $_{\pm\,1.01\text{E-}02}$ | 15.2 $_{\pm\,1.4}$ | 0.461 $_{\pm\,0.202}$ |
| $\sigma_y = 0.01$ | DTMPD-D | **24.6** | 0.090 $_{\pm\,0.033}$ | 4.87E-04 $_{\pm\,6.27\text{E-}04}$ | 34.3 $_{\pm\,2.8}$ | 0.931 $_{\pm\,0.036}$ |
| 'RANDOM' MASK | DPS-D | 32.7 | 0.137 $_{\pm\,0.033}$ | 5.57E-04 $_{\pm\,5.02\text{E-}04}$ | 33.4 $_{\pm\,2.6}$ | 0.913 $_{\pm\,0.031}$ |
| INPAINTING | ΠGDM-D | 24.7 | 0.069 $_{\pm\,0.023}$ | 4.71E-04 $_{\pm\,5.18\text{E-}04}$ | 34.5 $_{\pm\,3.0}$ | 0.940 $_{\pm\,0.026}$ |
| $\sigma_y = 0.05$ | DTMPD-D | **27.1** | **0.187** $_{\pm\,0.032}$ | 8.86E-04 $_{\pm\,4.84\text{E-}04}$ | 30.9 $_{\pm\,1.8}$ | **0.851** $_{\pm\,0.033}$ |
| 'RANDOM' MASK | DPS-D | 38.0 | 0.256 $_{\pm\,0.049}$ | 1.74E-03 $_{\pm\,1.04\text{E-}03}$ | 28.1 $_{\pm\,2.0}$ | 0.794 $_{\pm\,0.056}$ |
| INPAINTING | ΠGDM-D | 44.8 | 0.315 $_{\pm\,0.043}$ | 2.15E-03 $_{\pm\,6.65\text{E-}04}$ | 26.8 $_{\pm\,1.2}$ | 0.647 $_{\pm\,0.068}$ |
| $\sigma_y = 0.1$ | DTMPD-D | **29.9** | **0.247** $_{\pm\,0.038}$ | 1.74E-03 $_{\pm\,8.94\text{E-}03}$ | 28.7 $_{\pm\,1.8}$ | **0.787** $_{\pm\,0.049}$ |
| 'RANDOM' MASK | DPS-D | 34.8 | 0.274 $_{\pm\,0.054}$ | 2.21E-03 $_{\pm\,1.17\text{E-}03}$ | 27.1 $_{\pm\,2.1}$ | 0.763 $_{\pm\,0.065}$ |
| INPAINTING | ΠGDM-D | 62.7 | 0.490 $_{\pm\,0.059}$ | 6.71E-03 $_{\pm\,1.80\text{E-}03}$ | 22.0 $_{\pm\,1.7}$ | 0.410 $_{\pm\,0.129}$ |
| $\sigma_y = 0.2$ | DTMPD-D | **34.6** | **0.306** $_{\pm\,0.046}$ | 2.71E-03 $_{\pm\,2.89\text{E-}03}$ | 26.1 $_{\pm\,1.7}$ | **0.714** $_{\pm\,0.060}$ |
| 'RANDOM' MASK | DPS-D | 35.9 | 0.317 $_{\pm\,0.059}$ | 3.59E-03 $_{\pm\,1.73\text{E-}03}$ | 24.9 $_{\pm\,1.9}$ | 0.701 $_{\pm\,0.073}$ |
| INPAINTING | ΠGDM-D | 104.1 | 0.623 $_{\pm\,0.109}$ | 1.77E-02 $_{\pm\,8.62\text{E-}03}$ | 18.6 $_{\pm\,3.8}$ | 0.316 $_{\pm\,0.218}$ |

but, visually, only DPS-D and DTMPD-D successfully produce high quality reconstructions in the high noise regime.

Whereas DPS-D requires a hyperparameter search, there are no hyperparameters for DTMPD-D. For ΠGDM-D we use the algorithm and default hyperparameters as described in Song et al. (2023). For DPS-D we use the algorithm in the codebase provided by the authors Chung et al. (2022a) and we use their suggested hyperparameters for this task, such as step-size and using static-thresholding (clipping the denoised image at each step to a range $[-1, 1]$) whereas DTMPD-D does not require any hyperparameter tuning or static-thresholding.

**CIFAR10 64×64** Our CIFAR-10 64×64 experiment compares TMPD to DPS and ΠGDM, and also compares diffusion methods (DDPM) in Table 8 and score-based methods (discretized with Euler-Maruyama) in Table 9. Various samples used to produce the figures in Tables 8 and 9 are shown in Fig 10 and 11.

We found the hyperparameter suggested for the DDIM method for the VP-SDE in Song et al. (2023) $r_t^2 = v_t/(\alpha_t + v_t)$, which is calculated by assuming the data distribution $p_0(\mathbf{x}_0)$ is a standard normal and

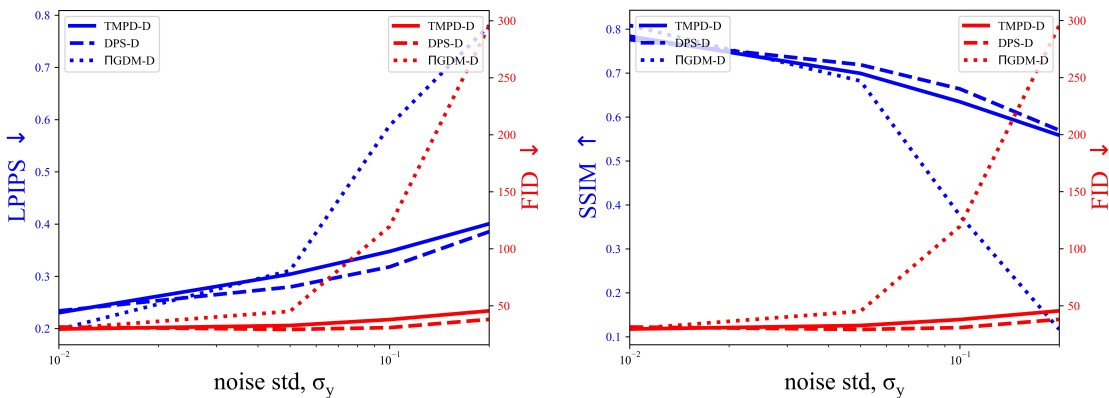

Figure 3: 4× bicubic super-resolution FID vs LPIPS (left) and SSIM (right) using the VP-SDE on FFHQ-1k validation dataset for increasing observation noise.

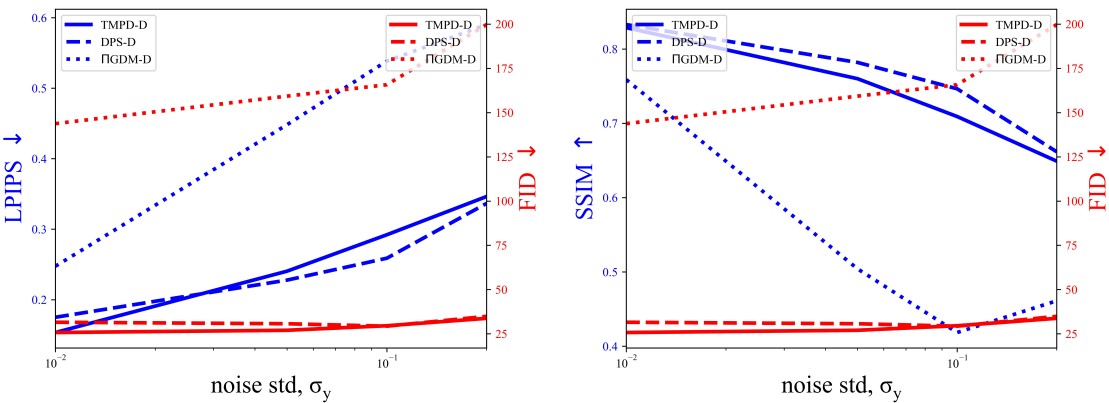

Figure 4: 'box' mask inpainting FID vs LPIPS (left) and SSIM (right) using the VP-SDE on FFHQ-1k validation dataset for increasing observation noise.

then calculating the posterior variance, to give unstable solutions for the algorithm given in Song et al. (2023). To make the method stable, we instead plug the ΠGDM posterior score approximation into a DDIM sampler in a similar way to Algorithm 1, which, for the VPSDE, brings the algorithm ΠGDM-D closer to our method; we are then able to choose $r_t^2 = v_t/(\alpha_t + v_t)$ for both VP DDIM and VP-SDE methods.

For DDPM we use the step-size constant suggested in Chung et al. (2022a) for inpainting, $\zeta_i = \zeta'/\|\mathbf{y} - \mathbf{Hm}_{0|t}\|$, where we tune $\zeta'$ over the suggested range of $\zeta' \in [0.1, 1.0]$ in Chung et al. (2022a) across LPIPS, MSE, PSNR and SSIM, as shown in Fig. 9 for each inverse problem (each line in the Tables 8 and 9).

### E.4.2  VE-SDE

**FFHQ 256×256** For VE-SDE, our FFHQ 256×256 experiment compares diffusion methods (DDPM) DTMPD-D to DPS-D and ΠGDM-D, and results are shown in Table 10. Fig 12 and Fig 13 are a visual summary of Table 10, plotting the LPIPS, SSIM and FID metrics against increasing noise. Some uncurated samples that were used to generate Table 10 are shown in Fig 14 and 15. Since the DPS-D and PiGDM algorithms were developed for the VP-SDE, the method is not performant for the VE-SDE, and the ΠGDM

Table 8: Noisy observation inpainting and super-resolution for VP DDPM on CIFAR-10 1k validation set.

| PROBLEM | METHOD | FID ↓ | LPIPS ↓ | MSE ↓ | PSNR ↑ | SSIM ↑ |
|---|---|---|---|---|---|---|
| $\sigma_y = 0.01$ 'BOX' MASK INPAINTING | DTMPD-D | 33.7 | $0.090_{\pm 0.048}$ | $0.007_{\pm 0.034}$ | $23.8_{\pm 3.7}$ | $0.784_{\pm 0.073}$ |
| | DPS-D | **31.5** | **0.064**$_{\pm 0.033}$ | $0.004_{\pm 0.003}$ | $25.8_{\pm 3.6}$ | $0.841_{\pm 0.068}$ |
| | ΠGDM-D | 37.1 | $0.316_{\pm 0.108}$ | $0.012_{\pm 0.029}$ | $19.9_{\pm 1.9}$ | $0.546_{\pm 0.143}$ |
| $\sigma_y = 0.01$ 'HALF' MASK INPAINTING | DTMPD-D | 40.4 | $0.272_{\pm 0.071}$ | $0.028_{\pm 0.020}$ | $16.4_{\pm 3.0}$ | $0.584_{\pm 0.071}$ |
| | DPS-D | **33.1** | $0.221_{\pm 0.072}$ | $0.028_{\pm 0.021}$ | $16.8_{\pm 3.6}$ | $0.637_{\pm 0.093}$ |
| | ΠGDM-D | 35.8 | $0.397_{\pm 0.103}$ | $0.031_{\pm 0.020}$ | $15.9_{\pm 2.8}$ | $0.419_{\pm 0.132}$ |
| $\sigma_y = 0.05$ 'BOX' MASK INPAINTING | DTMPD-D | 35.9 | $0.128_{\pm 0.061}$ | $0.006_{\pm 0.011}$ | $23.5_{\pm 3.3}$ | $0.763_{\pm 0.079}$ |
| | DPS-D | **31.1** | **0.078**$_{\pm 0.037}$ | $0.004_{\pm 0.003}$ | $25.5_{\pm 3.3}$ | **0.830**$_{\pm 0.070}$ |
| | ΠGDM-D | 36.3 | $0.319_{\pm 0.110}$ | $0.012_{\pm 0.031}$ | $20.0_{\pm 1.9}$ | $0.545_{\pm 0.142}$ |
| $\sigma_y = 0.05$ 'HALF' MASK INPAINTING | DTMPD-D | 40.4 | $0.292_{\pm 0.075}$ | $0.029_{\pm 0.020}$ | $16.4_{\pm 3.1}$ | $0.572_{\pm 0.076}$ |
| | DPS-D | **32.5** | $0.230_{\pm 0.075}$ | $0.028_{\pm 0.023}$ | $16.7_{\pm 3.5}$ | $0.626_{\pm 0.096}$ |
| | ΠGDM-D | 35.6 | $0.398_{\pm 0.109}$ | $0.032_{\pm 0.021}$ | $15.9_{\pm 2.8}$ | $0.421_{\pm 0.133}$ |
| $\sigma_y = 0.1$ 'BOX' MASK INPAINTING | DTMPD-D | 38.9 | $0.168_{\pm 0.072}$ | $0.007_{\pm 0.012}$ | $22.5_{\pm 2.9}$ | $0.728_{\pm 0.081}$ |
| | DPS-D | **31.6** | **0.101**$_{\pm 0.043}$ | $0.004_{\pm 0.003}$ | $24.5_{\pm 2.9}$ | $0.807_{\pm 0.078}$ |
| | ΠGDM-D | 36.3 | $0.318_{\pm 0.109}$ | $0.012_{\pm 8.553}$ | $19.7_{\pm 1.8}$ | $0.546_{\pm 0.140}$ |
| $\sigma_y = 0.1$ 'HALF' MASK INPAINTING | DTMPD-D | 43.8 | $0.350_{\pm 0.088}$ | $0.030_{\pm 0.028}$ | $16.2_{\pm 2.9}$ | $0.547_{\pm 0.076}$ |
| | DPS-D | **33.0** | $0.522_{\pm 0.097}$ | $0.031_{\pm 0.022}$ | $16.3_{\pm 3.3}$ | $0.602_{\pm 0.097}$ |
| | ΠGDM-D | 36.2 | $0.276_{\pm 0.085}$ | $0.034_{\pm 0.023}$ | $15.5_{\pm 2.7}$ | $0.412_{\pm 0.128}$ |
| $\sigma_y = 0.01$ 2× 'NEAREST' SUPER-RESOLUTION | DTMPD-D | 33.2 | $0.117_{\pm 0.051}$ | $0.004_{\pm 0.004}$ | $24.7_{\pm 3.0}$ | $0.835_{\pm 0.071}$ |
| | DPS-D | **32.5** | **0.099**$_{\pm 0.044}$ | $0.004_{\pm 0.003}$ | $25.1_{\pm 3.1}$ | $0.847_{\pm 0.073}$ |
| | ΠGDM-D | 35.6 | $0.407_{\pm 0.118}$ | $0.016_{\pm 0.006}$ | $18.2_{\pm 1.7}$ | $0.442_{\pm 0.152}$ |
| $\sigma_y = 0.01$ 4× 'BICUBIC' SUPER-RESOLUTION | DTMPD-D | 41.5 | $0.278_{\pm 0.084}$ | $0.011_{\pm 0.006}$ | $20.4_{\pm 2.7}$ | $0.563_{\pm 0.114}$ |
| | DPS-D | **33.9** | $0.220_{\pm 0.079}$ | $0.010_{\pm 0.006}$ | $20.8_{\pm 3.0}$ | $0.609_{\pm 0.135}$ |
| | ΠGDM-D | 39.4 | $0.279_{\pm 0.081}$ | $0.011_{\pm 0.006}$ | $20.1_{\pm 2.5}$ | $0.546_{\pm 0.111}$ |
| $\sigma_y = 0.05$ 2× 'NEAREST' SUPER-RESOLUTION | DTMPD-D | **33.9** | $0.156_{\pm 0.065}$ | $0.005_{\pm 0.025}$ | $24.1_{\pm 2.8}$ | $0.810_{\pm 0.079}$ |
| | DPS-D | 34.4 | $0.127_{\pm 0.048}$ | $0.004_{\pm 0.003}$ | $24.4_{\pm 2.7}$ | $0.825_{\pm 0.070}$ |
| | ΠGDM-D | 35.1 | $0.407_{\pm 0.118}$ | $0.016_{\pm 0.006}$ | $18.2_{\pm 1.7}$ | $0.440_{\pm 0.153}$ |
| $\sigma_y = 0.05$ 4× 'BICUBIC' SUPER-RESOLUTION | DTMPD-D | 38.3 | $0.332_{\pm 0.093}$ | $0.013_{\pm 0.032}$ | $19.6_{\pm 2.4}$ | $0.501_{\pm 0.116}$ |
| | DPS-D | 38.2 | $0.265_{\pm 0.087}$ | $0.011_{\pm 0.006}$ | $20.2_{\pm 2.4}$ | $0.567_{\pm 0.128}$ |
| | ΠGDM-D | **23.2** | $0.522_{\pm 0.101}$ | $0.034_{\pm 0.012}$ | $15.0_{\pm 1.7}$ | $0.215_{\pm 0.114}$ |
| $\sigma_y = 0.1$ 2× 'NEAREST' SUPER-RESOLUTION | DTMPD-D | 35.0 | $0.208_{\pm 0.085}$ | $0.006_{\pm 0.005}$ | $23.1_{\pm 2.5}$ | $0.760_{\pm 0.091}$ |
| | DPS-D | 32.1 | $0.152_{\pm 0.058}$ | $0.005_{\pm 0.003}$ | $23.8_{\pm 2.5}$ | $0.802_{\pm 0.074}$ |
| | ΠGDM-D | 35.0 | $0.407_{\pm 0.122}$ | $0.017_{\pm 0.006}$ | $18.1_{\pm 1.8}$ | $0.435_{\pm 0.150}$ |
| $\sigma_y = 0.1$ 4× 'BICUBIC' SUPER-RESOLUTION | DTMPD-D | 36.8 | $0.378_{\pm 0.102}$ | $0.015_{\pm 0.007}$ | $18.7_{\pm 2.1}$ | $0.435_{\pm 0.121}$ |
| | DPS-D | 34.8 | $0.308_{\pm 0.095}$ | $0.013_{\pm 0.006}$ | $19.4_{\pm 2.2}$ | $0.513_{\pm 0.132}$ |
| | ΠGDM-D | **23.0** | $0.521_{\pm 0.104}$ | $0.035_{\pm 0.013}$ | $14.9_{\pm 1.7}$ | $0.212_{\pm 0.117}$ |

Table 9: Noisy observation inpainting and super-resolution for the reverse VP-SDEs on CIFAR-10 1k validation set.

| PROBLEM | METHOD | FID ↓ | LPIPS ↓ | MSE ↓ | PSNR ↑ | SSIM ↑ |
|---|---|---|---|---|---|---|
| $\sigma_y = 0.01$ | TMPD | **33.2** | $0.088_{\pm 0.050}$ | $0.007_{\pm 0.019}$ | $23.8_{\pm 3.7}$ | $0.785_{\pm 0.071}$ |
| 'BOX' MASK | DPS | 61.8 | $0.646_{\pm 0.078}$ | $0.111_{\pm 0.049}$ | $9.9_{\pm 1.9}$ | $0.050_{\pm 0.068}$ |
| INPAINTING | ΠGDM | 34.8 | $0.073_{\pm 0.035}$ | $0.004_{\pm 0.004}$ | $25.1_{\pm 3.5}$ | $0.816_{\pm 0.068}$ |
| $\sigma_y = 0.01$ | TMPD | 37.4 | $0.267_{\pm 0.068}$ | $0.030_{\pm 0.036}$ | $16.3_{\pm 3.0}$ | $0.581_{\pm 0.070}$ |
| 'HALF' MASK | DPS | 65.8 | $0.645_{\pm 0.079}$ | $0.112_{\pm 0.050}$ | $9.9_{\pm 1.9}$ | $0.052_{\pm 0.069}$ |
| INPAINTING | ΠGDM | **33.3** | $0.236_{\pm 0.068}$ | $0.027_{\pm 0.021}$ | $17.0_{\pm 3.6}$ | $0.620_{\pm 0.087}$ |
| $\sigma_y = 0.05$ | TMPD | **33.5** | $0.117_{\pm 0.052}$ | $0.006_{\pm 0.017}$ | $23.5_{\pm 3.2}$ | $0.763_{\pm 0.074}$ |
| 'BOX' MASK | DPS | 62.2 | $0.647_{\pm 0.078}$ | $0.110_{\pm 0.048}$ | $10.0_{\pm 1.9}$ | $0.055_{\pm 0.067}$ |
| INPAINTING | ΠGDM | 35.2 | $0.103_{\pm 0.047}$ | $0.004_{\pm 0.003}$ | $24.8_{\pm 2.9}$ | $0.798_{\pm 0.074}$ |
| $\sigma_y = 0.05$ | TMPD | 37.9 | $0.286_{\pm 0.073}$ | $0.030_{\pm 0.023}$ | $16.2_{\pm 3.1}$ | $0.567_{\pm 0.075}$ |
| 'HALF' MASK | DPS | 65.1 | $0.647_{\pm 0.080}$ | $0.117_{\pm 0.054}$ | $9.7_{\pm 2.0}$ | $0.052_{\pm 0.063}$ |
| INPAINTING | ΠGDM | **34.1** | $0.256_{\pm 0.078}$ | $0.027_{\pm 0.022}$ | $16.8_{\pm 3.3}$ | $0.605_{\pm 0.094}$ |
| $\sigma_y = 0.1$ | TMPD | **35.1** | $0.159_{\pm 0.065}$ | $0.007_{\pm 0.006}$ | $22.6_{\pm 2.8}$ | $0.731_{\pm 0.078}$ |
| 'BOX' MASK | DPS | 61.9 | $0.647_{\pm 0.074}$ | $0.112_{\pm 0.052}$ | $9.9_{\pm 1.9}$ | $0.054_{\pm 0.071}$ |
| INPAINTING | ΠGDM | 36.2 | $0.140_{\pm 0.059}$ | $0.005_{\pm 0.003}$ | $23.7_{\pm 2.6}$ | $0.763_{\pm 0.083}$ |
| $\sigma_y = 0.1$ | TMPD | 41.9 | $0.310_{\pm 0.077}$ | $0.031_{\pm 0.020}$ | $16.0_{\pm 2.9}$ | $0.541_{\pm 0.078}$ |
| 'HALF' MASK | DPS | 81.0 | $0.646_{\pm 0.082}$ | $0.112_{\pm 0.052}$ | $10.0_{\pm 2.0}$ | $0.052_{\pm 0.069}$ |
| INPAINTING | ΠGDM | **38.2** | $0.279_{\pm 0.074}$ | $0.029_{\pm 0.021}$ | $16.4_{\pm 3.2}$ | $0.577_{\pm 0.095}$ |
| $\sigma_y = 0.01$ | TMPD | **32.4** | **$0.123_{\pm 0.055}$** | $0.005_{\pm 0.008}$ | $24.5_{\pm 3.0}$ | $0.826_{\pm 0.077}$ |
| 2× 'NEAREST' | DPS | 71.6 | $0.645_{\pm 0.079}$ | $0.118_{\pm 0.054}$ | $9.7_{\pm 2.0}$ | $0.049_{\pm 0.068}$ |
| SUPER-RESOLUTION | ΠGDM | 41.7 | $0.192_{\pm 0.079}$ | $0.005_{\pm 0.003}$ | $23.3_{\pm 2.4}$ | $0.773_{\pm 0.079}$ |
| $\sigma_y = 0.01$ | TMPD | **22.9** | $0.525_{\pm 0.098}$ | $0.035_{\pm 0.013}$ | $14.9_{\pm 1.7}$ | $0.206_{\pm 0.109}$ |
| 4× 'BICUBIC' | DPS | 77.9 | $0.640_{\pm 0.076}$ | $0.120_{\pm 0.060}$ | $9.7_{\pm 2.1}$ | $0.046_{\pm 0.070}$ |
| SUPER-RESOLUTION | ΠGDM | 36.7 | $0.461_{\pm 0.093}$ | $0.023_{\pm 0.009}$ | $16.7_{\pm 1.7}$ | $0.305_{\pm 0.106}$ |
| $\sigma_y = 0.05$ | TMPD | **32.7** | **$0.158_{\pm 0.060}$** | $0.005_{\pm 0.003}$ | $23.9_{\pm 2.6}$ | $0.800_{\pm 0.075}$ |
| 2× 'NEAREST' | DPS | 72.5 | $0.648_{\pm 0.078}$ | $0.118_{\pm 0.053}$ | $9.7_{\pm 1.9}$ | $0.050_{\pm 0.069}$ |
| SUPER-RESOLUTION | ΠGDM | 37.3 | $0.229_{\pm 0.086}$ | $0.006_{\pm 0.003}$ | $22.6_{\pm 2.1}$ | $0.735_{\pm 0.088}$ |
| $\sigma_y = 0.05$ | TMPD | 35.7 | **$0.328_{\pm 0.091}$** | **$0.013_{\pm 0.011}$** | **$19.5_{\pm 2.3}$** | **$0.495_{\pm 0.117}$** |
| 4× 'BICUBIC' | DPS | 77.9 | $0.641_{\pm 0.071}$ | $0.120_{\pm 0.061}$ | $9.7_{\pm 2.1}$ | $0.047_{\pm 0.070}$ |
| SUPER-RESOLUTION | ΠGDM | **34.8** | $0.479_{\pm 0.094}$ | $0.026_{\pm 0.010}$ | $16.2_{\pm 1.7}$ | $0.276_{\pm 0.112}$ |
| $\sigma_y = 0.1$ | TMPD | **32.9** | **$0.207_{\pm 0.081}$** | $0.006_{\pm 0.005}$ | $23.0_{\pm 2.4}$ | $0.753_{\pm 0.088}$ |
| 2× 'NEAREST' | DPS | 72.2 | $0.644_{\pm 0.077}$ | $0.118_{\pm 0.056}$ | $9.7_{\pm 2.0}$ | $0.051_{\pm 0.067}$ |
| SUPER-RESOLUTION | ΠGDM | 35.9 | $0.273_{\pm 0.098}$ | $0.007_{\pm 0.003}$ | $21.7_{\pm 2.1}$ | $0.682_{\pm 0.105}$ |
| $\sigma_y = 0.1$ | TMPD | 34.7 | **$0.379_{\pm 0.096}$** | **$0.016_{\pm 0.007}$** | **$18.5_{\pm 2.0}$** | **$0.429_{\pm 0.120}$** |
| 4× 'BICUBIC' | DPS | 78.3 | $0.639_{\pm 0.074}$ | $0.120_{\pm 0.060}$ | $9.7_{\pm 2.1}$ | $0.050_{\pm 0.068}$ |
| SUPER-RESOLUTION | ΠGDM | **34.4** | $0.492_{\pm 0.097}$ | $0.028_{\pm 0.010}$ | $15.8_{\pm 1.7}$ | $0.258_{\pm 0.112}$ |

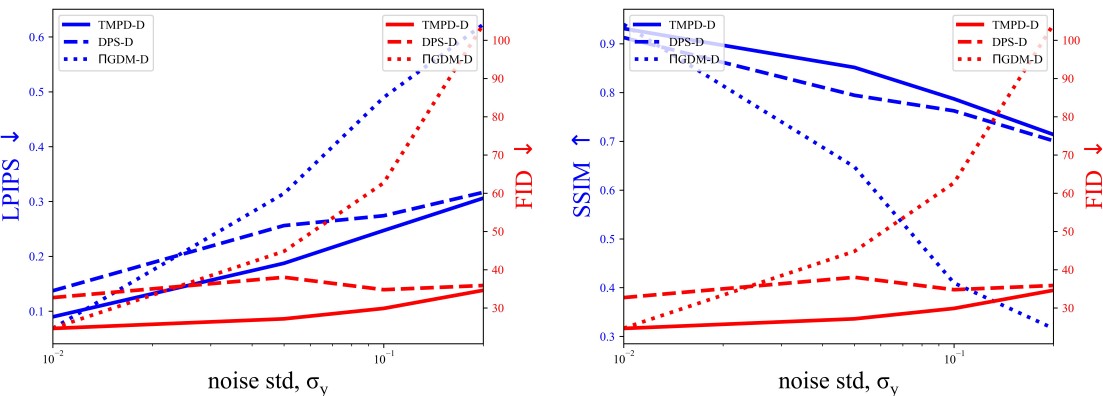

Figure 5: 'random' mask inpainting FID vs LPIPS (left) and SSIM (right) using the VP-SDE on FFHQ-1k validation dataset for increasing observation noise.

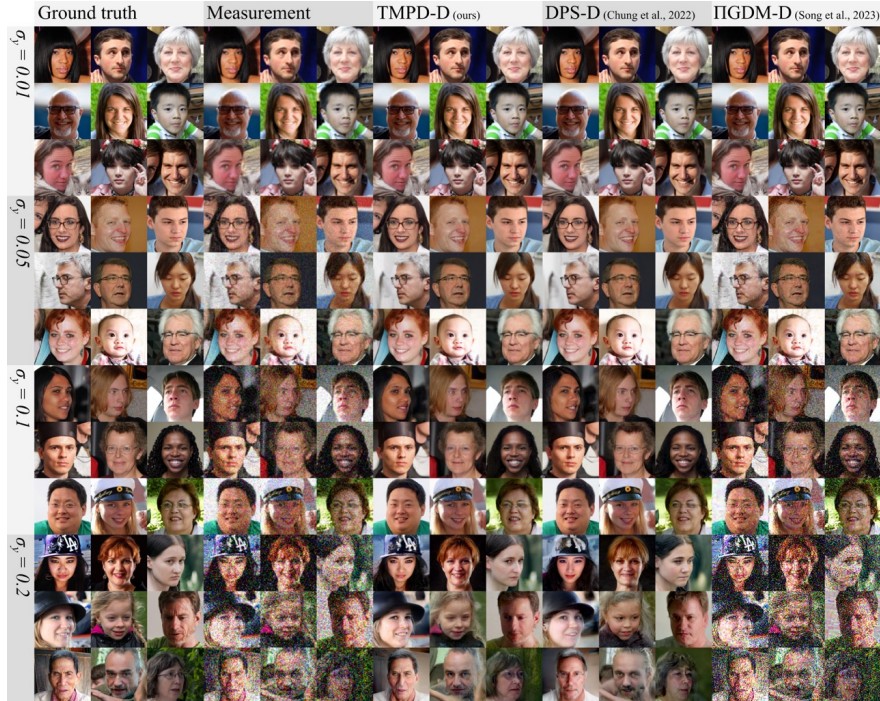

Figure 6: 4× bicubic super-resolution samples from the VP-SDE on FFHQ across a range of observation noise.

algorithm was only performant in low noise settings, whereas DTMPD-D is performant across a wide range of noise levels.

It has been observed that ΠGDM-D performs worse for 1000 steps than 100 steps (see (Mardani et al., 2023, Section 5)). We also provide results in Table 11 for 100 steps of DTMPD-D and ΠGDM-D for the same FFHQ VE-SDE model, to compare to ΠGDM-D in its optimal setting.

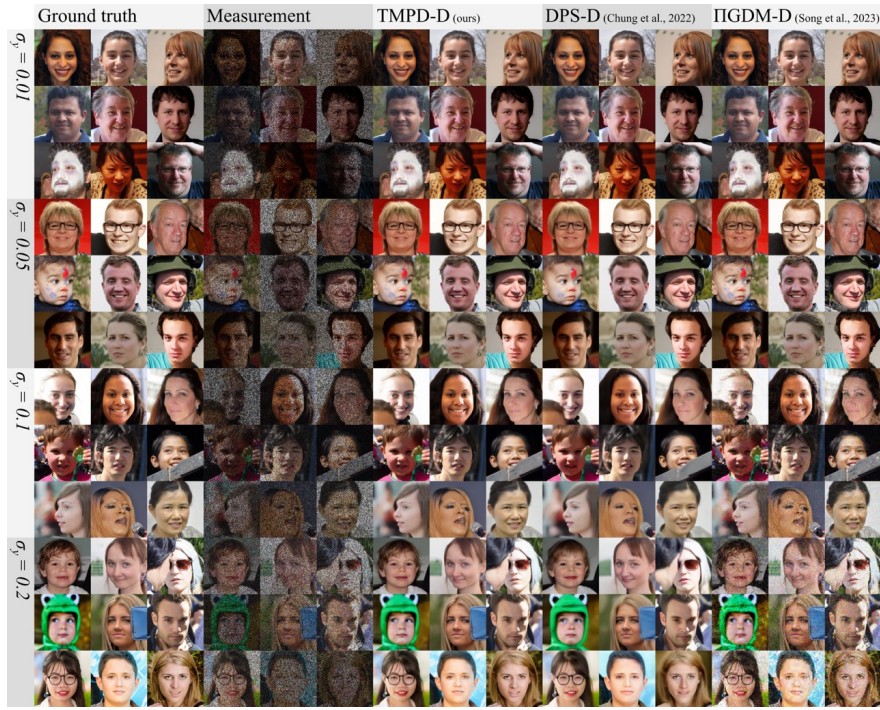

Figure 7: 'Random' mask inpainting samples from the VP-SDE on FFHQ across a range of observation noise.

Table 10: Quantitative evaluation of solving linear inverse problems for VE DDPM for 1000 steps with increasing noise on FFHQ 256×256-1k validation dataset.

| PROBLEM | METHOD | FID ↓ | LPIPS ↓ | MSE ↓ | PSNR ↑ | SSIM ↑ |
|---|---|---|---|---|---|---|
| $\sigma_y = 0.01$ | DTMPD-D | **32.3** | **0.203** $\pm$ 0.039 | 1.57E-03 $\pm$ 8.89E-04 | 28.6 $\pm$ 2.1 | **0.810** $\pm$ 0.052 |
| 4× 'BICUBIC' | DPS-D | 47.0 | 0.273 $\pm$ 0.031 | 1.70E-03 $\pm$ 8.36E-04 | 28.1 $\pm$ 1.8 | 0.747 $\pm$ 0.037 |
| SUPER-RESOLUTION | ΠGDM-D | 37.4 | 0.244 $\pm$ 0.030 | 1.74E-03 $\pm$ 9.56E-04 | 28.1 $\pm$ 2.0 | 0.755 $\pm$ 0.042 |
| $\sigma_y = 0.05$ | DTMPD-D | **32.1** | **0.268** $\pm$ 0.048 | 2.62E-03 $\pm$ 1.17E-03 | **26.2** $\pm$ 1.8 | **0.733** $\pm$ 0.066 |
| 4× 'BICUBIC' | DPS-D | 105.9 | 0.590 $\pm$ 0.036 | 6.18E-03 $\pm$ 8.50E-04 | 22.1 $\pm$ 0.6 | 0.404 $\pm$ 0.040 |
| SUPER-RESOLUTION | ΠGDM-D | 106.8 | 0.592 $\pm$ 0.041 | 7.92E-03 $\pm$ 1.07E-03 | 21.0 $\pm$ 0.6 | 0.353 $\pm$ 0.042 |
| $\sigma_y = 0.1$ | DTMPD-D | **32.7** | **0.310** $\pm$ 0.053 | 3.99E-03 $\pm$ 1.84E-03 | **24.3** $\pm$ 1.7 | **0.679** $\pm$ 0.071 |
| 4× 'BICUBIC' | DPS-D | 114.0 | 0.569 $\pm$ 0.044 | 8.47E-03 $\pm$ 4.95E-03 | 21.1 $\pm$ 1.7 | 0.483 $\pm$ 0.046 |
| SUPER-RESOLUTION | ΠGDM-D | 206.0 | 0.724 $\pm$ 0.034 | 2.46E-02 $\pm$ 2.16E-03 | 16.1 $\pm$ 0.4 | 0.176 $\pm$ 0.034 |
| $\sigma_y = 0.01$ | DTMPD-D | 30.2 | 0.114 $\pm$ 0.029 | 2.84E-03 $\pm$ 2.56E-03 | 26.5 $\pm$ 2.8 | 0.907 $\pm$ 0.020 |
| 'BOX' MASK | DPS-D | **23.9** | **0.093** $\pm$ 0.019 | 2.53E-03 $\pm$ 1.77E-03 | 26.9 $\pm$ 2.8 | 0.899 $\pm$ 0.017 |
| INPAINTING | ΠGDM-D | 27.1 | 0.108 $\pm$ 0.025 | 2.50E-03 $\pm$ 1.55E-03 | 26.7 $\pm$ 2.3 | 0.877 $\pm$ 0.015 |
| $\sigma_y = 0.05$ | DTMPD-D | **33.6** | **0.186** $\pm$ 0.036 | 3.24E-03 $\pm$ 3.19E-03 | 25.8 $\pm$ 2.5 | **0.847** $\pm$ 0.030 |
| 'BOX' MASK | DPS-D | 39.7 | 0.318 $\pm$ 0.044 | 3.81E-03 $\pm$ 1.92E-03 | 24.7 $\pm$ 2.0 | 0.677 $\pm$ 0.086 |
| INPAINTING | ΠGDM-D | 49.5 | 0.354 $\pm$ 0.044 | 4.67E-03 $\pm$ 1.50E-03 | 23.5 $\pm$ 1.3 | 0.555 $\pm$ 0.091 |
| $\sigma_y = 0.1$ | DTMPD-D | **34.0** | **0.223** $\pm$ 0.041 | 3.42E-03 $\pm$ 2.24E-03 | **25.3** $\pm$ 2.3 | **0.801** $\pm$ 0.045 |
| 'BOX' MASK | DPS-D | 59.1 | 0.467 $\pm$ 0.053 | 7.78E-03 $\pm$ 3.19E-03 | 21.4 $\pm$ 1.7 | 0.476 $\pm$ 0.110 |
| INPAINTING | ΠGDM-D | 72.6 | 0.529 $\pm$ 0.047 | 9.88E-03 $\pm$ 2.48E-03 | 20.2 $\pm$ 1.2 | 0.356 $\pm$ 0.118 |

**CIFAR10 64×64** For VE-SDE, our CIFAR-10 64×64 experiments compare TMPD to DPS and ΠGDM, and also compare diffusion methods (DDPM) in Table 12 and score-based methods (discretized with Euler-Maruyama) in Table 13. Various samples used to produce the figures in Tables 12 and 13 are shown in Fig 16 and 17.

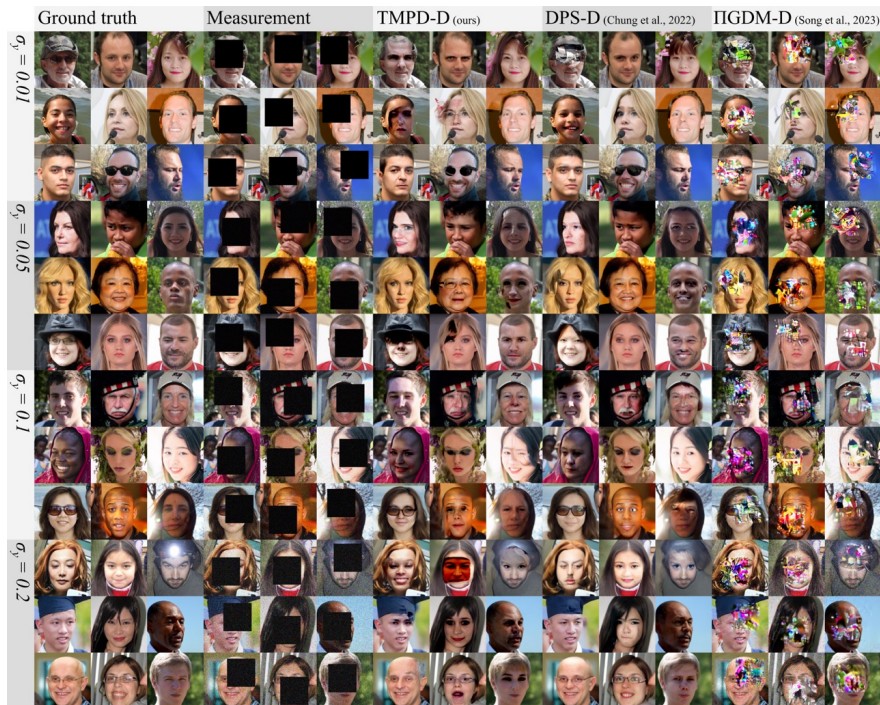

Figure 8: 'Box' mask inpainting samples from the VP-SDE on FFHQ across a range of observation noise.

Table 11: Quantitative evaluation of solving linear inverse problems for VE DDPM for 100 steps with increasing noise on FFHQ 256×256-1k validation dataset.

| PROBLEM | METHOD | FID ↓ | LPIPS ↓ | MSE ↓ | PSNR ↑ | SSIM ↑ |
|---|---|---|---|---|---|---|
| $\sigma_y = 0.01$ | DTMPD-D | **38.8** | **0.215** $_{\pm 0.042}$ | 0.001 $_{\pm 0.001}$ | **28.9** $_{\pm 2.2}$ | **0.821** $_{\pm 0.051}$ |
| 4× 'BICUBIC' SUPER-RESOLUTION | ΠGDM-D | 46.1 | 0.252 $_{\pm 0.042}$ | 0.002 $_{\pm 0.001}$ | 27.4 $_{\pm 1.9}$ | 0.788 $_{\pm 0.053}$ |
| $\sigma_y = 0.05$ | DTMPD-D | **39.2** | **0.279** $_{\pm 0.049}$ | 0.003 $_{\pm 0.001}$ | **26.2** $_{\pm 1.8}$ | **0.741** $_{\pm 0.065}$ |
| 4× 'BICUBIC' SUPER-RESOLUTION | ΠGDM-D | 43.3 | 0.302 $_{\pm 0.049}$ | 0.003 $_{\pm 0.001}$ | 25.1 $_{\pm 1.7}$ | 0.722 $_{\pm 0.067}$ |
| $\sigma_y = 0.1$ | DTMPD-D | **41.1** | **0.317** $_{\pm 0.054}$ | 0.004 $_{\pm 0.002}$ | **24.5** $_{\pm 1.7}$ | **0.693** $_{\pm 0.073}$ |
| 4× 'BICUBIC' SUPER-RESOLUTION | ΠGDM-D | 44.7 | 0.344 $_{\pm 0.053}$ | 0.005 $_{\pm 0.002}$ | 23.4 $_{\pm 1.6}$ | 0.672 $_{\pm 0.070}$ |
| $\sigma_y = 0.01$ | DTMPD-D | 31.4 | 0.124 $_{\pm 0.030}$ | 0.003 $_{\pm 0.003}$ | 26.5 $_{\pm 3.0}$ | 0.906 $_{\pm 0.020}$ |
| 'BOX' MASK INPAINTING | ΠGDM-D | **23.7** | **0.069** $_{\pm 0.021}$ | 0.002 $_{\pm 0.001}$ | **28.0** $_{\pm 2.6}$ | **0.925** $_{\pm 0.016}$ |
| $\sigma_y = 0.05$ | DTMPD-D | 38.2 | 0.199 $_{\pm 0.039}$ | 0.003 $_{\pm 0.003}$ | 26.0 $_{\pm 2.5}$ | **0.848** $_{\pm 0.034}$ |
| 'BOX' MASK INPAINTING | ΠGDM-D | **29.8** | **0.185** $_{\pm 0.028}$ | 0.002 $_{\pm 0.001}$ | **26.7** $_{\pm 2.1}$ | 0.825 $_{\pm 0.029}$ |
| $\sigma_y = 0.1$ | DTMPD-D | **39.7** | **0.241** $_{\pm 0.046}$ | 0.004 $_{\pm 0.003}$ | **25.2** $_{\pm 2.4}$ | **0.799** $_{\pm 0.048}$ |
| 'BOX' MASK INPAINTING | ΠGDM-D | 42.1 | 0.279 $_{\pm 0.040}$ | 0.003 $_{\pm 0.001}$ | 25.1 $_{\pm 1.6}$ | 0.716 $_{\pm 0.052}$ |

For ΠGDM(-D), we are able to use the hyperparameter $r_t^2 = v_t/(1 + v_t)$ as suggested by Song et al. (2023), which is calculated by assuming the data distribution $p_0(\mathbf{x}_0)$ is a standard normal and then calculating the posterior variance, for both VE DDIM and VE-SDE methods, but note some instability for the ΠGDM VE-SDE method for small noise, as shown in Fig 10.

For DDPM we use the step-size constant suggested in Chung et al. (2022a) for inpainting, $\zeta_i = \zeta'/\|\mathbf{y} - \mathbf{Hm}_{0|t}\|$, where we tune $\zeta'$ over the suggested range of $\zeta' \in [0.1, 1.0]$ in Chung et al. (2022a) across LPIPS, MSE, PSNR and SSIM, as shown in Fig. 9 for each inverse problem (each line in the Tables 12 and 13).

Table 12: Noisy observation inpainting and super-resolution for VE DDPM on CIFAR-10 1k validation set.

| Problem | Method | FID ↓ | LPIPS ↓ | MSE ↓ | PSNR ↑ | SSIM ↑ |
|---|---|---|---|---|---|---|
| $\sigma_y = 0.01$ | DTMPD-D | **34.1** | $0.094_{\pm 0.044}$ | $0.006_{\pm 0.005}$ | $23.6_{\pm 3.3}$ | $0.782_{\pm 0.067}$ |
| 'BOX' MASK | DPS-D | 40.6 | $0.085_{\pm 0.036}$ | $0.005_{\pm 0.004}$ | $24.2_{\pm 3.1}$ | $0.802_{\pm 0.064}$ |
| INPAINTING | ΠGDM-D | 40.0 | $0.076_{\pm 0.035}$ | $0.004_{\pm 0.003}$ | $25.0_{\pm 3.3}$ | $0.824_{\pm 0.066}$ |
| $\sigma_y = 0.01$ | DTMPD-D | **39.0** | $0.271_{\pm 0.069}$ | $0.028_{\pm 0.020}$ | $16.5_{\pm 3.1}$ | $0.582_{\pm 0.072}$ |
| 'HALF' MASK | DPS-D | 42.8 | $0.247_{\pm 0.065}$ | $0.029_{\pm 0.020}$ | $16.5_{\pm 3.2}$ | $0.595_{\pm 0.077}$ |
| INPAINTING | ΠGDM-D | 42.2 | $0.237_{\pm 0.067}$ | $0.029_{\pm 0.020}$ | $16.4_{\pm 3.1}$ | $0.610_{\pm 0.079}$ |
| $\sigma_y = 0.05$ | DTMPD-D | **36.7** | $\mathbf{0.163}_{\pm 0.070}$ | $0.007_{\pm 0.011}$ | $22.6_{\pm 2.8}$ | $0.724_{\pm 0.082}$ |
| 'BOX' MASK | DPS-D | 101.5 | $0.224_{\pm 0.077}$ | $0.008_{\pm 0.004}$ | $21.8_{\pm 2.1}$ | $0.681_{\pm 0.087}$ |
| INPAINTING | ΠGDM-D | 99.9 | $0.225_{\pm 0.076}$ | $0.007_{\pm 0.003}$ | $21.8_{\pm 1.7}$ | $0.682_{\pm 0.093}$ |
| $\sigma_y = 0.05$ | DTMPD-D | **40.9** | $0.320_{\pm 0.078}$ | $0.029_{\pm 0.020}$ | $16.4_{\pm 2.9}$ | $0.544_{\pm 0.073}$ |
| 'HALF' MASK | DPS-D | 105.5 | $0.362_{\pm 0.078}$ | $0.032_{\pm 0.021}$ | $15.7_{\pm 2.7}$ | $0.500_{\pm 0.075}$ |
| INPAINTING | ΠGDM-D | 97.1 | $0.366_{\pm 0.079}$ | $0.033_{\pm 0.020}$ | $15.4_{\pm 2.5}$ | $0.503_{\pm 0.079}$ |
| $\sigma_y = 0.1$ | DTMPD-D | **38.1** | $\mathbf{0.233}_{\pm 0.086}$ | $0.009_{\pm 0.009}$ | $21.4_{\pm 2.5}$ | $\mathbf{0.652}_{\pm 0.102}$ |
| 'BOX' MASK | DPS-D | 80.4 | $0.313_{\pm 0.078}$ | $0.012_{\pm 0.006}$ | $19.7_{\pm 2.1}$ | $0.596_{\pm 0.088}$ |
| INPAINTING | ΠGDM-D | 140.5 | $0.378_{\pm 0.086}$ | $0.013_{\pm 0.003}$ | $18.8_{\pm 0.9}$ | $0.550_{\pm 0.113}$ |
| $\sigma_y = 0.1$ | DTMPD-D | **40.8** | $0.359_{\pm 0.086}$ | $0.030_{\pm 0.019}$ | $16.1_{\pm 2.8}$ | $\mathbf{0.495}_{\pm 0.087}$ |
| 'HALF' MASK | DPS-D | 99.5 | $0.434_{\pm 0.072}$ | $0.038_{\pm 0.023}$ | $14.8_{\pm 2.4}$ | $0.400_{\pm 0.083}$ |
| INPAINTING | ΠGDM-D | 138.8 | $0.476_{\pm 0.078}$ | $0.039_{\pm 0.019}$ | $14.5_{\pm 1.9}$ | $0.402_{\pm 0.095}$ |
| $\sigma_y = 0.01$ | DTMPD-D | **32.7** | $0.126_{\pm 0.058}$ | $0.004_{\pm 0.003}$ | $24.4_{\pm 2.9}$ | $0.828_{\pm 0.073}$ |
| 2× 'NEAREST' | DPS-D | 42.3 | $0.134_{\pm 0.053}$ | $0.004_{\pm 0.002}$ | $24.8_{\pm 2.5}$ | $0.839_{\pm 0.063}$ |
| SUPER-RESOLUTION | ΠGDM-D | 34.9 | $0.110_{\pm 0.042}$ | $0.004_{\pm 0.002}$ | $24.8_{\pm 2.8}$ | $0.839_{\pm 0.066}$ |
| $\sigma_y = 0.01$ | DTMPD-D | 38.6 | $0.295_{\pm 0.082}$ | $0.011_{\pm 0.006}$ | $20.2_{\pm 2.4}$ | $0.544_{\pm 0.109}$ |
| 4× 'BICUBIC' | DPS-D | 53.4 | $0.296_{\pm 0.074}$ | $0.011_{\pm 0.006}$ | $20.1_{\pm 2.3}$ | $0.547_{\pm 0.101}$ |
| SUPER-RESOLUTION | ΠGDM-D | **37.8** | $0.250_{\pm 0.077}$ | $0.011_{\pm 0.006}$ | $20.4_{\pm 2.5}$ | $0.577_{\pm 0.119}$ |
| $\sigma_y = 0.05$ | DTMPD-D | **34.0** | $\mathbf{0.204}_{\pm 0.082}$ | $0.006_{\pm 0.004}$ | $23.1_{\pm 2.3}$ | $0.762_{\pm 0.089}$ |
| 2× 'NEAREST' | DPS-D | 98.8 | $0.283_{\pm 0.077}$ | $0.007_{\pm 0.003}$ | $21.9_{\pm 1.6}$ | $0.711_{\pm 0.081}$ |
| SUPER-RESOLUTION | ΠGDM-D | 99.6 | $0.298_{\pm 0.081}$ | $0.008_{\pm 0.003}$ | $21.1_{\pm 1.4}$ | $0.677_{\pm 0.092}$ |
| $\sigma_y = 0.05$ | DTMPD-D | **36.8** | $0.380_{\pm 0.097}$ | $0.015_{\pm 0.006}$ | $18.8_{\pm 2.0}$ | $0.440_{\pm 0.113}$ |
| 4× 'BICUBIC' | DPS-D | 126.1 | $0.467_{\pm 0.072}$ | $0.017_{\pm 0.007}$ | $18.2_{\pm 1.7}$ | $0.434_{\pm 0.096}$ |
| SUPER-RESOLUTION | ΠGDM-D | 104.2 | $0.419_{\pm 0.088}$ | $0.016_{\pm 0.006}$ | $18.1_{\pm 1.6}$ | $0.454_{\pm 0.117}$ |
| $\sigma_y = 0.1$ | DTMPD-D | **37.1** | $\mathbf{0.285}_{\pm 0.100}$ | $0.008_{\pm 0.006}$ | $\mathbf{21.5}_{\pm 2.1}$ | $0.657_{\pm 0.112}$ |
| 2× 'NEAREST' | DPS-D | 144.6 | $0.421_{\pm 0.082}$ | $0.012_{\pm 0.003}$ | $19.4_{\pm 1.1}$ | $0.571_{\pm 0.101}$ |
| SUPER-RESOLUTION | ΠGDM-D | 175.8 | $0.456_{\pm 0.085}$ | $0.016_{\pm 0.003}$ | $17.9_{\pm 0.9}$ | $0.514_{\pm 0.115}$ |
| $\sigma_y = 0.1$ | DTMPD-D | **36.1** | $\mathbf{0.439}_{\pm 0.099}$ | $0.019_{\pm 0.008}$ | $17.5_{\pm 1.9}$ | $0.353_{\pm 0.115}$ |
| 4× 'BICUBIC' | DPS-D | 155.1 | $0.535_{\pm 0.067}$ | $0.022_{\pm 0.007}$ | $16.8_{\pm 1.3}$ | $0.360_{\pm 0.092}$ |
| SUPER-RESOLUTION | ΠGDM-D | 196.2 | $0.539_{\pm 0.078}$ | $0.030_{\pm 0.007}$ | $15.3_{\pm 1.1}$ | $0.327_{\pm 0.106}$ |

Table 13: Noisy observation inpainting and super-resolution for the reverse VE-SDEs on CIFAR-10 1k validation set.

| PROBLEM | METHOD | FID ↓ | LPIPS ↓ | MSE ↓ | PSNR ↑ | SSIM ↑ |
|---|---|---|---|---|---|---|
| $\sigma_y = 0.01$ | TMPD | **40.0** | $0.102_{\pm 0.047}$ | $0.005_{\pm 0.004}$ | $23.7_{\pm 3.1}$ | $0.773_{\pm 0.069}$ |
| 'BOX' MASK | DPS | 103.6 | $0.637_{\pm 0.074}$ | $0.114_{\pm 0.052}$ | $9.9_{\pm 1.9}$ | $0.050_{\pm 0.068}$ |
| INPAINTING | ΠGDM | 78.9 | $0.094_{\pm 0.039}$ | $0.005_{\pm 0.004}$ | $24.1_{\pm 3.1}$ | $0.787_{\pm 0.071}$ |
| $\sigma_y = 0.01$ | TMPD | **45.8** | $0.279_{\pm 0.066}$ | $0.030_{\pm 0.029}$ | $16.3_{\pm 3.0}$ | $0.569_{\pm 0.069}$ |
| 'HALF' MASK | DPS | 110.8 | $0.638_{\pm 0.072}$ | $0.117_{\pm 0.056}$ | $9.8_{\pm 2.0}$ | $0.045_{\pm 0.067}$ |
| INPAINTING | ΠGDM | 50.5 | $0.264_{\pm 0.068}$ | $0.027_{\pm 0.020}$ | $16.7_{\pm 3.1}$ | $0.585_{\pm 0.077}$ |
| $\sigma_y = 0.05$ | TMPD | **45.3** | $0.167_{\pm 0.065}$ | $0.007_{\pm 0.024}$ | $22.6_{\pm 2.7}$ | $0.718_{\pm 0.078}$ |
| 'BOX' MASK | DPS | 103.4 | $0.638_{\pm 0.070}$ | $0.115_{\pm 0.055}$ | $9.8_{\pm 1.9}$ | $0.047_{\pm 0.067}$ |
| INPAINTING | ΠGDM | 81.9 | $0.160_{\pm 0.056}$ | $0.006_{\pm 0.004}$ | $22.8_{\pm 2.4}$ | $0.720_{\pm 0.078}$ |
| $\sigma_y = 0.05$ | TMPD | **51.1** | $0.319_{\pm 0.071}$ | $0.029_{\pm 0.019}$ | $16.2_{\pm 2.8}$ | $0.532_{\pm 0.071}$ |
| 'HALF' MASK | DPS | 109.3 | $0.638_{\pm 0.070}$ | $0.116_{\pm 0.057}$ | $9.8_{\pm 2.0}$ | $0.044_{\pm 0.067}$ |
| INPAINTING | ΠGDM | 56.0 | $0.311_{\pm 0.074}$ | $0.029_{\pm 0.021}$ | $16.4_{\pm 3.0}$ | $0.540_{\pm 0.078}$ |
| $\sigma_y = 0.1$ | TMPD | **47.6** | $0.231_{\pm 0.079}$ | $0.008_{\pm 0.005}$ | $21.5_{\pm 2.3}$ | $0.650_{\pm 0.097}$ |
| 'BOX' MASK | DPS | 104.3 | $0.639_{\pm 0.068}$ | $0.113_{\pm 0.049}$ | $9.9_{\pm 1.9}$ | $0.048_{\pm 0.065}$ |
| INPAINTING | ΠGDM | 84.6 | $0.220_{\pm 0.079}$ | $0.008_{\pm 0.004}$ | $21.6_{\pm 2.1}$ | $0.665_{\pm 0.105}$ |
| $\sigma_y = 0.1$ | TMPD | **54.1** | $0.366_{\pm 0.080}$ | $0.032_{\pm 0.020}$ | $15.8_{\pm 2.7}$ | $0.480_{\pm 0.085}$ |
| 'HALF' MASK | DPS | 110.4 | $0.639_{\pm 0.068}$ | $0.117_{\pm 0.054}$ | $9.7_{\pm 2.0}$ | $0.046_{\pm 0.067}$ |
| INPAINTING | ΠGDM | 58.5 | $0.349_{\pm 0.081}$ | $0.031_{\pm 0.021}$ | $16.1_{\pm 2.9}$ | $0.499_{\pm 0.090}$ |
| $\sigma_y = 0.01$ | TMPD | **43.5** | $\mathbf{0.141}_{\pm 0.070}$ | $0.007_{\pm 0.034}$ | $24.1_{\pm 3.1}$ | $\mathbf{0.810}_{\pm 0.092}$ |
| 2× 'NEAREST' | DPS | 118.7 | $0.641_{\pm 0.066}$ | $0.117_{\pm 0.054}$ | $9.7_{\pm 1.9}$ | $0.048_{\pm 0.065}$ |
| SUPER-RESOLUTION | ΠGDM | 59.4 | $0.258_{\pm 0.083}$ | $0.007_{\pm 0.003}$ | $22.0_{\pm 2.2}$ | $0.716_{\pm 0.075}$ |
| $\sigma_y = 0.01$ | TMPD | **49.6** | $\mathbf{0.439}_{\pm 0.093}$ | $\mathbf{0.020}_{\pm 0.008}$ | $\mathbf{17.3}_{\pm 1.8}$ | $\mathbf{0.345}_{\pm 0.115}$ |
| 4× 'BICUBIC' | DPS | 128.3 | $0.642_{\pm 0.071}$ | $0.122_{\pm 0.059}$ | $9.6_{\pm 2.0}$ | $0.046_{\pm 0.068}$ |
| SUPER-RESOLUTION | ΠGDM | 52.9 | $0.561_{\pm 0.083}$ | $0.042_{\pm 0.015}$ | $14.1_{\pm 1.7}$ | $0.167_{\pm 0.085}$ |
| $\sigma_y = 0.05$ | TMPD | **47.5** | $\mathbf{0.213}_{\pm 0.087}$ | $0.007_{\pm 0.012}$ | $\mathbf{22.8}_{\pm 2.5}$ | $\mathbf{0.744}_{\pm 0.100}$ |
| 2× 'NEAREST' | DPS | 119.1 | $0.638_{\pm 0.070}$ | $0.118_{\pm 0.059}$ | $9.7_{\pm 2.0}$ | $0.049_{\pm 0.068}$ |
| SUPER-RESOLUTION | ΠGDM | 60.0 | $0.326_{\pm 0.097}$ | $0.009_{\pm 0.004}$ | $20.7_{\pm 1.9}$ | $0.619_{\pm 0.104}$ |
| $\sigma_y = 0.05$ | TMPD | **51.2** | $\mathbf{0.379}_{\pm 0.094}$ | $\mathbf{0.016}_{\pm 0.029}$ | $\mathbf{18.6}_{\pm 2.1}$ | $\mathbf{0.428}_{\pm 0.117}$ |
| 4× 'BICUBIC' | DPS | 128.0 | $0.642_{\pm 0.068}$ | $0.122_{\pm 0.061}$ | $9.6_{\pm 2.0}$ | $0.043_{\pm 0.067}$ |
| SUPER-RESOLUTION | ΠGDM | 53.8 | $0.548_{\pm 0.084}$ | $0.038_{\pm 0.014}$ | $14.5_{\pm 1.7}$ | $0.186_{\pm 0.087}$ |
| $\sigma_y = 0.1$ | TMPD | **51.6** | $\mathbf{0.292}_{\pm 0.097}$ | $0.009_{\pm 0.020}$ | $\mathbf{21.3}_{\pm 2.2}$ | $\mathbf{0.646}_{\pm 0.116}$ |
| 2× 'NEAREST' | DPS | 120.5 | $0.644_{\pm 0.074}$ | $0.121_{\pm 0.061}$ | $9.7_{\pm 2.0}$ | $0.048_{\pm 0.068}$ |
| SUPER-RESOLUTION | ΠGDM | 61.9 | $0.386_{\pm 0.100}$ | $0.012_{\pm 0.005}$ | $19.5_{\pm 1.9}$ | $0.523_{\pm 0.117}$ |
| $\sigma_y = 0.1$ | TMPD | **49.6** | $\mathbf{0.439}_{\pm 0.093}$ | $\mathbf{0.020}_{\pm 0.008}$ | $\mathbf{17.3}_{\pm 1.8}$ | $\mathbf{0.345}_{\pm 0.115}$ |
| 4× 'BICUBIC' | DPS | 128.3 | $0.642_{\pm 0.071}$ | $0.122_{\pm 0.059}$ | $9.6_{\pm 2.0}$ | $0.046_{\pm 0.068}$ |
| SUPER-RESOLUTION | ΠGDM | 52.9 | $0.561_{\pm 0.083}$ | $0.042_{\pm 0.015}$ | $14.1_{\pm 1.7}$ | $0.167_{\pm 0.085}$ |

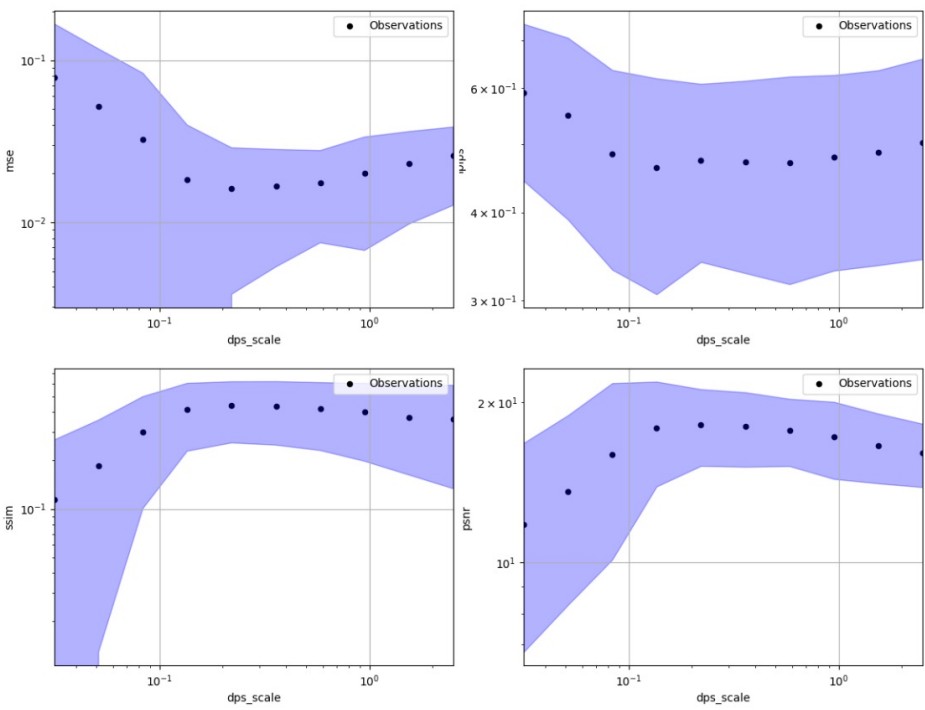

Figure 9: DPS scale hyperparameter search across LPIPS, MSE, PSNR and SSIM for CIFAR-10 $4\times$ bicubic interpolation super-resolution with $\sigma = 0.05$. Plotted are the mean values $\pm$ 2 standard deviations over a 128 sample/batch size, calculated over 10 values of $\zeta'$. We chose an optimal value of 0.15 for the DPS scale hyperparameter in this case.

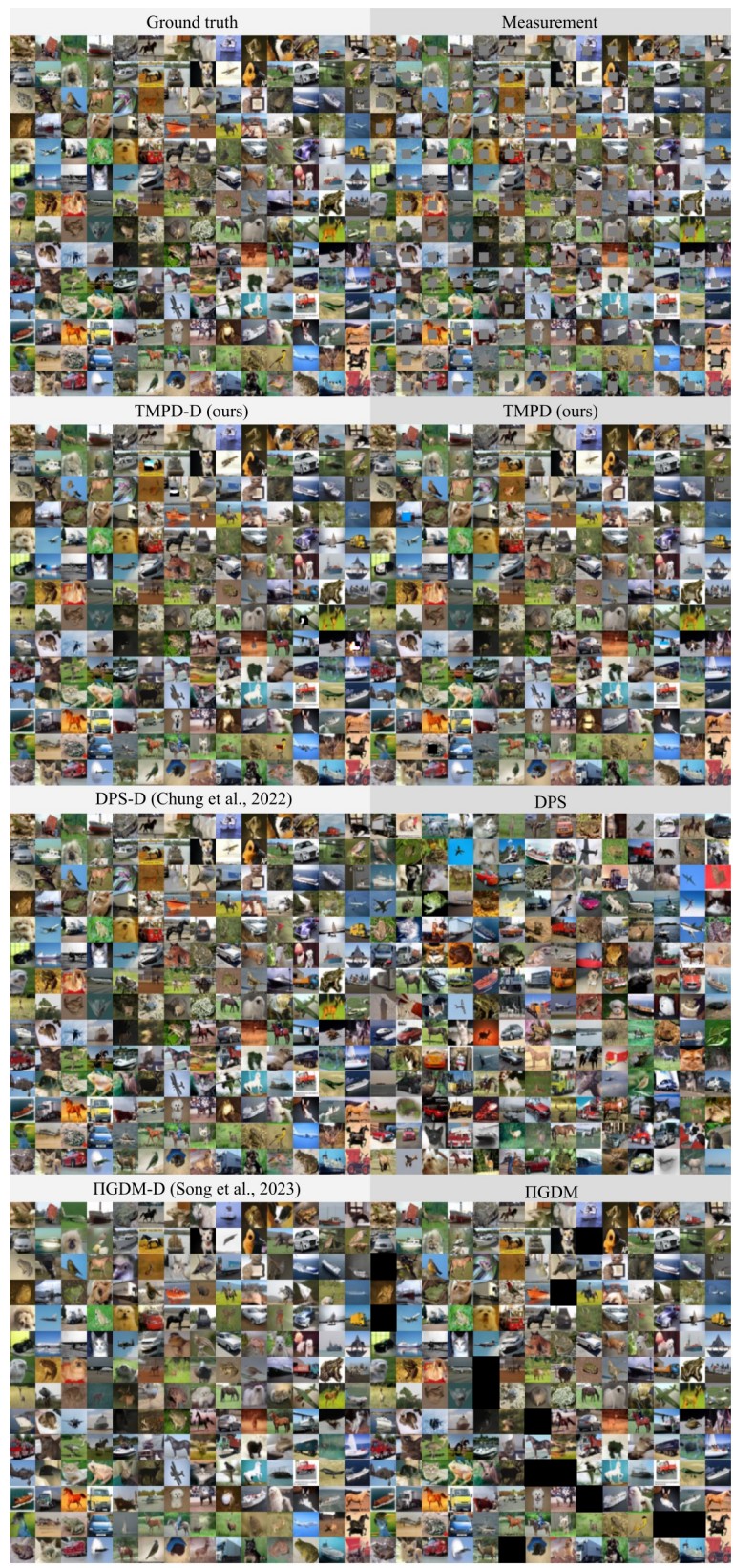

Figure 10: Inpainting samples from the VP-SDE on CIFAR-10. The observation model was 'box' mask with Gaussian ($\sigma_y = 0.05$) noise.

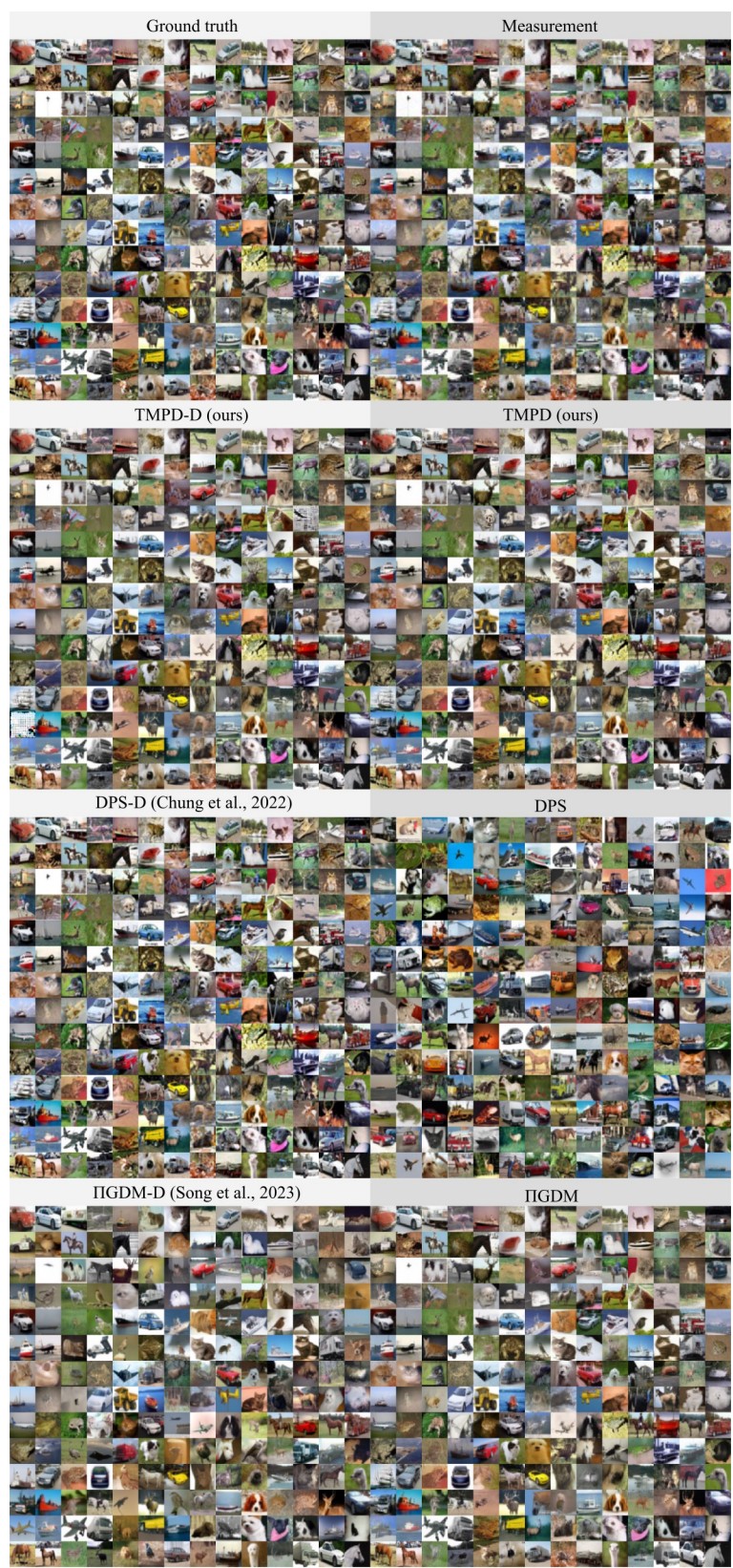

Figure 11: 2× nearest-neighbour super-resolution samples from the VP-SDE on CIFAR-10. The observation model was Gaussian ($\sigma_{\mathrm{y}} = 0.05$) noise.

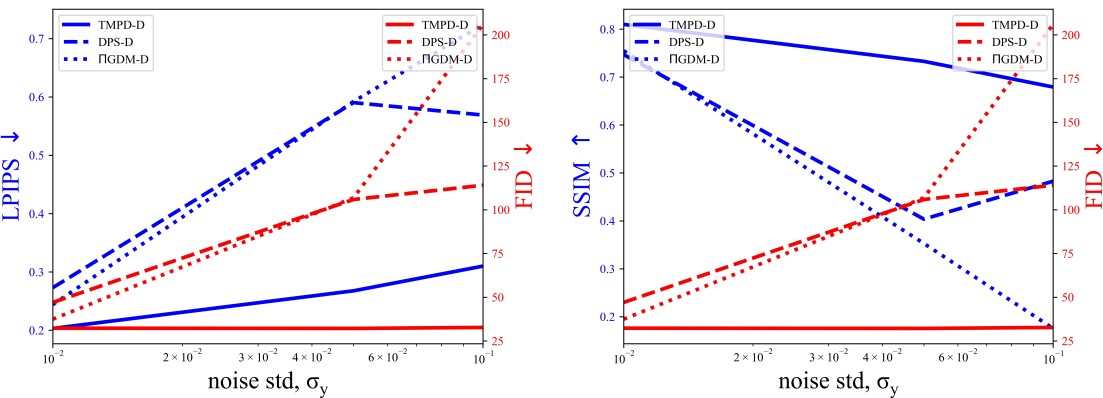

Figure 12: $4\times$ bicubic super-resolution FID vs LPIPS (left) and SSIM (right) using the VE-SDE on FFHQ-1k validation dataset for increasing observation noise.

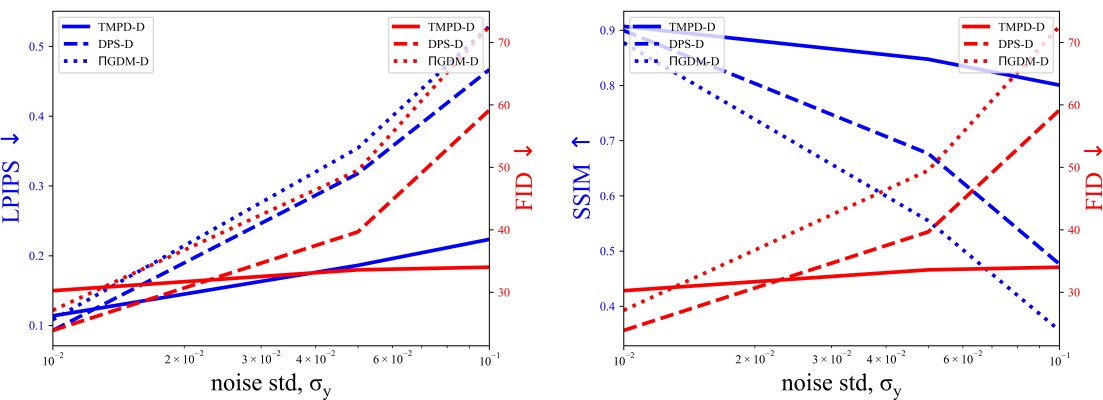

Figure 13: 'box' mask inpainting FID vs LPIPS (left) and SSIM (right) using the VE-SDE on FFHQ-1k validation dataset for increasing observation noise.

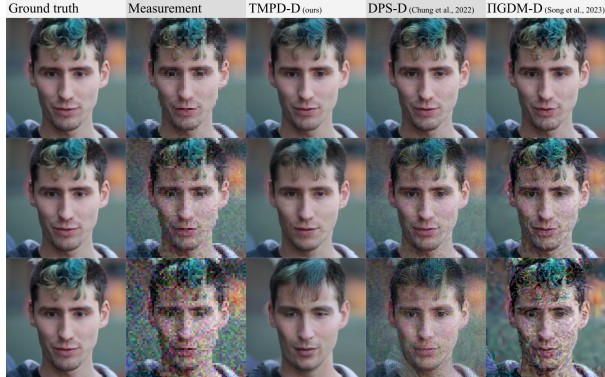

Figure 14: Whilst DTMPD-D with the VE-SDE remains robust to increasing noise, DPS-D and ΠGDM-D do not. This is illustrated by samples from the different guidance methods for a $4\times$ super-resolution $(256 \times 256 \rightarrow 64 \times 64)$ problem with increasing Gaussian observation noise distorting a ground truth image. The top row measurement has $\sigma_y = 0.01$, the middle row $\sigma_y = 0.05$ and the bottom row $\sigma_y = 0.1$. For the full results, see Table 10.

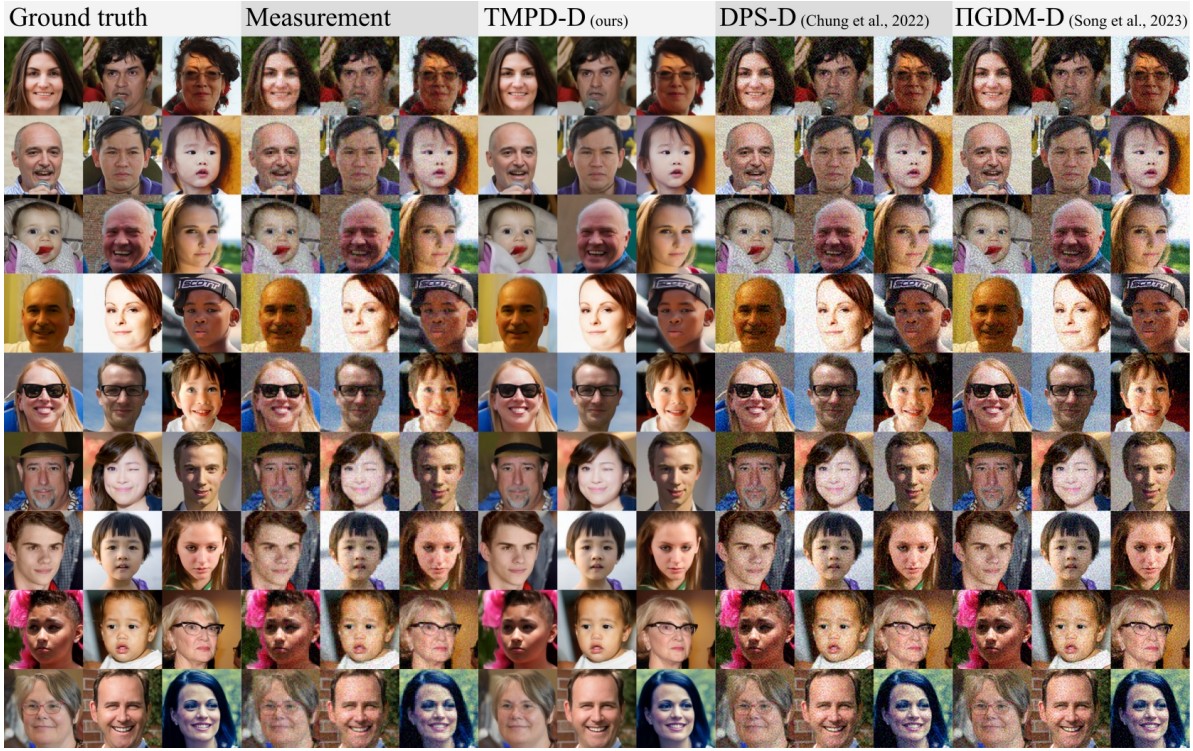

Figure 15: $4\times$ bicubic super-resolution samples from the VE-SDE on FFHQ. The observation model was Gaussian ($\sigma_y = 0.05$) noise.

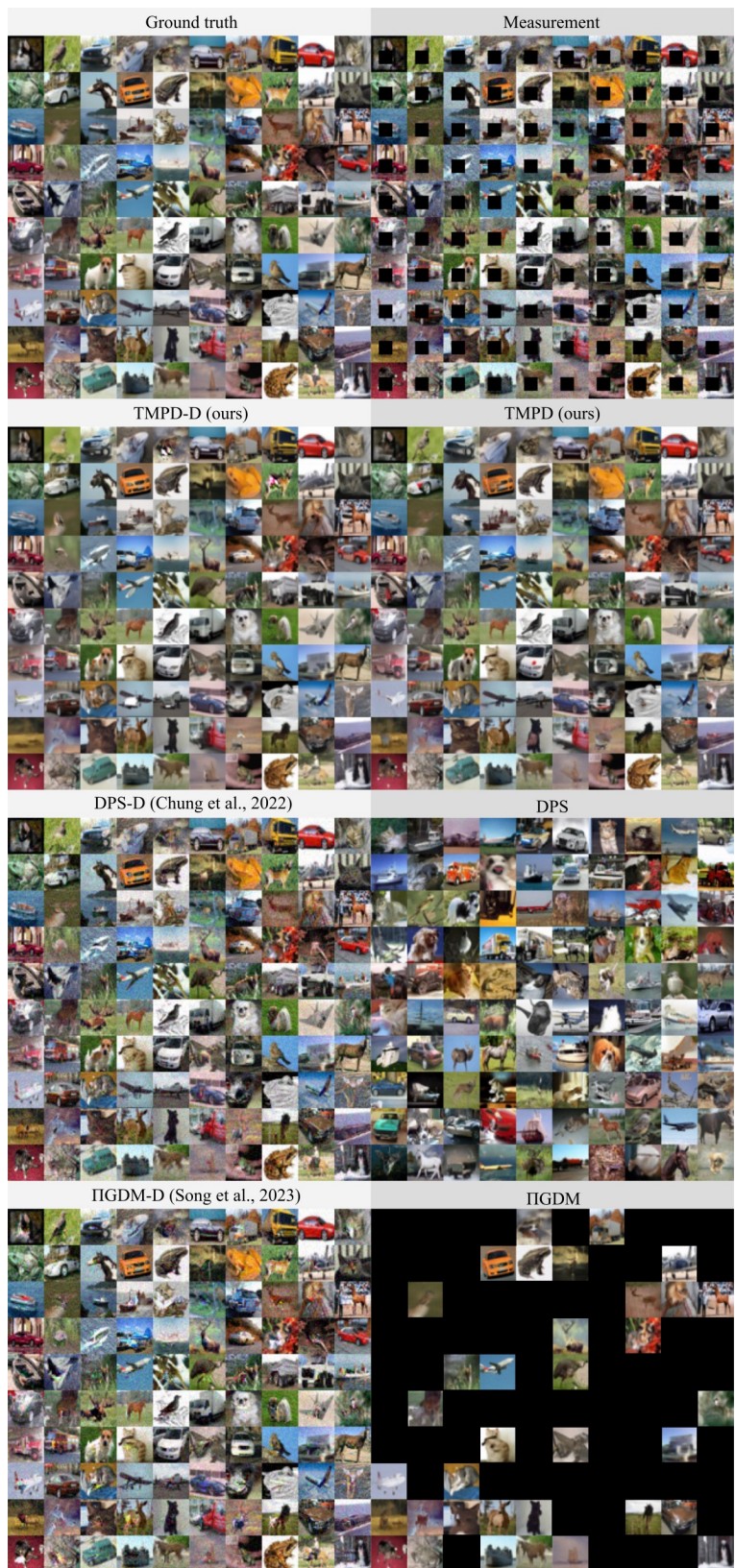

Figure 16: Inpainting samples from the VE-SDE on CIFAR-10. The observation model was 'box' mask with Gaussian ($\sigma_y = 0.05$) noise.

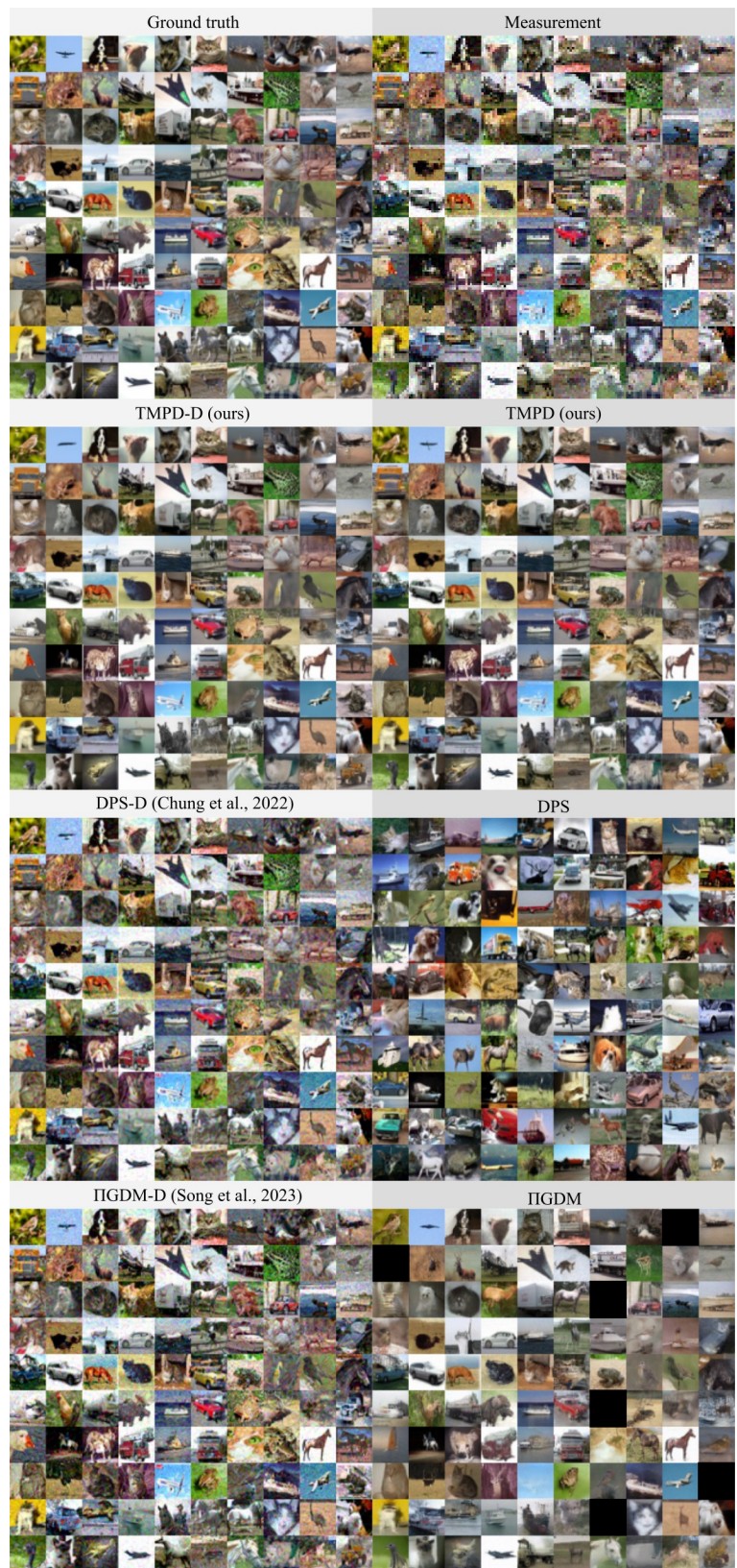

Figure 17: 2× nearest-neighbour super-resolution samples from the VE-SDE on CIFAR-10. The observation model was Gaussian ($\sigma_{\mathrm{y}} = 0.05$) noise.

