# OpenReview forum: "Tweedie Moment Projected Diffusions for Inverse Problems"
_TMLR — Accepted by TMLR_

### Review · Reviewer_uMGN · 2024-08-07

**Summary Of Contributions:**

The submission focuses on solving inverse problems with diffusion models, where computing the conditional score $\nabla_{x_t} \log p_t(x_t | y)$ in the reverse SDE is, in general, intractable. Building upon previous works, the submission proposes to also match the second moment of the Tweedie estimate in order to resolve known issues of hyperparameter tuning of existing alternatives (this method is called TMPD).

The submission discusses the computational complexity of this algorithm, its practical implementation and limitations, and theoretical results on when TMPD provides good approximations to the true posterior distribution.

Experiments are conducted on both synthetic data and two real-world image datasets, where TMPD is shown to outperform two commonly used methods: DPS and PGDM (read $\Pi$GDM).

**Audience:**

Yes

**Claims And Evidence:**

Yes

**Requested Changes:**

**Equation 9**

I would suggest to expand the discussion paragraph after the equation. In particular, all approximations used to obtain Eq. (9) should be clearly stated and made precise.

**Approximation based on properties of $H$**

Could the authors expand on what other strategies could be employed in a general case where the operator $H$ may not be sparse?

**Proposition 3**

Could the authors expand on their claim "if there are no errors in the initial condition". What does this mean in practice? Is this equivalent to saying that the measurement should be noiseless?

**Theorem 1**

- The symbol $\Phi$ is never defined, what is it referring to?
- All equations should be numbered, including within Theorems.
- I am not sure I fully understand the claim in the Theorem: is it bounding the total variation at the **end** of the reverse process (i.e., $t=T$), or **at any** time $t$? The text reads "where $q_t$", but the equation contains $q_T$ and $T$, not $t$.
- I am not sure I fully understand the relation between $t$ and $T$ in the bound. Could the authors expand on an intuitive discussion of the claim in the Theorem? For example, what happens at the same time $t$ of the reverse process when starting from different $T$'s?

**Remark 1**

In its current formulation, this is too strong of a claim. If the extensions are trivial, they should be included in the current manuscript. If not, more evidence needs to support the claim that all of these (ambitious) extensions can be attained.

**Paragraph after Remark 1**

- Clarify what the "limit" being taken is.
- I am confused by the next sentence. "The $\Phi$ ... converges to 1 ... and therefore $M \to 1$". But $\exp(1) = e > 1$, what am I misunderstanding here? Should it be "the $\Phi$ in Eq. (12)" converges to 0?

**Related work**

> The mean is chosen the same way as our method.

This sentence should be rephrased, as DPS introduced ideas of using Tweedie's estimate in inverse problems before the current submission.

**GMM experiment**

In the text, it is stated a few times "non-linear SDE" and "non-linear problems in high dimension", but I thought the submission was about linear inverse problems? In the Introduction, "... a known linear observation map $H$". What non-linearity are the authors referring to here?

**Comparison with SNIPS**

Continuing on my previous point of confusion, if this submission is only about linear inverse problems, an important comparison with SNIPS (Kawar et al., 2021) is missing. This comparison is important because SNIPS introduced a closed-form solution and efficient approximation of the **exact** reverse conditional likelihood for inverse linear problems (if the SVD of $H$ can be computed).

Could the authors expand on their motivation not to compare with SNIPS and the advantages/limitations of TMPD compared to SNIPS? For example, could ideas of TMPD be extended to non-linear measurement models? This point should be made clearer, especially given the significant computational limitations of the submission (e.g., image results are shown on 32x32 pixels and 256x256 pixels images, whereas problems of practical relevance can be of the order of 1024x1024 pixels).

---

**Minor comments**

- Typo in "reasonance" imaging
- Typo in "is devoted developing"
- Sec. 2, Technical background: "In the SDE paradigm ... ". So far, the paradigms have been introduced as "SGM" and "DDPM". But "SDE" is only introduced afterwards "In SGMs, a stochastic different equation (SDE) ...". I would flip the order to clarify which paradigm uses SDEs.
- Please conform cross reference formats: references to equations should be capitalized and have numbers in parentheses (e.g., "equation 2" -> "Equation (2)").
- Please rephrase "Optimally, we would like to sample from the reverse ... to the forward SDE started in $p_{\text{data}}(\cdot | y)$. This sentence reads as if the forward SDE is started at the measurement $y$, whereas its the reverse.
- Typo in "while $m_{0|t}$ and $C_{0|t}$ gives"
- Typo in "PiDGM-D is not able to produce high quality ..."

**Strengths And Weaknesses:**

Strengths:

- The submission is well-motivated: existing methods are extremely sensitive to hyperparameter tuning in terms of their reconstruction quality.
- Providing theoretical guarantees is an important but often underestimated aspect of guidance methods.
- Assumptions and limitations are presented clearly.

Weaknesses:

- Scalability of the second moment estimate is the biggest limitation of the proposed method (as it is discussed in the text).
- Discussion of the theoretical results and their implications could be improved.
- Missing comparison with SNIPS (Kawar et al., 2021), which is cited in the text.

I will expand on these points below, and I am looking forward to discussing with the authors to clarify my questions.

---

> ### Author Response · Authors · 2024-09-08
> **Replies to reviewer uMGN**
>
> We thank the reviewer for the review and helpful comments. We reply below.
>
> > Discussion of the theoretical results and their implications could be improved
>
> Thank you for the comment, we have improved the discussion of the theoretical results and their implications with response to the related comments, below.
>
> > Proposition 3
> Could the authors expand on their claim "if there are no errors in the initial condition". What does this mean in practice? Is this equivalent to saying that the measurement should be noiseless?''
>
> We would like to clarify that "if there are no errors in the initial condition" is not equivalent to saying that the measurement should be noiseless. We provide our reasons for the caveat "no errors in the initial condition" for the setting of Theorem 1, below:
>
> In general, diffusion models should be started with initial distribution $z_0 \sim p_T$, where $p_T$ is the final distribution of the forward process when started in $p_0 = p_{\text{data}}$. However, this distribution is, in general, unavailable, therefore, one starts the diffusion model in $\mathcal{N}(0, I)$ instead. Since the forward process converges to $\mathcal{N}(0, I)$ this approximation is generally quite small. With the caveat we are saying that we are not treating this error term here. The resulting terms would make the theorem more technical. Since they are exponentially small in $T$, they are generally not the main contributor to the error, we decided to exclude them for readability but added the caveat "if there are no errors in the initial condition" to emphasise which case we are talking about.
>
> > Theorem 1 The symbol $\Phi$ is never defined, what is it referring to?
>
> $\Phi$ is an arbitrary function which measures how much $p_{\text{data}}$ differs from a Gaussian. Alternatively one could also define $\Phi$ as
> $$
> \exp(\Phi(x)) := \frac{p_\text{data}(x_0)}{\mathcal{N}(x_0; \mu_0, \Sigma_0)}.
> $$
> However we would like to emphasise that $\mu_0$ and $\Sigma_0$ can be chosen in an arbitrary way here. One would normally try to chose them so that the Lipschitz constant and size bounds ($L$ and $M$) are as small as possible.
>
> > I am not sure I fully understand the claim in the Theorem: is it bounding the total variation at the end of the reverse process ...
>
> Yes, it is bounding the total variation between the terminal distribution of the TMPD reverse process and the target, the conditional data distribution. We have stated in the Theorem that $q_{t}$ is the law of the corresponding reverse-time process for the reverse SDE sampling from the conditional distribution, so $q_{T}$ is the conditional distribution at the end of the TMPD reverse process.
>
> > I am not sure I fully understand the relation between between $t$ and $T$ in the bound. Could the authors expand on an intuitive discussion of the claim in the Theorem? For example, what happens at the same time t of the reverse process when starting from different $T$'s?
>
> $T$ is the final time of the forward process. To compare this result to other papers, note that those often use a time changed SDE
> \begin{equation}
>   d X_t = -\frac{1}{2}\beta(t) X_t dt + \sqrt{\beta(t)} d W_t
> \end{equation}
> which they run until time $1$. We do not change the time for the theory, since the results will stay the same (this is only a time change), but it clutters notation. In the other notation however, $T := \int_0^1 \beta(t) d t$. We hope this clarifies the role of $T$. The choice of $T$ (or in general $\beta(t)$, also called the "noising schedule") is an interesting topic which is an active area of research, but we didn't include the analysis in this paper since it would have been out of the scope.
>
>
> > Remark 1: In its current formulation, this is too strong of a claim. If the extensions are trivial, they should be included in the current manuscript. If not, more evidence needs to support the claim that all of these (ambitious) extensions can be attained.
>
> We do not think the extensions are trivial. The calculations would be rather technical and lengthy, but the calculations would be similar those done in the many recent works on proving convergence bounds for diffusion models.
> Including the SDE discretization would be rather technical, but not too challenging. We assume that our initial distribution is relatively close to a Gaussian, therefore the score does not explode, which is a common issue when discretizing the reverse SDE. Therefore, one could use stripped down versions of the proofs for diffusion models or plug in some classical SDE discretization arguments, since the score is globally Lipschitz.
> Adding the error in the initial conditions is a straightforward addition due to the data processing inequality. However, it would add more terms to the Theorem and make it less readable. However, if the reviewer wants us to add this, we can do so.
>
> Furthermore, we removed the remark as we agree with the reviewer that it might be more confusing to the reader than helpful in this form.

---

> ### Author Response · Authors · 2024-09-08
> **Replies to reviewer uMGN**
>
> > Equation 9 I would suggest to expand the discussion paragraph after the equation. In particular, all approximations used to obtain Eq. (9) should be clearly stated and made precise.
>
> Thank you for this comment. We have clarified the presentation of the approximations used to obtain Eq.(9). In particular, recall that we approximate the density of $x_0$ given $x_t$ as
> $$p_{0|t}(x_{0}| x_{t}) = \mathcal{N}(x_0; m_{0|t}, C_{0|t}),$$
> where $$m_{0|t} = \mathbb{E}[x_0 | x_t] = \frac{1}{\sqrt{\alpha_t}}(x_t + v_t \nabla_{x_t} \log p_t(x_t)),$$
> as given in our equation (6) of the manuscript and
> \begin{equation}
> C_{0|t} = \frac{v_{t}}{\sqrt{\alpha_{t}}} \nabla_{x_{t}} m_{0|t}
> \end{equation}
> as given in the equation (7) or our paper. It can be seen that since $m_{0|t}$ is a function of $x_t$, similarly $C_{0|t}$ is also a function of $x_t$. The resulting approximation of $p_{y | t}(y |x_t)$ is given by
> \begin{equation}
>  p_{y| t}(y| x_t) \approx \int p_{y | 0}(y |x_0) p_{0|t}(x_0 | x_t) \mathrm{d} x_t = \mathcal{N}(y; H m_{0|t}, H C_{0|t} H^{T} + \sigma^{2}_{y} I).
> \end{equation}
>
> We have now used the formulae above in our equation (11) to clarify the covariance matrix (instead of using $\nabla_{x_t} m_{0|t}$). We now clarify in the manuscript that, explicitly, the mean and the covariance are functions of $x_t$, i.e., we can write them as $m_{0|t}(x_t)$ and $C_{0|t}(x_t)$. If we write out explicitl, the log density is given by
>
> \begin{equation}
> \log p\_{y|t}(y|x\_{t}) = -(y - H m\_{0|t}(x\_t))^{\top}  (H C\_{0|t}(x\_t) H^{\top} + \sigma^{2}\_{y} I)^{-1}  (y - H m\_{0|t}(x\_t)) -\frac{1}{2} \log \det C\_{0|t}(x\_t)
> \end{equation}
>
> Due to $x_t$ dependence of both $m_{0|t}$ and $C_{0|t}$, the derivatives of this log density w.r.t. $x_t$ is too computationally demanding to compute. Therefore, we approximate the gradient at this point ignoring the $x_t$ dependence in $C_{0|t}$ which results in the approximation
>
> \begin{equation}
>     f^y(x_t) :=  \nabla_{x_{t}} m_{0|t}(x_{t}) H^{\top}(H {C_{0|t}}(x_{t}) H^{\top} + \sigma^{2}\_{y} I)^{-1}  (y - H m_{0|t}(x_t)).
> \end{equation}
>
> Another way to write this approximation is to use equation (7) and notice
>
> \begin{equation}
>     f^y(x_t) := \frac{\sqrt{\alpha_t}}{v_t} {C}\_{0|t}(x_t)H^{\top}(H {C\_{0|t}}(x_t)H^{\top} + \sigma^{2}\_{y} I)^{-1}  (y - Hm_{0|t}(x_t)).
> \end{equation}
>
> These details are now added to the manuscript on page 5, Section 3.2.

---

> ### Author Response · Authors · 2024-09-08
>
> > Approximation based on properties of $H$
> Could the authors expand on what other strategies could be employed in a general case where the operator $H$ may not be sparse?
>
> When $H : d_{y} \times d_{x}$ is not sparse, then one strategy is to compute $\text{min}(d_{y}, d_{x})$ vector Jacobian products to obtain the full $H \nabla_{x_{t}}m_{0|t}$ term, but of course this will be computationally expensive for large $\text{min}(d_{y}, d_{x})$, and naively computing the inverse of $H \nabla_{x_{t}}m_{0|t} H^{T}$ + $\sigma_{y}^{2}I$ is also memory intensive and prone to numerical error.
>
> Instead, we note a very recent work [3] that circumvents our heuristic by applying the conjugate gradient (CG) method, using the approximation $f^y(x_t)$ in practice for non-sparse $H$. We thank the reviewer for their comment and we have added the following discussion to Section 3.4 and Appendix E.1 on page 24, in light of the new work [3] that extends our method:
>
> "However, heuristics can be circumvented by noting in the definition of $f^y(x_t)$ does not require the inverse but rather solving a system of linear equations with right hand side $y - H m_{0|t}(x_{t})$. A very recent work [3], uses the natural choice of the conjugate gradient (CG) method to solve this linear system noting that $HC_{0|t}H^{T}$ + $\sigma_{y}^{2}I$ is symmetric positive definite (SPD) and is therefore compatible with the conjugate gradient (CG) method. The CG method is an iterative algorithm to solve linear systems of the form $m v = b$ where SPD matrix $m$ and vector $b$ are known. Importantly, the CG method only requires implicit access to $m$ through an operator that performs the matrix-vector product $m v$ given a vector $\mathbf{v}$. In our case, the linear system to solve is $y - H m_{0|t} = (H C_{0|t} H^{T} + \sigma_{y}^{2}I) v = H\frac{\sqrt{\alpha}}{v_{t}}\nabla_{x_{t}} m_{0|t} H^{T} v + \sigma_{y}^{2}I\mathbf{v}$. The vector-jacobian product $v^{T} H \nabla_{x_{t}}^{T} m_{0|t}$ can be cheaply evaluated. In practice, there is no restriction on the score network to be the gradient of a potential, and the gradient of the score need not be SPD. Due to this and numerical errors, [3] observed that CG method becomes unstable after a large number of iterations and fails to reach an exact solution. Fortunately, the authors find that using very few iterations (1 to 3) of the CG method as part of the computation of the posterior score approximation leads to significant improvements over using heuristics for the covariance. [3] have successfully applied the CG method to non-sparse $H$, such as accelerated MRI (where $H$ is the composition of the Fourier transform and frequency subsampling)."
>
> > "The mean is chosen the same way as our method." This sentence should be rephrased, as DPS introduced ideas of using Tweedie's estimate in inverse problems before the current submission.
>
> Thank you for the comment. We have changed this to "DPS introduced the use of Tweedie estimate, and we use the same approximation as them"
>
> > GMM experiment
> In the text, it is stated a few times "non-linear SDE" and "non-linear problems in high dimension", but I thought the submission was about linear inverse problems? In the Introduction, ""... a known linear observation map". What non-linearity are the authors referring to here?
>
> The reverse SDE will be a non-linear one if it is not started in a Dirac measure or a Gaussian distribution (since the drift term $\beta(T-t)\nabla_{z_{t}} \log p_{T-t}(z_{t})$ in Equation (2) is a nonlinear function of the random variable $z_{t}$), so in this case TMPD will not be exact. The non-linear problem refers to the non-linearity in the reverse SDE drift, from the neural network approximation of the score function. We thank the reviewer for helping us to clarify the writing by always stating "non-linear SDE" or "non-Gaussian problem" in place of "non-linear".
>
> > All equations should be numbered, including within Theorems.
>
> Thank you, we have now numbered equations in the updated PDF for Proposition 1 (Tweedie's formula), Proposition 2 (Moment projection) and Theorem 1 (General data distribution).
>
> > Minor comments
>
> Thank you for the minor comments, we have made corrections based on all of the minor comments, which has improved the overall quality of writing.
>
> We have also made further minor changes to the quality of writing of the paper, such as displaying the definition of $\alpha_{t}$ and the transition kernels of the VP SDE clearly in an equation, and correcting using “Then …” twice in Proposition 1 and “Optimally …” twice in Section 2.1.
>
> > Please conform cross reference formats: references to equations should be capitalized and have numbers in parentheses (e.g., "equation 2" $\rightarrow$ "Equation (2)").
>
> We follow the suggestion of capitalisation and that equations should have numbers in parentheses but we note that it required a change in the TMLR default 'math\_commands.tex' template that altered the defaults provided by TMLR.

---

> ### Author Response · Authors · 2024-09-08
> **Replies to reviewer uMGN**
>
> > Comparison with SNIPS. Continuing on my previous point of confusion, if this submission is only about linear inverse problems, an important comparison with SNIPS (Kawar et al., 2021) is missing. This comparison is important because SNIPS introduced a closed-form solution and efficient approximation of the exact reverse conditional likelihood for inverse linear problems (if the SVD of $H$ can be computed)
>
> Thank you for this suggestion. Here are the preliminary experimental results. We follow the experimental setup of DDRM [1] and PiGDM [2] using a pretrained ImageNet model from [4], evaluating on a specified 1000 images from ImageNet 1k validation set as specified in [1] on the $4 \times$ super-resolution task. We consider a noiseless case and $\sigma_{y}=0.05$ independent Gaussian noise. We have replicated the numerical results from [2] and shown the results from [2] for reference, giving us confidence in our results for $\Pi$GDM and that our numbers can be compared to SNIPS numbers from [1] in this table. From the results in [2], we had concluded that any diffusion based method would likely outperform SNIPS for the inverse problems we consider. Our preliminary experimental results of the diffusion methods outperform SNIPS on this task, providing further evidence of this. We did not find it necessary to include these results in the original manuscript, but have now added them as Table 6 in the Appendix.
>
> | Method                                 | 4 $\times$ noiseless super-resolution |                        |                 |                 | 4 $\times$ noisy super-resolution |                        |                 |                 |
> |----------------------------------------|--------------------------------------|------------------------|-----------------|-----------------|-----------------------------------|------------------------|-----------------|-----------------|
> |                                        | PSNR$\uparrow$                       | SSIM$\uparrow$         | FID$\downarrow$ | KID$\downarrow$ | PSNR$\uparrow$                    | SSIM$\uparrow$         | FID$\downarrow$ | KID$\downarrow$ |
> | SNIPS ([1] Table 1, 2 and [2] Table 9) | 17.58                                | 0.22                   | -               | 35.17           | 16.30                             | 0.14                   | -               | 67.77           |
> | $\Pi$GDM-D ([2] Table 9)               | -                                    | 0.73                   | -               | 1.24            | -                                 | -                      | -               | -               |
> | $\Pi$GDM-D                             | 26.0                   | 0.748  | 39.4            | 1.30            | 20.7                | 0.429  | 74.5            | 17.84           |
> | DPS-D                                  | 25.1                   | 0.694  | 59.4            | 4.08            | 23.4                | 0.628  | 48.4            | 3.09            |
> | TMPD-D                                 | 26.0                   | 0.747  | 63.1            | 1.10            | 23.1                | 0.624  | 62.9            | 8.85            |
>
>
> > Could the authors expand on their motivation not to compare with SNIPS and the advantages/limitations of TMPD compared to SNIPS? For example, could ideas of TMPD be extended to non-linear measurement models? This point should be made clearer, especially given the significant computational limitations of the submission (e.g., image results are shown on 32x32 pixels and 256x256 pixels images, whereas problems of practical relevance can be of the order of 1024x1024 pixels).
>
> The computation of SVD is costly and even prohibitive when the forward model gets more complex. For example, [1] only considered seperable Gaussian kernels for deblurring, since they were restricted to the family of inverse problems where they could effectively perform the SVD. Hence, the applicability of such methods is restricted, and it would be useful to devise a method to solve noisy inverse problems without the computation of the SVD.
>
> Our initial motivation not to compare with SNIPS is since $\Pi$GDM is shown to outperform SNIPS on $4 \times$ super-resolution on ImageNet1k (256 x 256) (see Tab. 9 in [2]), and we compare to $\Pi$GDM in our paper. We have reproduced this experiment in Table 1. We have also provided numerical results for noisy inverse problems $4 \times$ super-resolution for TMPD-D, $\Pi$GDM-D and DPS-D, where we make a comparison to SNIPS, and show our method has comparable performance to DPS-D and outperforms both SNIPS and $\Pi$GDM-D.

---

> ### Author Response · Authors · 2024-09-08
> **Replies to reviewer uMGN**
>
> **References**
>
> [1] Kawar, Bahjat, et al. "Denoising diffusion restoration models." Advances in Neural Information Processing Systems 35 (2022): 23593-23606.
>
> [2] Song, Jiaming, et al. "Pseudoinverse-guided diffusion models for inverse problems." International Conference on Learning Representations. 2023.
>
> [3] Rozet, François, et al. "Learning Diffusion Priors from Observations by Expectation Maximization." arXiv preprint arXiv:2405.13712 (2024).
>
> [4] Dhariwal, Prafulla, and Alexander Nichol. "Diffusion models beat gans on image synthesis." Advances in neural information processing systems 34 (2021): 8780-8794.

---

> > ### Comment · Reviewer_uMGN · 2024-09-12
> > **Thank you for your response!**
> >
> > I sincerely thank the authors for their detailed response---which has answered my questions---and for updating the manuscript. I have updated "Claims and evidence" in my preliminary review.
> >
> > ---
> >
> > I have a brief follow-up question on
> >
> > > However we would like to emphasise that $\mu_0$ and $\Sigma_0$ can be chosen in an arbitrary way here.
> >
> > Does this also include data-dependent ones?

---

> > > ### Author Response · Authors · 2024-09-17
> > > **Reply to follow up question**
> > >
> > > We sincerely thank the reviewer for reading our response and updating their preliminary review. We answer the follow up question below.
> > >
> > > Yes, they can be chosen in a data-dependent way. The Theorem holds for any choice of mu_0 and Sigma_0. In particular it also holds for the best choice of mu_0 and Sigma_0 ,which is the choice which makes L as small as possible and M as close to 1 as possible. Which mu_0 and Sigma_0 are optimal depends on the data distribution.

---

### Review · Reviewer_QhaZ · 2024-08-16

**Summary Of Contributions:**

The authors consider conditional sampling from diffusion models (DMs). They build on recent work that uses Tweedie's formula to approximate the reverse process, by estimating a full or diagonal covariance matrix. This enables exact sampling in the Gaussian case, new theoretical guarantees, and promising empirical results. The main downside is computational cost, although with approximations this is reasonable in the important inpainting and superresolution settings.

**Audience:**

Yes

**Claims And Evidence:**

Yes

**Requested Changes:**

Releasing code should be a requirement of publication since the methodology is reasonably involved so having a working implementation will be important for those wanting to try the approach out.

I don't know if it would be practical to compare to but the state-of-the-art work AFAIK in this space is https://arxiv.org/abs/2306.17775. It should at least be cited/discussed.

Minor comments:
I found the sentence "Optimally, we would like to sample from the reverse SDE to the forward SDE started in..." hard to grok.
In eq 9 I think f should be indexed by t.

**Strengths And Weaknesses:**

The paper is clearly written, although with a number of grammatical mistakes. I would suggest the authors run the text through grammarly or writefull or similar. I'm also a bit confused by the "inverse problem" intro/motivation. In principle the p(y|x) they consider here is only the linear case, which is not a very interesting inverse problem.

I didn't check the proofs carefully but the technical contributions appear sound, well motivated and backed by both theory and empirical results.

---

> ### Author Response · Authors · 2024-09-08
> **Replies to reviewer QhaZ**
>
> We thank the reviewer for the review and helpful comments. We reply to your comments below.
>
> > Strengths And Weaknesses: The paper is clearly written, although with a number of grammatical mistakes. I would suggest the authors run the text through grammarly or writefull or similar.
>
> Thank you for the comment, we have done as suggested and run the manuscript through a grammar checker, and corrected the grammatical errors. The writing itself has become much clearer.
>
> > I'm also a bit confused by the "inverse problem" intro/motivation. In principle the $p(y|x)$ they consider here is only the linear case, which is not a very interesting inverse problem.
>
> We agree and thank the reviewer for this suggestion, and reduce the length of the introduction to be narrowly targeted to the rest of paper.
>
> > Releasing code should be a requirement of publication since the methodology is reasonably involved so having a working implementation will be important for those wanting to try the approach out.
>
> We have attached both JAX and pytorch version in the supplementary material, which we plan to release on github as part of publication/ deanonymisation. We think that this code will be useful for those who would like to replicate both our method, and other baseline methods, such as PiGDM, for which the code has not been released.
>
> > I don't know if it would be practical to compare to but the state-of-the-art work AFAIK in this space is https://arxiv.org/abs/2306.17775. It should at least be cited/discussed.
>
> Thank you for this comment, we have added discussion of this work in Section 5, related work.
>
> "Finally, sequential Monte Carlo (SMC) methods are particle-based posterior sampling schemes that are asymptotically exact in the limit of infinitely many particles.  Cardoso et al. (2023) target the posterior for linear inverse problems, and our first experiment makes a comparison to  Cardoso et al. (2023) for a Gaussian Mixture model. However, for image experiments, this method can exhibit the well studied SMC problem of weight degeneracy, where particle weights collapse. We note a recent state of the art work (Wu et al., 2024) that tackles weight degeneracy by introducing the Twisted Diffusion Sampler (TDS), with heuristic twisting functions approximating $p_{y|t}(y | x_{t})$ that increase computational efficiency of SMC. The method is still asymptotically exact as long as the approximation of $p_{y|t}(y | x_{t})$ converges to $p(y|x_{0})$ as $t$ approaches $0$. The authors note that the efficiency of TDS (Wu et al., 2024) depends on how closely the twisting function approximates the exact likelihood. TMPD (i.e. Equation (10)) could be a good twisting candidate."
>
> > Minor comments: I found the sentence "Optimally, we would like to sample from the reverse SDE to the forward SDE started in..." hard to grok. In eq 9 I think f should be indexed by t.
>
> We have simplified this sentence and improved the readability:
> "we would like to sample from the reverse SDE targeting $p_\text{data}(\cdot| y)$, instead of the one targeting $p_\text{data}$."

---

### Review · Reviewer_ymX8 · 2024-08-31

**Summary Of Contributions:**

The paper presents a method to approximate the diffusion posterior more expressively than prior works. This approximation can solve inverse problems by leveraging pre-trained diffusion models as priors. The authors demonstrate empirical results on toy datasets and the FFHQ image dataset for super-resolution and inpainting tasks.

**Audience:**

Yes

**Broader Impact Concerns:**

Have been addressed.

**Claims And Evidence:**

Yes

**Requested Changes:**

**General Comments/Requested Changes:**

I don't have any major criticisms of this work (a primary concern of computational efficiency due to higher order gradients of the score function has already been addressed by the authors in the main text). I have some queries and comments on improving the paper further.

**On the technical background**: The authors state: “There are two main diffusion modeling paradigms, namely, score-based models (SGMs) (Song et al., 2020) and denoising diffusion probabilistic models (DDPM) (Ho et al., 2020)”. This seems misleading. A lot of prior work (including the DDPM [Ho et al.] and the SGM [Song et al.] papers) shows that these are not separate paradigms but are equivalent depending on the score/denoiser parameterization. Please rephrase this introductory statement appropriately, as you later mentioned that these are closely related.

This section generally seems dated, and references to more recent diffusions like EDM/EDM2 [Karras et al.] or interpolants [Albergo et al., Lipman et al.] are missing. It would be great if the authors could update this section appropriately.

Nit: Equations 1 and 2 can be converted to single-line equations for better readability. However, this is only my preference, and I leave this to the author's discretion.

**On Section 4**: While I appreciate the theoretical results, I don't think they add much value to the main text since the diffusion posterior is much more complicated than a simple Gaussian for any practical scenario I can imagine. In this context, it's best to move this section to the Appendix for better readability and faster access to the experimental section, where the empirical results on actual datasets are good.

**On empirical results**:
1. It's interesting that DPS and Pi-GDM don't quite work well with the VE-SDE process. Can the authors elaborate on whether the hyperparameters were tuned for the best performance for the DPS and Pi-GDM baselines for VE-SDE? If yes, is there any intuition on why these methods might not work in VE-SDE?
2. For the experimental setup, what is the number of sampling steps used for Pi-GDM? I’m guessing this is 1000 since the authors set the eta parameter in the DDIM sampler to 1.0. However, the optimal setting for Pi-GDM is to set the number of sampling steps to 100 using DDIM. It has been observed that Pi-GDM performs worse for 1000 steps (See Red-Diff [Mardani et al.]). I'm curious how the proposed method scales for lesser sampling budgets (like 100 steps) compared to Pi-GDM. Can the authors include results either for super-resolution or inpainting for this setting?
3. Are the results in Table 2 based on the diagonal variant of TMPD or the full covariance matrix? Based on the discussion in Section 3.4, these results appear for DTMPD. If yes, please change the method name in Table 2. If not, please add wall clock time estimates since, in that case, Pi-GDM would be significantly faster.

**On related works**: Discussion on some recent related works like Red-Diff [1], D-Flow [3], Pi-GDM for flows [2], inverse samplers for latent diffusion [4,5], and fast samplers for inverse problems [6] is missing. Please note that the mentioned list of works is not exhaustive, and I leave it to the authors to improve the discussion in this section.

[1] A Variational Perspective on Solving Inverse Problems with Diffusion Models - Mardani et al.

[2] Training-free Linear Image Inverses via Flows - Pokle et al.

[3] D-Flow: Differentiating through Flows for Controlled Generation - Hamu et al.

[4] Solving Linear Inverse Problems Provably via Posterior Sampling with Latent Diffusion Models - Rout et al.

[5] Beyond First-Order Tweedie: Solving Inverse Problems using Latent Diffusion - Rout et al.

[6] Fast Samplers for Inverse Problems in Iterative Refinement Models - Pandey et al.

**Strengths And Weaknesses:**

**Strengths**:

1. The paper is well-written and makes the method straightforward to understand
2. The authors address the computational drawbacks of this method in Section 3, which is nice.
3. The empirical results look better than the baselines, but I have some queries about this (see below).

**Weaknesses**: See the Requested Changes section.

---

> ### Author Response · Authors · 2024-09-08
> **Replies to reviewer ymX8**
>
> Thank you very much for your review and comments. We reply to your comments below.
>
> > The authors state: “There are two main diffusion modeling paradigms, namely, score-based models (SGMs) (Song et al., 2020) and denoising diffusion probabilistic models (DDPM) (Ho et al., 2020)”. This seems misleading. A lot of prior work (including the DDPM [Ho et al.] and the SGM [Song et al.] papers) shows that these are not separate paradigms but are equivalent depending on the score/denoiser parameterization. Please rephrase this introductory statement appropriately, as you later mentioned that these are closely related.
>
> We changed the introductory statement to "In score based generative models (SGMs) (Song et al., 2020) and denoising diffusion probabilistic models (DDPM) (Ho et al., 2020)" and "In the SGM paradigm, a stochastic differential equation (SDE) is used to noise the data, and the interpolation parameter $t$ will take continuous values in $t \in [0, T]$. In the DDPM setting, $t$ is discrete. However, the DDPM Markov chain can be seen as a discretization of the SDE (Song et al., 2020)." We hope that this is more clear.
>
> > This section generally seems dated, and references to more recent diffusions like EDM/EDM2 [Karras et al.] or interpolants [Albergo et al., Lipman et al.] are missing. It would be great if the authors could update this section appropriately.
>
> We have appended the following statement to this section "More recent advances in generative modelling include methods with artificial paths based on stochastic interpolants or continuous normalizing flows [Albergo et al., Lipman et al.]. Conditional flow matching draws inspiration from the denoising score matching approach, but generalizes to matching vector fields directly. We have focused our approach to denoising-diffusion models in the SGM paradigm because of readily available pretrained diffusion models."
>
> We have also included the statement "Recent developments have improved the noising schedule of score-based diffusion models for image data that parameterise the transition kernels of the forward process in terms of the signal-to-noise ratio $\sigma_{t}^{2}$, $p_{t}(x_{t}| x_{0}) = \mathcal{N}(x_{t}; s_{t}x_{0}, s_{t}^{2}\sigma_{t}^{2} I_{d_{x}})$ [Karras et al.]."
>
> > Equations 1 and 2 can be converted to single-line equations for better readability. However, this is only my preference, and I leave this to the author's discretion.
>
> Thank you for this suggestion, and we have changed all of the SDEs to this format, as it shortens the length of the paper and aids readability. The extra space allowed us to put the tables in the main text in to normal font size, which also improves the readability of the paper.
>
> > On Section 4: While I appreciate the theoretical results, I don't think they add much value to the main text since the diffusion posterior is much more complicated than a simple Gaussian for any practical scenario I can imagine. In this context, it's best to move this section to the Appendix for better readability and faster access to the experimental section, where the empirical results on actual datasets are good.
>
> We are happy to move the section to the Appendix, if the other reviewers would also agree. We would like to ask the other reviewers for their opinion and if there is consensus that it would increase the quality of the paper, then we will be happy to do so.

---

> ### Author Response · Authors · 2024-09-08
> **Replies to reviewer ymX8**
>
> > On empirical results: It's interesting that DPS and Pi-GDM don't quite work well with the VE-SDE process. Can the authors elaborate on whether the hyperparameters were tuned for the best performance for the DPS and Pi-GDM baselines for VE-SDE? If yes, is there any intuition on why these methods might not work in VE-SDE?
>
> **DPS for VE-SDE**
>
> For the VE SDE, we tune the DPS hyperparameters for best performance. We have set up DPS-D properly, since we port the code and models from the original pytorch repository that the DPS paper used for VP SDE, and have, for each inverse problem (each row of each table), tuned the DPS scale hyperparameter on a 128 size validation set to optimize LPIPS, PSNR, MSE and SSIM (see Fig. 9 for an example).
>
> However, we find it difficult to reasonably adapt the DPS algorithm to the VE setting, since the DPS step-size hyperparameter in "Algorithm 1 DPS - Gaussian" [7] does not directly follow the theoretical reasoning (the delta approximation for $x_{0}|x_{t}$) in [7], and so the hyperparameter was empirically tuned for the VP SDE especially. In short, the scaling compared to the (inverse of the) covariance $C_{y y} = \mathbb{E} [ (y - \mathbb{E}(y))(y - \mathbb{E}(y)^{\top} ]$ (step size) in their paper will not work for VE SDE, and we explain our reasoning below.
>
> The heuristic step-size hyperparameter in DPS is derived through experimentation for the VP SDE, as can be seen in the implementation
> \begin{align}
>     x_{n-1}' \leftarrow \frac{\sqrt{1 - \beta_{n}} (1 - \alpha_{n-1})}{1 - \alpha_{n}} x_{n} + \frac{\sqrt{\alpha_{n-1}}\beta_{n}}{1 - \alpha_{n}}m_{0|t} + \sigma_{n} z_{n}
> \end{align}
>
> \begin{align}
>     x_{n-1} \leftarrow x_{n-1}' - \psi \nabla_{x_{n}} \|y - H m_{0|t} \|_{2},
> \end{align}
>
> where $\psi$ is a DPS hyperparameter, and is usually set for each problem, in the range $\psi \in [0.3, 1.0]$. Comparing this algorithm to TMPD,
> \begin{align}
>     m_{0|t}^{y} \leftarrow m_{0|t}  + \frac{{1 - \alpha_{n}}}{\sqrt{\alpha_{n}}}\nabla_{x_{n}}m_{0|t} H^{\top} (H\frac{{1 - \alpha_{n}}}{\sqrt{\alpha_{n}}}\nabla_{x_{n}}m_{0|t}H^{\top} + \sigma_{y}^{2} I_{d_{y}} )^{-1}(y - H m_{0|t})
> \end{align}
> \begin{align}
>     x_{n-1} \leftarrow \frac{\sqrt{1 - \beta_{n}} (1 - \alpha_{n-1})}{1 - \alpha_{n}} x_{n} + \frac{\sqrt{\alpha_{n-1}}\beta_{n}}{1 - \alpha_{n}}m_{0|t}^{y} + \sigma_{n} z_{n}
> \end{align}
>
> we first apply the Bayes update on $m_{0|t}$ and then do DDPM. We can successfully apply the same logic to SMLD, which is a discretization of VE. On the other hand, when we apply the reasoning of the DPS algorithm to SMLD, the algorithm fails, since the scaling no longer corresponds to the variance $C_{y y}$. To state this in more concrete terms, we can compare the term in TMPD $\frac{\sqrt{\alpha_{n-1}}\beta_{n}}{1 - \alpha_{n}} \frac{{1 - \alpha_{n}}}{\sqrt{\alpha_{n}}}\nabla_{x_{n}}m_{0|t} H^{\top} C_{y y}^{-1}(y - H m_{0|t})$ to the term in DPS $-\psi \nabla_{x_{n}} \|y - H m\_{0|t} \|\_{2} = \psi \nabla m\_{0|t} H^{\top} \frac{1}{\|y - H m\_{0|t} \|\_{2}} (y - H m\_{0|t})$ where $C_{y y} = (H\frac{{1 - \alpha_{n}}}{\sqrt{\alpha_{n}}}\nabla_{x_{n}}m_{0|t}H^{\top} + \sigma_{y}^{2} I)$.
>
> We can eliminate like terms to compare the scaling of $C_{y y}$ for each method. $\frac{1}{\beta_{n}}C_{y y}^{-1}$ to $\psi \frac{1}{|y - H m\_{0|t} |\_{2}}$. There are two surprises here: whilst $\frac{1}{d\_{y}}\|y - H m\_{0|t} \|^{2}\_{2}$ has the scale of an empirical sample variance and so is the same scale as $C_{y y}$, the term used is it's square root. Secondly, the $\frac{1}{\beta_{n}}$ term is missing for DPS. A similar analysis on a discretization of VE would show that the variance $C_{y y}$ has the incorrect scaling in the DPS algorithm.

---

> ### Author Response · Authors · 2024-09-08
> **Replies to reviewer ymX8, continued**
>
> **PiGDM for VE-SDE**
>
> The PiGDM methodology and algorithm are similar to ours in the derivation of step-size, so there is no need to select a step-size hyperparameter for the PiGDM method. However, PiGDM makes the assumption that the image distribution is standard normal distribution. For VE SDE we use $r\_{t} = \frac{\sigma^{2}\_{t} \sigma\_{\text{data}}^{2}}{\sigma^{2}\_{t}  + \sigma^{2}\_{\text{data}}}$ as recommended in section 'A.3 About the variance of the approximation' in [8].
>
> We found this hyperparameter to work well (sometimes outperforming DPS-D and DTMPD-D) in the noiseless setting, but we observe visual artifacts in the noisy setting. DTMPD-D outperforms PiGDM-D in the noisy setting. For the VP-SDE, we use exactly 'Algorithm 1 PiGDM for VP-SDE' in [2], with the same derivation of $r_t$, but adapted to the VP setting, $r_{t} = \frac{v_{t} \sigma_{\text{data}}^{2}}{v_{t}  + \alpha_{t} \sigma^{2}_{\text{data}}}.$ Again, we found this hyperparameter to work well (outperforming DPS-D and DTMPD-D) in the noiseless setting, but we observe visual artifacts in the noisy setting. DTMPD-D outperforms PiGDM-D in the noisy setting.
>
> We believe that part of the successful adaptability of our algorithm to DDPM, DDIM, VP and VE settings for noise and noiseless is because we apply a Bayes update to $m_{0|t}$ in each algorithm, which works well empirically. Please see our results for the DDIM setting, below.
>
> Finally, we note that we found that static thresholding (clipping the denoised image estimate to the range of the data, e.g. [-1, 1] in [9] at each sampling step) is critical for the stability and performance of both DPS and PiGDM, but not TMPD.

---

> ### Author Response · Authors · 2024-09-08
>
> > For the experimental setup, what is the number of sampling steps used for Pi-GDM? I’m guessing this is 1000 since the authors set the eta parameter in the DDIM sampler to 1.0. However, the optimal setting for Pi-GDM is to set the number of sampling steps to 100 using DDIM. It has been observed that Pi-GDM performs worse for 1000 steps (See Red-Diff [Mardani et al.]). I'm curious how the proposed method scales for lesser sampling budgets (like 100 steps) compared to Pi-GDM. Can the authors include results either for super-resolution or inpainting for this setting?
>
> This is correct, we use 1000 steps for all experiments in the paper. Here are preliminary results for super-resolution and inpainting for FFHQ with DDIM using VE SDE. We note that since we have used $\eta = 1.0$, this makes DDPM and DDIM equivalent.
>
> Table: Quantitative evaluation of solving linear inverse problems for VE DDPM for 100 steps with increasing noise on FFHQ 256$\times$256-1k validation dataset. (See Table 11 in the updated PDF for full results).
>
> | Problem                               | Method     | FID $\downarrow$ | LPIPS $\downarrow$ | MSE $\downarrow$ | PSNR $\uparrow$ | SSIM $\uparrow$ |
> |---------------------------------------|------------|------------------|--------------------|------------------|-----------------|-----------------|
> | $\sigma_{y}=0.01$                   | DTMPD-D     | **38.8**      | **0.215**     | 0.001            | **28.9**   | **0.821**  |
> | $4 \times$ `bicubic' super-resolution | $\Pi$GDM-D | 46.1             | 0.252              | 0.002            | 27.4            | 0.788           |
> | $\sigma_{y}=0.05$                   | DTMPD-D     | **39.2**    | **0.279**     | 0.003            | **26.2**   | **0.741**  |
> | $4 \times$ `bicubic' super-resolution | $\Pi$GDM-D | 43.3             | 0.302              | 0.003            | 25.1            | 0.722           |
> | $\sigma_{y}=0.1$                    | DTMPD-D     | **41.1**    | **0.317**     | 0.004            | **24.5**   | **0.693**  |
> | $4 \times$ `bicubic' super-resolution | $\Pi$GDM-D | 44.7             | 0.344              | 0.005            | 23.4            | 0.672           |
> | $\sigma_{y}=0.01$                   | DTMPD-D     | 31.4             | 0.124              | 0.003            | 26.5            | 0.906           |
> | `box' mask inpainting                 | $\Pi$GDM-D | **23.7**    | **0.069**     | 0.002            | **28.0**   | **0.925**  |
> | $\sigma_{y}=0.05$                   | DTMPD-D     | 38.2             | 0.199              | 0.003            | 26.0            | **0.848**  |
> | `box' mask inpainting                 | $\Pi$GDM-D | **29.8**    | **0.185**     | 0.002            | **26.7**   | 0.825           |
> | $\sigma_{y}=0.1$                    | DTMPD-D     | **39.7**    | **0.241**     | 0.004            | **25.2**   | **0.799**  |
> | `box' mask inpainting                 | $\Pi$GDM-D | 42.1             | 0.279              | 0.003            | 25.1            | 0.716           |
>
> We observe with these results that our algorithm with DDPM remains competitive with PiGDM with the optimal setting of 100 steps. This is contrary to DPS, which does not outperform PiGDM at 100 steps [8], which we believe is due to the difference between our algorithm and the DPS algorithm outlined in the above answer "**DPS for VE-SDE**" that makes our method easily applicable to DDPM, DDIM and across different SDEs and noise levels.

---

> ### Author Response · Authors · 2024-09-08
> **Replies to reviewer ymX8**
>
> > Are the results in Table 2 based on the diagonal variant of TMPD or the full covariance matrix? Based on the discussion in Section 3.4, these results appear for DTMPD. If yes, please change the method name in Table 2. If not, please add wall clock time estimates since, in that case, Pi-GDM would be significantly faster.
>
> The results in Table 2 are based on the diagonal variant of TMPD, DTMPD-D. We have corrected the method name from TMPD-D to DTPMD-D in Table 2, and in the Appendix tables where the same applied.
>
> > On related works: Discussion on some recent related works like Red-Diff [1], D-Flow [3], Pi-GDM for flows [2], inverse samplers for latent diffusion [4,5], and fast samplers for inverse problems [6] is missing. Please note that the mentioned list of works is not exhaustive, and I leave it to the authors to improve the discussion in this section.
>
> We have mentioned Red-Diff [1] in the introduction "variational approaches". We depart from flow related works in the technical background because of the focus on the use of pretrained diffusion models. However, extending TMPD to flows is certainly a possible direction, so we have added the following to the future work section.
>
> "Using flow based methods [Albergo et al., Lipman et al.] as unconditional priors and building conditional sampling methods that leverage flows as pretrained models [3, 6] is related to the approach used in this paper. For example, it is possible to directly substitute in the TMPD approximation to the PiGDM method applied to flows [2], which would be a fruitful direction for future work."
>
> Extending TMPD to nonlinear observation operators with observation noise is not immediately obvious and may require extended-Kalman type approximations. Therefore, as explained in the future work section, our method is not directly applicable to latent diffusion and so we will leave the latent diffusion references out as we think that including them would be confusing.
>
> **References**
>
> [1] A Variational Perspective on Solving Inverse Problems with Diffusion Models - Mardani et al.
>
> [2] Training-free Linear Image Inverses via Flows - Pokle et al.
>
> [3] D-Flow: Differentiating through Flows for Controlled Generation - Hamu et al.
>
> [4] Solving Linear Inverse Problems Provably via Posterior Sampling with Latent Diffusion Models - Rout et al.
>
> [5] Beyond First-Order Tweedie: Solving Inverse Problems using Latent Diffusion - Rout et al.
>
> [6] Fast Samplers for Inverse Problems in Iterative Refinement Models - Pandey et al.
>
> [7] Chung, Hyungjin, et al. "Diffusion posterior sampling for general noisy inverse problems." arXiv preprint arXiv:2209.14687 (2022).
>
> [8] Song, Jiaming, et al. "Pseudoinverse-guided diffusion models for inverse problems." International Conference on Learning Representations. 2023.
>
> [9] Song, Yang, et al. "Score-based generative modeling through stochastic differential equations." arXiv preprint arXiv:2011.13456 (2020).

---

> > ### Comment · Reviewer_ymX8 · 2024-09-18
> > **Response**
> >
> > Thanks for your response, which clarified my concerns.

---

### Decision · Action_Editor_WZSd · 2024-09-19

**Recommendation:** Accept as is

**Comment:**

This work develops a method to better tackle conditional sampling in diffusion models when approximating the inverse/reverse process.  The method takes advantage of Tweedie's formula to enable exact sampling under Gaussian assumptions and provide new theoretical insights.  The authors demonstrate the method using a GMM example and image inpainting and superresolution problems.  Overall, the reviewers all found that the paper was well written (modulo some typos and grammar issues), interesting to the community, novel and sound.  The main concern before the rebuttal period seemed to be regarding discussion of recently related work, but the authors appear to have sufficiently addressed this in their response.  The remaining concern, highlighted by all reviewers, is that the proposed method is computationally expensive and thus the practical usefulness of the approach might be limited.   They suggested that developing efficient approximations would make the work more compelling and impactful.  Nonetheless, the reviewers all recommended to accept the paper.

**Audience:**

Diffusion models are a highly active area of research in generative modeling currently, with practical applications.  Thus this seems quite on topic for TMLR.

**Claims And Evidence:**

The reviewers all found that the the claims were well supported by evidence.